# Genkwanin glycosides are major active compounds in *Phaleria nisidai* extract mediating improved glucose homeostasis by stimulating glucose uptake into adipose tissues

Carla Horvath[1,11], Joëlle Houriet[2,3,11], Alexandra Kellenberger[1,11], Caroline Moser[1], Lucia Balazova[1,4], Miroslav Balaz[1,4,5], Hua Dong[6], Aron Horvath[7], Isabel Reinisch[1], Vissarion Efthymiou[1], Adriano Rutz[2,3], Laurence Marcourt[2,3], Christopher Kitalong[7], Bertrand Graz[3,8,9], Victor Yano[8], Emerson Ferreira Queiroz[2,3], Jean-Luc Wolfender[2,3] ✉ & Christian Wolfrum[1,10] ✉

Natural remedies are used as standalone treatments or complementary to modern medicine to control type 2 diabetes. In Palau, the traditional leaf decoction of *Phaleria nisidai* (PNe) is selected to treat hyperglycemia and its efficacy has been supported by a small clinical trial. As part of a reverse pharmacology approach, we here investigated the anti-diabetic potential of PNe and its bioactive compounds to alleviate insulin resistance in diet-induced obese, male mice. Dietary supplementation with PNe improves insulin sensitivity and promotes glucose uptake into adipose depots. In vitro, PNe triggers glucose disposal into murine and human adipocytes by upregulating *Glut1* expression through PKC-ERK1/2 signaling. To identify active constituents in PNe, we conducted bioactivity-guided fractionations and deciphered genkwanin flavone glycosides as bioactive principles. Moreover, we demonstrate that the aglycone genkwanin (GE) improves insulin resistance to a comparable extent to the anti-diabetic drug, metformin. Our findings present GE as promising glucoregulatory phytochemical that facilitates glucose uptake into adipocytes, thereby reducing systemic glucose load and enhancing insulin sensitivity.

Global diabetes prevalence has quadrupled since 1980, causing reduced life quality and life expectancy. Type 2 diabetes mellitus (T2DM) accounts for 90–95% of diabetes cases, with obesity being the strongest risk factor[1]. Defective blood glucose clearance mechanisms contribute to hyperglycemia, the hallmark of T2DM[2]. In healthy skeletal muscle and adipose tissue, glucose transporter 4 (GLUT4) ensures the efficient removal of postprandial glucose in response to insulin. However, impaired insulin signaling leads to dysfunctional GLUT4 shuttling from intracellular vesicles to the plasma membrane (PM) and blunted glucose influx through GLUT4[3,4]. Apart from GLUT4,

adipocytes and myocytes express the insulin-independent glucose transporter 1 (GLUT1), whose physiological relevance in systemic glucose homeostasis is less well studied[5]. Corresponding tissue-specific knockout mice have not been characterized and homozygous *Glut1*-null mice are not viable, while global heterozygous *Glut1* deficiency results in a high degree of still birth and severe developmental impairment[6]. In skeletal muscle, reduced GLUT1 protein levels are observed in insulin-resistant individuals, leading to blunted basal glucose uptake[7,8]. An equivalent reduction in baseline glucose uptake has been reported in white adipose tissue (AT) of Wistar fatty rats and tissue explants isolated from individuals with obesity[9,10]. Various chemical or hormonal cues can promote GLUT1 PM-translocation and glucose entry into the cell. For example, β3-adrenergic stimulation of brown adipocytes induces mTORC2-dependent GLUT1 PM-incorporation and insulin has been reported to mildly trigger GLUT1 shuttling in 3T3-L1 adipocytes[11,12]. In endothelial cells, PKC phosphorylates GLUT1 at S226 in response to VEGF, which targets GLUT1 to the PM and facilitates glucose uptake[13]. Similarly, the PKC agonist PMA has been shown to stimulate glucose disposal into 3T3-L1 adipocytes by upregulating *Glut1* expression and by increasing GLUT1 cell surface levels[14]. Taken together, these examples illustrate that GLUT1 in adipose tissue and skeletal muscle might have an impact on glucose homeostasis and represents a poorly investigated route to mitigate hyperglycemia.

Diabetic patients rely on pharmacotherapy to achieve glycemic control when efforts to increase physical activity and improved diet appear insufficient. Modern T2DM drug classes such as SGLT2-inhibitors, biguanides, or GLP-1 receptor agonists originate from natural products[15]. Whether by necessity or choice, populations worldwide use medicinal plants to treat diseases, which provide a valuable pool of potentially bioactive compounds. However, the effectiveness and chemical composition of such remedies are rarely evaluated[16]. Reverse pharmacology describes a drug discovery process, which prioritizes traditional knowledge and human clinical efficacy, followed by chemical characterization, bioactivity assays, and mode of action studies in suitable disease models to discover active substances and define a lead compound[17]. A main argument in favor of this approach is that the clinical evaluation in terms of safety and efficacy is carried out before laboratory studies, which increases the chances to pinpoint a constituent with drug-like quality.

The Republic of Palau is heavily affected by obesity and other Non-Communicable Diseases due to rapid lifestyle changes post second World War. Due to the limited accessibility to modern treatment regimes, the search for effective local remedies through scientific validation was initiated prior to 2007. A program was launched in 2013 to identify currently used Palauan therapies as antidiabetic interventions based on the population's traditional habits[18]. A retrospective-treatment outcome study highlighted a putative effect of the *Phaleria nisidai* (PN) Kaneh (Thymelaeaceae) leaf decoction on glycaemia and a subsequent randomized, double-blind crossover trial confirmed this decoction as an adjuvant to stabilize diabetic patients with insufficient glycemic control[19]. Here, we show that the *Phaleria nisidai* leaf extract (PNe) compensates for insulin resistance in diet-induced obese mice by elevating glucose uptake into adipose depots partially via GLUT1 and enhancing AKT phosphorylation at S473 in both brown and white AT. Additionally, we identify genkwanin glycosides as the major active components, which confers part of the beneficial properties of this plant decoction and improves glucose homeostasis with a similar efficacy as metformin.

## Results

### Dietary supplementation with a PN extract (PNe) improves glucose homeostasis in DIO mice

Based on previous findings, which have reported a glycated hemoglobin-lowering effect for a PN decoction[19], we first aimed to reproduce these beneficial effects on glucose control using a diet-induced obese (DIO) mouse model paired with a dietary intervention (DI) (Supplementary Fig. 1a, b). To closely adapt the traditional preparation from Palau and preserve the chemical composition of the PN drink, a leaf decoction was prepared and freeze-dried to generate the PN extract (PNe). After an initial high-fat diet (HFD) feeding period of 12 weeks (Fig. 1a), obese and insulin-resistant animals were randomly allocated to weight-matched groups and subsequently fed control HFD or HFD enriched with PNe (130 mg/kg body weight (BW)). The mouse dose was calculated from an estimated human intake of 50 mg/kg BW using the human equivalent dosage formula, which corresponded to a lyophilized decoction dose of 10.5 mg/kg × day[20]. Food intake measurements showed that there was no food aversion due to PNe supplementation (Supplementary Fig. 1c). In accordance with the reported human data[19], there was no differences in weight gain or body composition between the two groups (Fig. 1b–d). However, PNe-fed mice were more insulin sensitive after 5 weeks of DI (Fig. 1e, f). PNe feeding reduced fasting insulin (Fig. 1g) concentrations without altering fasting blood lipid parameters (Fig. 1h–j). We next shortened the initial HFD-feeding period to 6 weeks to explore whether PNe supplementation exerts preventive effects if introduced during early onset of insulin resistance (Fig. 1k). Weight gain was not altered with PNe during the DI (Supplementary Fig. 1d, e). Although PNe did not significantly improve oral glucose tolerance after only two weeks intervention (Fig. 1l, m), we detected lower glucose-stimulated plasma insulin concentrations 30 min after the oral glucose load (Fig. 1n), which supports an improvement in insulin sensitivity[21]. PNe furthermore dampened urinary glucose excretion (Fig. 1o). As we could show that PNe regulates systemic glucose homeostasis, we investigated if PNe-supplementation increases glucose uptake into metabolically relevant tissues using [14]C-2-deoxyglucose tracing and counted accumulated phospho-deoxyglucose in the fasted state (Fig. 1p–u). We did not detect a difference in glucose disposal into skeletal muscles (Fig. 1q, r). However, higher amounts of glucose uptake were observed in inguinal white adipose tissue (iWAT) and a trend ($p = 0.085$) for the interscapular brown adipose tissue (iBAT) (Fig. 1s, t), but not in the epididymal depot (eWAT) (Fig. 1u). Fasting blood insulin levels remained unchanged between PNe and HFD-fed mice in a comparable, independent experiment, indicating that the observed differences in adipose tissue glucose uptake rates are not due to changes in insulin levels (Supplementary Fig. 1f). These results from chronic PNe treatment regimens suggest that PNe regulates blood glucose concentrations, by directly stimulating signaling pathways and/or by inducing a long-term adaptive response cumulating in improved insulin sensitivity.

### PNe promotes glucose uptake by upregulating GLUT1 protein in adipocytes

To gain insights into the molecular mechanisms underlying the elevated insulin sensitivity observed in vivo and considering our glucose tracing data, we assessed the effects of PNe on adipocyte glucose metabolism using immortalized murine white (3T3-L1) and brown adipocyte (iBAs) cell lines. After 3 days of daily PNe treatment (200 μg/mL), we detected higher insulin-stimulated [14]C-2-deoxyglucose uptake into white (Fig. 2a) and brown (Fig. 2b) adipocytes. Interestingly, we noted a 30% increase in insulin-independent glucose disposal (Fig. 2a, b). These measurements suggest that PNe does not primarily act on the insulin-dependent arm of glucose clearance but rather boosts basal glucose uptake. Consistent with augmented baseline glucose uptake, GLUT4 protein content was unchanged by PNe, whereas GLUT1 protein was upregulated (Fig. 2c–g). We next examined if PNe potentially modulates the phosphorylation state of Protein Kinase B (AKT) at S473 and T308, which controls the translocation of insulin-responsive GLUT4 to the plasma membrane. In absence of insulin, AKT T308 phosphorylation increased in iBAs but decreased in 3T3-L1 adipocytes. This opposing observation

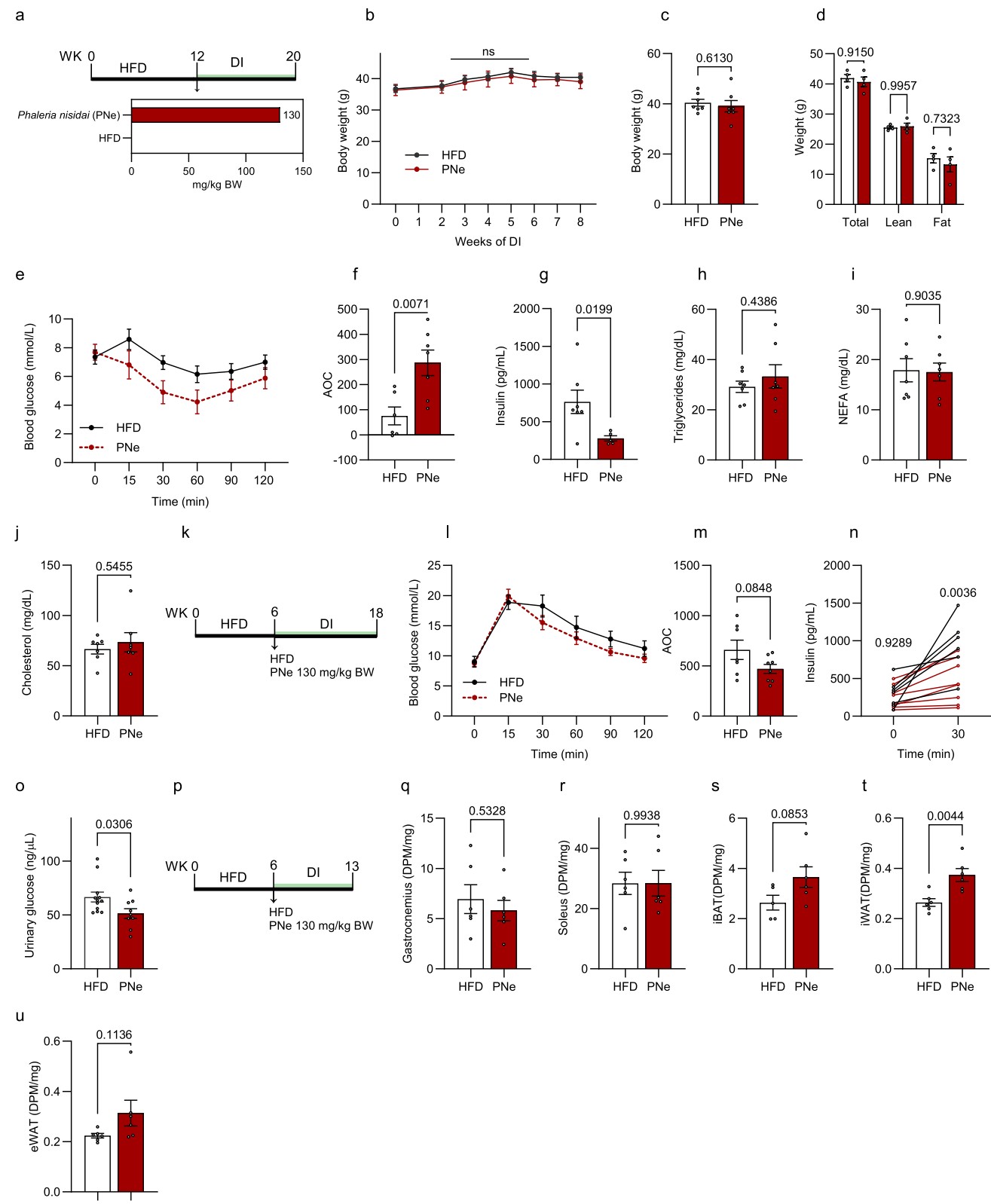

may be an artefact from very low basal phosphorylation at T308 levels, which compromises quantification accuracy. Moreover, insulin-mediated phosphorylation of T308 was not affected by PNe exposure (Fig. 1h–j). PNe blunted AKT S473 phosphorylation in the insulin-stimulated condition (Fig. 2h, k, l), implying that increased glucose disposal is likely mediated by GLUT1, which acts independently of insulin. To assess post-uptake metabolization, we approximated

glycolysis from the extracellular acidification of the media, which demonstrated that PNe dose-dependently increased glycolysis as well as glycolytic capacity in both white and brown adipocytes (Fig. 2m–r). In turn, lipolysis was not affected in iBAs by PNe treatment (Supplementary Fig. 2a, b). To substantiate our findings, we directly measured glycolytic flux using $^3$H-labelled glucose. We could corroborate that PNe treatment for 3 days significantly enhanced glycolytic rate by more than

**Fig. 1 | PNe alleviates glucose homeostasis in diet-induced obese mice.**
**a** Experimental design for (**b**–**j**). **b** Body weight development. HFD $n = 7$, PNe $n = 7$.
**c** Final body weight. DI = 8 weeks. HFD $n = 7$, PNe $n = 7$. **d** Body composition.
DI = 5 weeks. HFD $n = 4$, PNe $n = 4$. **e** ITT and **f** corresponding AOC. DI = 5 weeks.
HFD $n = 6$, PNe $n = 7$. **g**–**j** Fasting plasma **g** insulin (HFD $n = 7$, PNe $n = 5$),
**h** triglyceride (HFD $n = 7$, PNe $n = 7$), **i** NEFA (HFD $n = 7$, PNe $n = 7$), and **j** cholesterol
(HFD $n = 7$, PNe $n = 7$) concentrations. DI = 8 weeks. **k** Experimental design for (**l**–**o**).
**l** oGTT and **m** corresponding AOC. DI = 2.5 weeks DI. HFD $n = 7$, PNe $n = 8$. **n** Plasma
insulin levels during oGTT in **l**, **m** at baseline (HFD $n = 7$, PNE $n = 8$) and 30 min after
the glucose load (HFD $n = 6$, PNe $n = 8$). **o** Urinary glucose concentration.
DI = 6 weeks. (HFD $n = 12$, PNe $n = 9$). **p** Experimental design for (**q**–**u**). **q**–**u** Tissue
specific $^{14}$C-2-deoxyglucose glucose uptake into **q** gastrocnemius, **r** soleus, **s** iBAT,
**t** iWAT, and **u** eWAT. DI = 7 weeks. HFD $n = 6$, PNe $n = 6$ for all tissues except BAT.

For iBAT, HFD $n = 5$, PNe $n = 6$. Results are reported as mean ± SEM. Two-tailed
student's $t$-test for comparisons between two groups was applied in
(**c**, **f**–**j**, **m**, **o**, **r**–**u**). Two-way ANOVA with Sidak's post-hoc test was applied in (**d**).
Repeated measures two-way ANOVA with diet × time interaction and Sidak's post-
hoc test for each time point was applied in (**b**). Repeated measures mixed-effects
analysis with diet × time interaction and Sidak's post-hoc test for each time point
was applied in (**n**). Statistical test results are indicated as exact $p$-values with
*$p < 0.05$ considered significant. Source data are provided in Source Data 1 AOC
area of the curve, NEFA non-esterified fatty acids, DPM decays per minute, DI
dietary intervention, PNe *Phaleria nisidai* extract, iWAT inguinal white adipose
tissue, eWAT epididymal white adipose tissue, iBAT interscapular brown adipose
tissue, ITT insulin tolerance test, oGTT oral glucose tolerance test.

25% (Fig. 2s, t). A concurrent upregulation of glycolytic genes (*Pdk*, *Pgk*)
in iBAs (Fig. 2u) underlines that PNe not only promotes adipocyte glu-
cose uptake but also glucose utilization. In addition, we detected an
upregulation of *Glut1* mRNA expression in response to PNe treatment
(Fig. 2u, v). Of note, the gene expression analysis was performed after
3 days of treatment, more pronounced effects on *Glut1* expression were
observed after acute stimulation (4 h, Fig. 3d). We could recapitulate the
main effects of PNe on glucose uptake (Supplementary Fig. 2c, d),
GLUT1 protein content (Supplementary Fig. 2e–i), glycolytic activity
(Supplementary Fig. 2j–o) as well as gene expression signature includ-
ing additional genes of the glycolytic pathway such as *Gpi*, *Aldoa*, *Eno1*,
*Pgam1* (Supplementary Fig. 2p, q) in primary murine adipocytes dif-
ferentiated from the stromal vascular fraction of the iWAT and iBAT
depots. Certain genes encoding subunits of the pyruvate dehy-
drogenase complex that connects glycolysis to the citric acid cycle were
downregulated in primary white adipocytes, suggesting that glycolysis
may be the main ATP-source in PNe-treated adipocytes. Contrasting the
results from the immortalized cell lines, GLUT4 protein levels were
reduced in the primary white adipocytes (Supplementary Fig. 2e, g), but
no other adipogenic markers were significantly affected (Supplemen-
tary Fig. 2l, m).

As the PNe is effective in humans, we tested the extract on human
multipotent adipose derived stem (hMADS) cells differentiated into
white and beige adipocytes. We could validate higher GLUT1 and
GLUT4 protein levels stimulated by PNe in beige but not white hMADS
(Supplementary Fig. 3a–c). Similarly, the dose-dependent enhance-
ment of glycolysis and glycolytic capacity was only observable in beige
hMADS (Supplementary Fig. 3d–g). These findings were consistent
with a 38% increase in basal glucose uptake rates in beige hMADS,
which was sustained during insulin stimulation (Supplementary
Fig. 3h). Since PNe was ineffective on the white hMADS, we included in
vitro differentiated primary subcutaneous human adipocytes as sec-
ond human model. Here, we could confirm increased glucose uptake
rates (Supplementary Fig. 3i) paralleled by elevated GLUT1 levels in
response to PNe (Supplementary Fig. 3j, k, m). However, GLUT4 levels
were reduced, consistent with the data from primary murine white
adipocytes (Supplementary Fig. 3j–l).

Our findings provide evidence that higher GLUT1 protein levels
induced by PNe derive from increased *Glut1* transcription. We could
translate these findings to our mouse model (Fig. 2w–y), where we
detected higher GLUT1 protein content (Fig. 2x) in the iWAT (+50%,
$p = 0.032$) and iBAT (+18% increase, $p = 0.035$) of PNe-fed animals
compared to HFD controls (Fig. 2s, t). GLUT4 protein levels (Fig. 2y)
were not modulated by PNe in the iWAT (−9%, $p = 0.33$) but trended to
increase in the iBAT (+20%, $p = 0.090$) (Fig. 2s, u). Lastly, subcellular
fractionation of the cell lysate from iBAs (Supplementary Fig. 3n, o, q)
and primary human subcutaneous adipocytes (Supplementary Fig. 3r, s)
affirmed elevated GLUT1 membrane localization after PNe treatment.
However, we also noted higher GLUT4 levels in the PM fraction from
iBAs (Supplementary Fig. 3p). Collectively, our data from murine and
human adipocyte models indicate that PNe simulates insulin-

independent glucose uptake and concomitantly enhances glycolysis,
possibly by increasing total and surface GLUT1 protein levels via upre-
gulated *Glut1* mRNA expression. This effect may be potentiated by a
shift of GLUT4 localization towards the PM in the basal state.

## Short-term PNe treatment increases adipocyte glucose uptake by activating the PKC-ERK1/2-*Glut1* axis

We hypothesized that PNe might function via Protein Kinase C (PKC)
signaling, as it was shown that activated PKC upregulates *Glut1* mRNA
and GLUT1 protein levels, leading to higher glucose uptake rates[14,22,23].
The PKC agonist PMA (phorbol 12-myristate 13-acetate) was used as
positive control. We could show that PNe increased the phosphoryla-
tion of various PKC substrates (Fig. 3a) while levels of phospho-PKC
substrates were decreased when cells were pre-exposed to the PKC
inhibitor Gö−6983 for 30 min (Supplementary Fig. 4a). ERK1/2 is a
downstream target of PKC, which translocates to the nucleus after
phosphorylation and initiates *Glut1* gene transcription[14,23]. We pre-
dicted that the increase in GLUT1 protein content upon chronic PNe
treatment is mediated through the sequential activation of ERK1/
2 signaling by PKC and upregulation of *Glut1* expression. PNe triggered
ERK1/2 phosphorylation in murine iBAs (Fig. 3b, c) and beige hMADS
(Supplementary Fig. 4b, c). Moreover, 4 h of PNe treatment sig-
nificantly increased *Glut1* mRNA expression in iBAs, which was abro-
gated when cells were pre-exposed to the ERK1/2 inhibitor PD184352
(Fig. 3d). Upregulation of *Glut1* but not *Glut4* by PNe was confirmed
after 3 h of stimulation, when ERK1/2 was in it is activated, phos-
phorylated state (Supplementary Fig. 4d, e). To substantiate these
findings, we performed immunofluorescence staining and visualized a
dominant nuclear localization of ERK1/2 in response to PNe. In con-
trast, ERK1/2 remained localized in the cytosol, when PKC was inhib-
ited by Gö−6983 (Fig. 3e, f). Images at higher magnification are
displayed in Supplementary Fig. 4f.

Lastly, we stimulated iBAs with PNe for 16 h to ensure an increase
in the GLUT1 protein pool in presence or absence of PD184352 and
found that suppression of ERK1/2 prevents PNe-stimulated glucose
uptake (Fig. 3g). Moreover, glucose uptake rates were increased after
3 h of stimulation, which was dependent on the upstream PKC activity
(Supplementary Fig. 4g).

Since *Erk1* knockout mice exhibit impaired adipogenesis and ERK
phosphorylation is induced during early adipogenic differentiation[24,25],
we examined whether PNe regulates adipocyte formation. Therefore,
we treated immortalized brown preadipocytes and 3T3-L1 fibroblasts
with PNe the day before induction and throughout adipogenic differ-
entiation. PNe treatment inhibited adipogenesis at the highest con-
centration (Supplementary Fig. 4h–j), while a stimulatory effect was
observed at lower doses in 3T3-L1 cells (Supplementary Fig. 4i).

In summary, our data is consistent with a PKC-activation potential
of PNe, which enables higher glucose uptake by initiating *Glut1* tran-
scription through ERK1/2. This mode of action presents an interesting
strategy to target hyperglycemia, since established diabetes therapies
do not target insulin-independent glucose uptake via GLUT1.

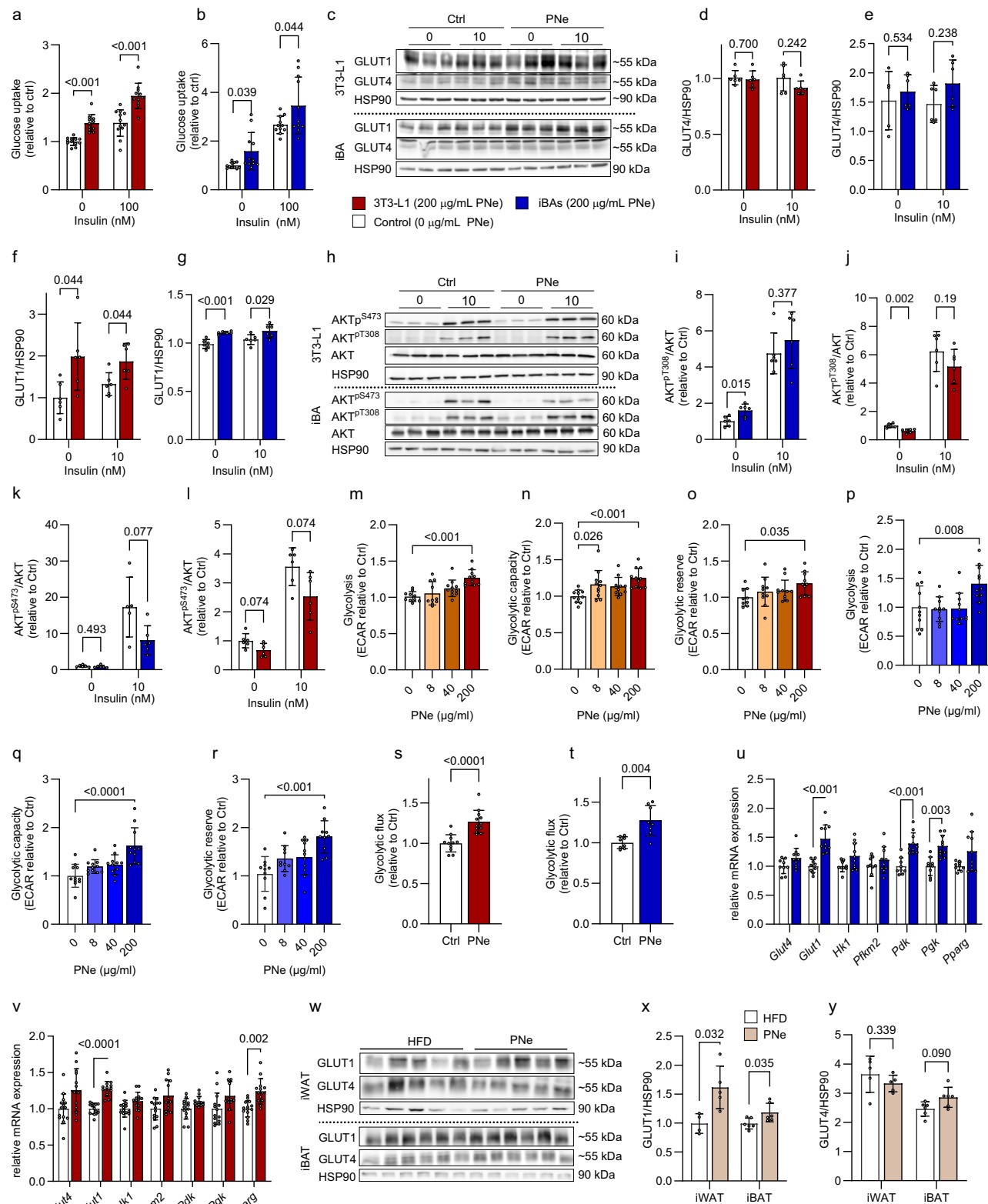

## A bioactivity-guided fractionation approach reveals other constituents than the major compound mangiferin as functionally relevant

We subsequently focused on identifying the active principle which governs the function of PNe. An initial analysis of the phytochemical composition of PNe using ultra high-performance liquid chromatography with corona-charged aerosol detection (UHPLC-CAD) and high-resolution mass-spectrometry (UHPLC-HRMS/MS) identified the xanthone mangiferin (MG) as the main constituent present in PNe (Supplementary Fig. 5a). The MG level measured in PNe was $14.5 \pm 0.4\%$ ($n = 4$), which is consistent with earlier chemical profiles of PN[19] and Apontes et al. have previously described in detail the anti-hyperglycemic activity of MG[26].

To explore potentially active compounds in the PNe, we fractionated the extract in four fractions (F1–F4) (Supplementary Figs. 5 and 6a). F1 contained polar compounds (mainly saccharides)

**Fig. 2 | PNe increases adipocyte glucose uptake via insulin independent GLUT1 in vitro.** Cells were treated daily with PNe (0, 200 µg/mL) for 3 days. **a** Glucose uptake in 3T3-L adipocytes. 0 nM insulin: Ctrl $n = 12$, PNe $n = 11$. 10 nM insulin (60 min): Ctrl $n = 12$, PNe $n = 11$. **b** Glucose uptake in iBAs. 0 nM insulin: Ctrl $n = 11$, PNe $n = 11$. 10 nM insulin (60 min): Ctrl $n = 11$, PNe $n = 11$. **c** Western blots for GLUT1 and GLUT4 in adipocyte cell lines. **d, e** GLUT4 quantification for **d** 3T3-L1 adipocytes (0 nM, 10 nM insulin: Ctrl $n = 6$, PNe $n = 6$) and **e** iBAs (0 nM insulin: Ctrl $n = 5$, PNe $n = 6$. 10 nM insulin: Ctrl $n = 6$, PNe $n = 6$). **f, g** GLUT1 quantification for **f** 3T3-L1 adipocytes. (0 nM, 10 nM insulin: Ctrl $n = 6$, PNe $n = 6$) and **g** iBAs (0 nM, 10 nM insulin: Ctrl $n = 6$, PNe $n = 6$). **h** AKT$^{pT308}$ and AKT$^{pS473}$ Western blots with or without insulin stimulation (20 min, 10 nM) in adipocyte cell lines. **i, j** AKT$^{pT308}$ quantification for **i** iBAs (Ctrl $n = 6$, PNe $n = 6$ for 0 nM and 10 nM insulin) and **j** for 3T3-L1 adipocytes (Ctrl $n = 6$, PNe $n = 6$ for 0 nM and 10 nM insulin). **k, l** AKT$^{pS473}$ quantification for **k** iBAs (0 nM insulin: Ctrl $n = 6$, PNe $n = 6$. 10 nM insulin: Ctrl $n = 5$, PNe $n = 6$) and **l** 3T3-L1 adipocytes (Ctrl $n = 6$, PNe $n = 6$ for 0 nM and 10 nM insulin). **m, n** Glycolytic stress test in 3T3-L1 adipocytes. **m** Glycolysis (0 µg/mL $n = 10$, 8 µg/mL $n = 10$, 40 µg/mL $n = 11$, 200 µg/mL $n = 10$), **n** Glycolytic capacity (0 µg/mL $n = 11$, 8 µg/mL $n = 11$, 40 µg/mL $n = 11$, 200 µg/mL $n = 11$) and **o** Glycolytic reserve (0 µg/mL $n = 9$, 8 µg/mL $n = 10$, 40 µg/mL $n = 11$, 200 µg/mL $n = 10$). **p–r** Glycolytic stress test in iBAs. **p** Glycolysis (0 µg/mL $n = 11$, 8 µg/mL $n = 10$, 40 µg/mL $n = 11$, 200 µg/mL $n = 11$), **q** Glycolytic capacity (0 µg/mL $n = 11$, 8 µg/mL $n = 11$, 40 µg/mL $n = 11$, 200 µg/mL $n = 11$) and **r** Glycolytic reserve (0 µg/mL $n = 10$, 8 µg/mL $n = 10$, 40 µg/mL $n = 11$, 200 µg/mL $n = 10$). **s, t** $^3$H-Glycolytic flux analysis of (s) 3T3-L1 (Ctrl $n = 11$, PNe $n = 12$) and **t** iBAs (Ctrl $n = 8$, PNe $n = 8$). **u, v** mRNA expression of targets regulating glucose metabolism. **u** 3T3-L1 (Ctrl $n = 14$, PNe $n = 13$). **v** iBAs (Ctrl $n = 9$, PNe $n = 9$). **w** Western blots of **x** GLUT1 in iWAT (HFD $n = 4$, PNe $n = 5$) and iBAT (HFD $n = 6$, PNe $n = 6$). **y** GLUT4 in iWAT (HFD $n = 5$, PNe $n = 5$) and iBAT (HFD $n = 6$, PNe $n = 6$) from mice. DI = 12 weeks. iBAs blue graphs, 3T3-L1 adipocytes red graphs. Results are reported as mean ± SD. Data points are pooled from two independent experiments, expect for (**x, y**). Multiple two-tailed *t*-tests with Holm–Sidak's multiple-comparison test were applied in (**a, b, d–g, i–l, x, y**). Two-tailed student's *t*-test for comparisons between two groups was applied in (**s, t**). Two-tailed student's *t*-test with Sidak's multiple comparison adjustment was applied in (**u, v**). One-Way ANOVA with Dunnett's post-hoc test was applied in (**m–r**) to compare PNe against 0 µg/mL. Statistical test results are indicated as exact *p*-values with *$p < 0.05$ considered significant. Source data are provided in Source Data 1. PNe *Phaleria nisidai* extract, iBAs immortalized brown adipocytes, ECAR Extracellular acidification rate, iWAT inguinal white adipose tissue, eWAT epididymal white adipose tissue.

(Supplementary Figs. 5b and 6a), F3 consisted of MG and iriflophenone-2-O-β-glucoside (Supplementary Figs. 5d and 6a, b), while F2 and F4 corresponded to fractions devoid of MG (Supplementary Figs. 5c, e and 6a, b). Fractions F2, F3, and F4 were tested in vivo concomitantly with PNe after an initial HFD-feeding regime of 6 weeks (Fig. 4a) with doses corresponding to the amounts found in the total PNe (see "Method" section: "Fraction preparation and dosage for in vivo testing"). As chronic intake of complex plant preparations can cause herbal-induced-liver injury, we quantified plasma alanine transaminase (ALT) levels in response to the fractions and PNe[27,28]. After 14 weeks of exposure, none of the applied fractions or the extract induced higher plasma ALT activity (Supplementary Fig. 7a). Furthermore, no differences in food intake patterns (Supplementary Fig. 7b), body weight development (Supplementary Fig. 7c, d) or composition (Supplementary Fig. 7e) were observed between groups, comparable to PNe (Fig. 1b). Over time, F4 lowered random fed blood glucose (RBG) concentrations during the DI (Fig. 4b). Counterintuitively, RBG levels did not significantly increase during the study, which has been reported previously[29,30]. No significant improvement in insulin sensitivity was induced by any fraction after 4 weeks of treatment (Fig. 4c, d), while only F4 effectively ameliorated insulin sensitivity after prolonged DI of 10 weeks (Fig. 4e, f). F2, F3, and F4 enhanced blood glucose clearance after an oral glucose load (Fig. 4g, h). Interestingly, only F3-fed mice displayed higher circulating active GLP-1 concentrations 2 min after an oral glucose bolus (Fig. 4i). In agreement with the described benefits of PNe fractions on glucose homeostasis, terminal fasting blood glucose was reduced for all fractions (Fig. 4j). Collectively, F2, F3, and F4 modulate glucose homeostasis in DIO mice by improving different metabolic readouts as summarized in Supplementary Table 1, however prolonged insulin sensitivity is conferred by F4.

To assess the effectiveness of F4 versus a known intervention, we compared the efficacy of F4 treatment against the diabetes drug metformin (MET) at a dose of 0.25% w/w[31] (Supplementary Fig. 8a). We additionally increased the initial HFD-feeding period to 12 weeks to further probe the efficacy of F4 in a model of more severe insulin resistance. F4 and MET did not affect RBG (Supplementary Fig. 8b), when measured repeatedly over the course of the DI. F4 or MET did not inhibit intestinal alpha-glucosidase activity, as the magnitude and rate of appearance of blood glucose spikes did not differ between groups in an oral sucrose tolerance test (Supplementary Fig. 8c, d). Similarly, insulin sensitivity was not significantly improved by F4 or MET after 9 weeks of DI (Supplementary Fig. 8e, f), suggesting that higher doses maybe required in more severe diabetic models. To quantify potential improvement in insulin sensitivity caused by F4 intake after 6 weeks of initial HFD and to decipher the responsible tissues more precisely, we performed hyperinsulinemic-euglycemic clamps (Fig. 4k) after 7 to 8 weeks of DI. Compared to control fed animals (36.3 ± 5.7 mg/kg × min), F4 (43.0 ± 6.0 mg/kg × min) and MET (47 ± 6.4 mg/kg × min) treated mice had a significantly increased glucose infusion rate (GIR, Fig. 4l). Glucose infusion rate over time is presented as mean for each treatment group (Supplementary Fig. 8g) and for each mouse within a group (Supplementary Fig. 8h–j). Blood glucose concentrations during the procedure are in Supplementary Fig. 8k. Basal whole-body glucose turnover (Fig. 4m) was not different between groups and tended to increase in F4 treated mice ($p = 0.151$) during the insulin-stimulated condition (Fig. 4n). Endogenous hepatic glucose output was consistent between all groups, but robustly repressed after insulin infusion (Fig. 4o and Supplementary Fig. 8l). A single bolus of $^{14}$C-2-deoxyglucose was infused into the jugular vein in steady state conditions and tissue-specific glucose uptake rates assessed. iWAT, iBAT, and soleus muscle displayed higher glucose influx for MET and F4 treatments (Fig. 4r–t). F4 enhanced glucose uptake also into gastrocnemius (Fig. 4u). Glucose uptake rates into the eWAT (Fig. 4q) was unaffected by either treatment. Because F4-treatment increased skeletal muscle glucose uptake, we tested PNe and F4 on C2C12 myotubes. Neither the whole extract nor F4 enforced glucose uptake or stimulated PM localization of GLUT1 or GLUT4 (Supplementary Fig. 9m–p). We hypothesize that F4 constituents underwent chemical modifications after ingestion, a metabolic step involving structural modifications of constituents and referred to as biotransformation[32]. These metabolites resulting from biotransformation would subsequently be responsible for bioactivity.

Overall, we could show that dietary supplementation with F4 has a profound impact on glucose uptake kinetics in various tissues to a slightly lower magnitude than metformin.

## Chemical composition of fraction F4

We next aimed to unravel the chemical composition of F4 to specify the constituents, which mediate the improvement in insulin sensitivity. F4 was further fractionated to formally identify its constituents. Chromatographic separation of F4 was optimized to purify seven compounds, which were unambiguously characterized by NMR analyses (see method section: "Description of the isolated compounds", and Supplementary Figs. 5 and 6). Two acetylmangiferins xanthones were identified; 6'-*O*-acetylmangiferin[33] and 2'-*O*-acetylmangiferin, which was never reported to our knowledge. In addition, two mangiferin glucofuranosyl-derivatives, 2-*C*-β-glucofuranosylmangiferin and

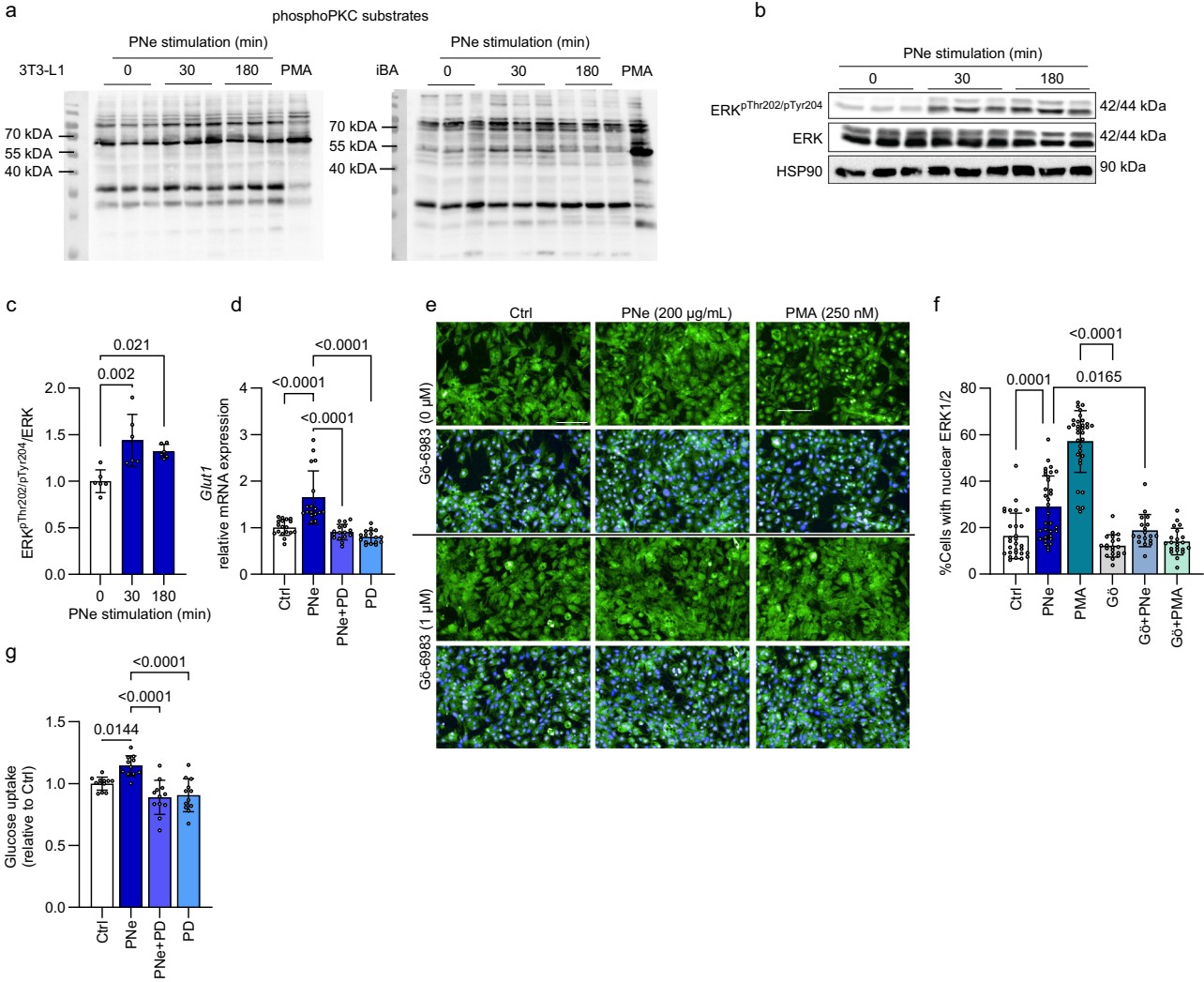

**Fig. 3 | PNe activates the PKC-ERK1/2 axis in brown adipocytes. a** PhosphoPKC substrates western blot. PMA (250 nM) as positive control. **b**, **c** ERK1/2 phosphorylation (Thr204, Tyr202) induced by PNe in **b** iBAs. **c** Quantification of ERK$^{pThr202/pTyr204}$. PNe $n = 6$, Ctrl $n = 6$ per time point. 2 independent experiments. **d** Glut1 mRNA levels in iBAs stimulated for 4 h with PNe ± pre-treatment with ERK1/2 inhibitor PD184352 (500 nM, 30 min). Ctrl $n = 18$, PNe $n = 17$, PNe+PD $n = 16$, PD $n = 16$. 3 independent experiments. **e** Representative immunofluorescence pictures stained for nuclei (blue) and ERK1/2 (green) in iBAs. Scale bar 100 μM. **f** Quantification of ERK1/2 localization. 180 min of PNe/PMA treatment ± PKC inhibitor Gö−6983 (1 μM). Ctrl $n = 29$, PNe $n = 33$, PMA $n = 32$, Gö $n = 20$, Gö + PNe $n = 18$, Gö + PMA $n = 21$ images/condition.

3 independent experiments. Average 328 cells/image. **g** Glucose uptake rates in iBAs after 16 h of PNe ± pre-treatment with ERK1/2 inhibitor PD184352 (500 nM, 30 min). Ctrl $n = 12$, PNe $n = 11$, PNe + PD $n = 12$, PD $n = 12$. 3 independent experiments. Results are reported as mean ± SD. One-way ANOVA with Tukey's post-hoc test between all groups was applied in (**c**, **d**, **f**, and **g**). Only biologically relevant comparisons are displayed. All comparisons are listed in Source Data 1. Statistical test results are indicated as exact $p$-values with *$p < 0.05$ considered significant. Source data are provided in Source Data 1. PMA phorbol 12-myristate 13-acetate, PNe Phaleria nisidai extract, PD PD184352.

2-C-α-glucofuranosyl-mangiferin are newly described. The presence of the flavone genkwanin 5-O-β-primeveroside (synonym of yuankanin), already described for PN[34], was confirmed. The C-glucoside isovitexin[35], and genkwanin 5-O-β-glucoside[36] have not been previously reported in PN.

Since the MG-containing F3 was not as effective as F4 with respect to insulin sensitivity, we limited our in vivo test to the flavones isovitexin (IX), genkwanin 5-O-β-glucoside (GG), and genkwanin 5-O-β-primeveroside (GP) to identify the active phytochemicals. We additionally tested genkwanin (GE), the common aglycone of genkwanin 5-O-β-primeveroside and genkwanin 5-O-β-glucoside as well as apigenin (AP), the aglycone of isovitexin (Fig. 5a). Indeed, glycosylated flavonoids are known to be cleaved by intestinal metabolism after ingestion[37]. The tested concentrations were calculated from the estimated proportions of the individual compounds in PN, assuming that glycosides are converted to their aglycone at a 1:1 ratio

(Fig. 5b). After 7–8 weeks of DI, GE ($p\_unadj. = 0.048$) and GG ($p\_unadj. = 0.048$) treatment tended to improve insulin resistance (Fig. 5c, d), whereas both GE and GG robustly ameliorated glucose tolerance (Fig. 5e, f). Similarly, RBG was reduced by the end of the intervention (Fig. 5g), while the other tested compounds had no detectable impact on glucose homeostasis (Supplementary Fig. 9a–d). Furthermore, GE-treated mice displayed increased urinary glucose concentrations, which indicates higher renal glucose excretion (Fig. 5h). Indeed, phloretin is a predicted metabolite of GE and structurally related to phlorizin (Supplementary Fig. 12a), the precursor compound of the current SGLT2-inhibitor drug class[38,39]. In human, a lower visceral to subcutaneous fat ratio is metabolically favorable and correlates with insulin sensitivity[40]. While amounts of iWAT and eWAT did not differ between treated and HFD-fed mice (Supplementary Fig. 9e, f), the eWAT/iWAT-ratio was lower in the GG group (Fig. 5i). This finding implies a shift in fat storage towards the

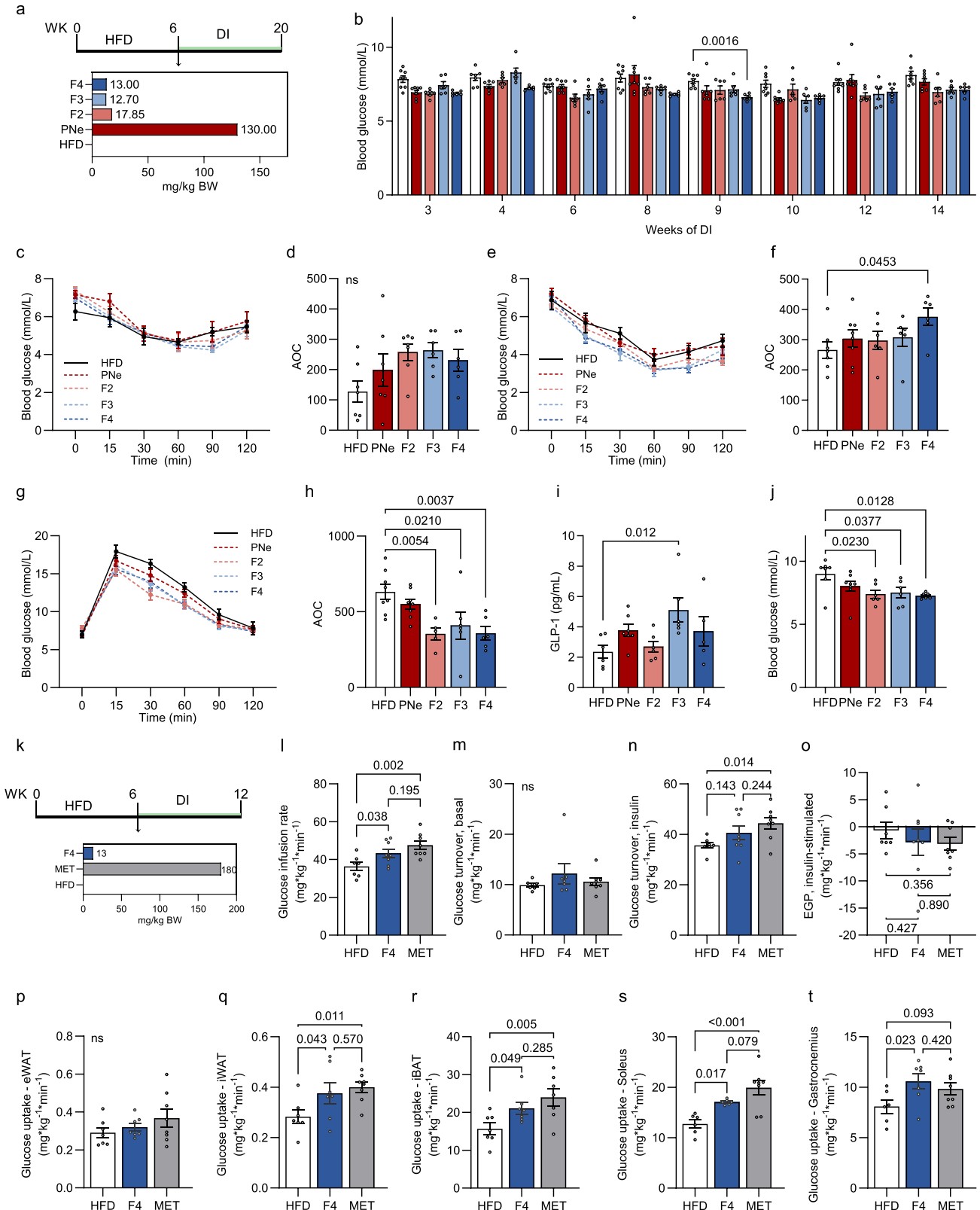

subcutaneous depot, which might lead to a healthier adipose tissue expansion in response to chronic overnutrition and contribute to improved glucose homeostasis. Circulating blood lipid and insulin concentrations remained unchanged in the fasted state after four weeks of DI (Supplementary Fig. 9g–j). At the end of the intervention, random fed non-esterified fatty acids, cholesterol (Supplementary Fig. 9k, l) and insulin levels (Supplementary Fig. 9n) were likewise

indifferent. However, triglyceride concentrations were reduced for GG and IX-fed mice (Supplementary Fig. 9m). Comparing the effects of F4 and individual F4 constituents based on the AOC from the GTT, we measured a 19.7% decrease in the AOC from F4 vs HFD animals, a 12.7% decrease for GE vs HFD and a 13.9% for GG vs HFD. Taken together, our data suggests that GE glycosides confer some of the glucose-sensitizing effects of F4, whereas the other tested

**Fig. 4 | Bioactivity-guided fractionation of PNe reveals F4 as source of bioactive substances. a** Experimental design for (**b–j**). **b** Random fed blood glucose measured at 11 am. HFD $n = 8$, PNe $n = 8$, F2 $n = 6$, F3 $n = 6$, F4 $n = 6$. **c** ITT and **d** AOC. DI = 4 weeks. HFD $n = 7$, PNe $n = 7$, F2 $n = 7$, F3 $n = 6$, F4 $n = 6$. **e** ITT and **f** AOC. DI = 10 weeks. HFD $n = 7$, PNe $n = 8$, F2 $n = 6$, F3 $n = 6$, F4 $n = 6$. **g** oGTT (2 g/kg BW) and **h** AOC. DI = 12 weeks. HFD $n = 8$, PNe $n = 8$, F2 $n = 5$, F3 $n = 6$, F4 $n = 6$. **i** Circulating active GLP-1 levels 2 min after an oral glucose load. HFD $n = 6$, PNe $n = 7$, F2 $n = 6$, F3 $n = 6$, F4 $n = 5$. **j** Terminal fasting blood glucose concentrations. HFD $n = 6$, PNe $n = 8$, F2 $n = 6$, F3 $n = 6$, F4 $n = 6$. **k** Experimental design for (**l–t**). **l** Glucose infusion rate (GIR). HFD $n = 7$, F4 $n = 8$, MET $n = 8$. **m, n** Whole-body glucose turnover in **m** basal (HFD $n = 7$, F4 $n = 7$, MET $n = 8$) and **n** insulin-stimulated (HFD $n = 7$, F4 $n = 8$, MET $n = 8$) states. **o** Endogenous glucose production (EGP) in insulin-stimulated state (HFD $n = 7$, F4 $n = 8$, MET $n = 8$). **p–t** $^{14}$C-2-deoxyglucose uptake rates into **p** eWAT (HFD $n = 7$, F4 $n = 7$, MET $n = 8$), **q** iWAT (HFD $n = 7$, F4

$n = 7$, MET $n = 8$), **r** iBAT (HFD $n = 7$, F4 $n = 7$, MET $n = 7$), **s** soleus (HFD $n = 6$, F4 $n = 6$, MET $n = 8$) and **t** gastrocnemius (HFD $n = 6$, F4 $n = 7$, MET $n = 8$) in clamped condition. Results are reported as mean ± SEM. Mixed-effects analysis with Diet × Time interaction and Sidak's multiple comparison was used in (**b**) to compare each treatment to HFD at each time point. Comparisons were grouped into one family to control FWER, $\alpha = 0.05$. One-way ANOVA with Dunnet's post-hoc test compared to HFD was applied in (**d, f, h–j**). One-way ANOVA with uncorrected Fisher's LSD post-hoc test between all groups was applied in (**l–t**). Statistical test results are indicated as exact $p$-values with *$p < 0.05$ considered significant. Source data are provided in Source Data 1. AOC area of the curve, EGP endogenous glucose production, iWAT inguinal white adipose tissue, eWAT epididymal white adipose tissue, iBAT interscapular brown adipose tissue, MET Metformin, ITT insulin tolerance test, oGTT oral glucose tolerance test, FWER family-wise error rate, DI dietary intervention.

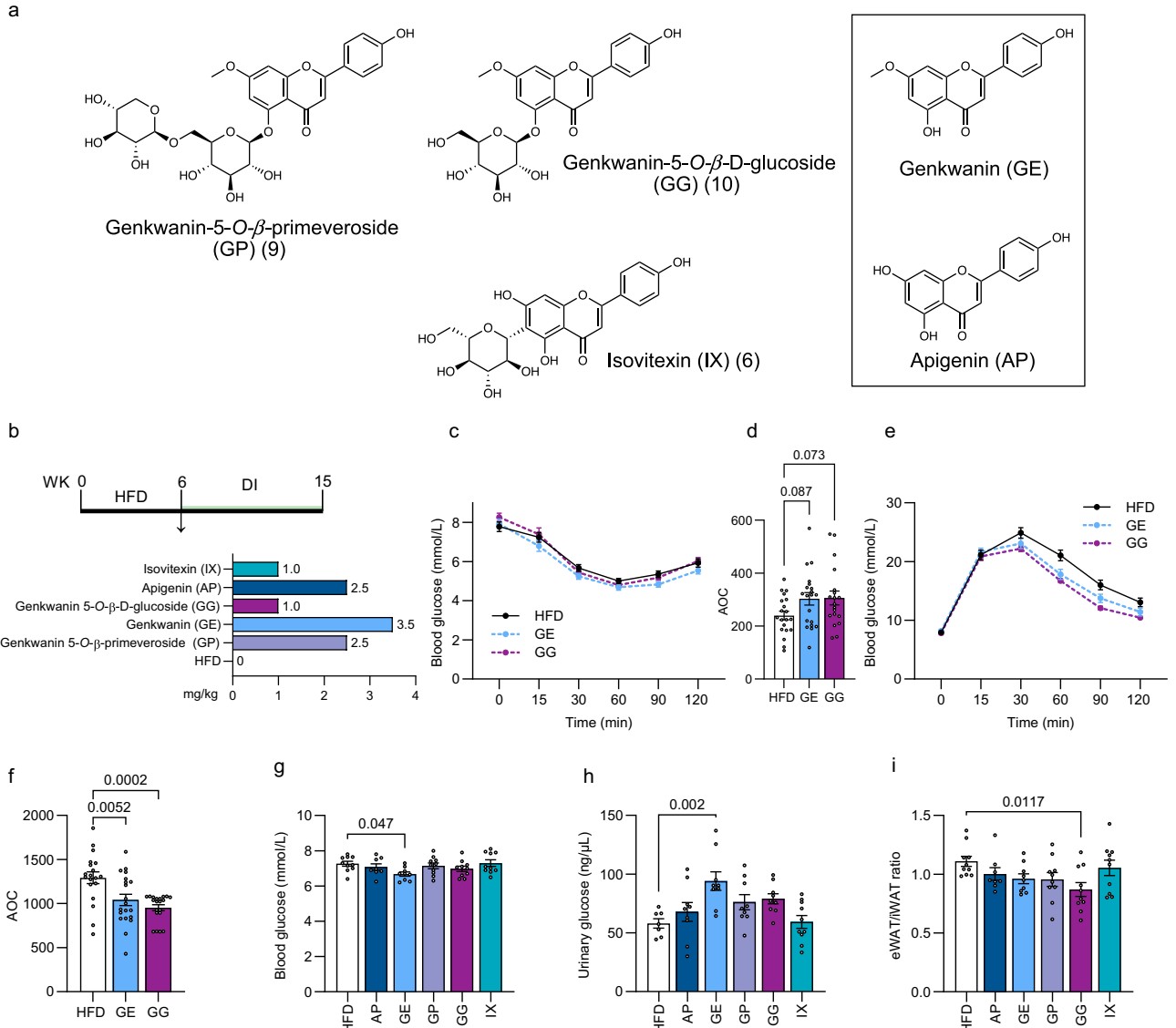

**Fig. 5 | GE glycosides mediate the beneficial effects of F4. a** Structures of flavones tested in vivo. GP, GG, and IX were isolated in F4. GE and AP were selected as potential metabolites biotransformed in the digestive system after ingestion of the F4-constituents. **b** Experimental design and applied doses for (**c–i**). **c** Insulin tolerance tes and **d** AOC. DI = 7 weeks. HFD $n = 20$, GG $n = 19$, GE $n = 20$. **e** Glucose tolerance test and **f** AOC. DI = 8 weeks. HFD $n = 19$, GG $n = 20$, GE $n = 20$. **g** Random fed blood glucose concentrations after DI = 9 weeks. HFD $n = 10$, AP $n = 8$, GE $n = 9$, GP $n = 10$, GG $n = 10$, IX $n = 10$. **h** Fasting urinary glucose concentrations. DI = 4 weeks. HFD $n = 7$, AP $n = 9$, GE $n = 9$, GP $n = 9$, GG $n = 9$, IX $n = 10$. **i** Ratio of epididymal (eWAT) to

inguinal (iWAT) adipose tissue. HFD $n = 10$, AP $n = 8$, GE $n = 9$, GP $n = 10$, GG $n = 10$, IX $n = 10$. Results are reported as mean ± SEM. One-way ANOVA with Dunnett´s post-hoc test against HFD control was applied in (**d, f–h**, and **i**). One-way ANOVA with uncorrected Fisher's LSD post-hoc test for (**d, f**) is displayed in Source Data 1. Statistical test results are indicated as exact $p$-values with *$p < 0.05$ considered significant. Source data are provided in Source Data 1 and Source Data 2. iWAT, inguinal white adipose tissue; eWAT, epididymal white adipose tissue.

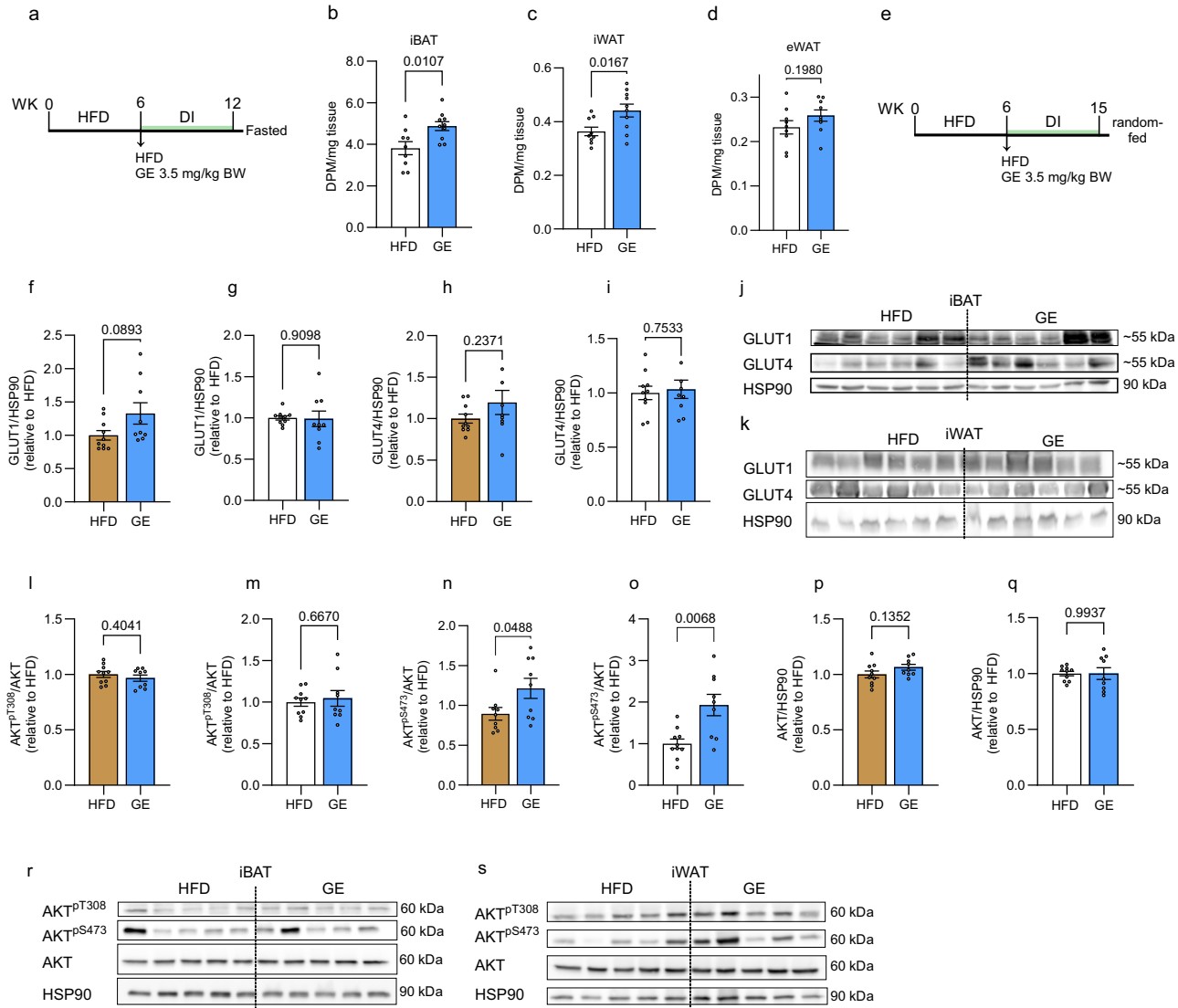

**Fig. 6 | GE supports insulin action by promoting basal glucose uptake.**
**a** Experimental design for (**b**-**d**) and ¹⁴C-2-deoxyglucose uptake into **b** iBAT (HFD *n* = 9, GE *n* = 10), **c** iWAT (HFD *n* = 9, GE *n* = 10), and **d** eWAT (HFD *n* = 9, GE *n* = 9). DI = 6 weeks. **e** Experimental design for Western blots depicted in (**f**–**q**). **f**, **g** Effect of GE on GLUT1 protein in **f** iBAT and **g** iWAT. HFD *n* = 10, GE *n* = 9. **h**, **i** Effect of GE on GLUT4 protein in **h** iBAT and **i** iWAT. HFD *n* = 10, GE *n* = 9. **j**, **k** GLUT1 and GLUT4 Western blots quantified from **j** BAT and **k** iWAT of random fed mice. DI = 9 weeks. HFD *n* = 10, GE *n* = 9. **l**, **m** Effect of GE on AKT phosphorylation at T308 in **l** iBAT and **m** iWAT. HFD *n* = 10, GE *n* = 9. **n**, **o** Effect of GE on AKT phosphorylation at S473 in

**n** iBAT (HFD *n* = 10, GE *n* = 9) and **o** iWAT (HFD *n* = 9, GE *n* = 9). **p**, **q** Effect of GE on total AKT in **p** iBAT and **q** iWAT. HFD *n* = 10, GE *n* = 9. **r**, **s** Western blots quantified from **r** iBAT and **s** iWAT of random fed animals, DI = 9 weeks. *n* = 9–10 per group. Results are reported as mean ± SEM. Two-tailed student's test was applied in all comparisons. Statistical test results are indicated as exact *p*-values with *\*p* < 0.05 considered significant. Source data are provided in Source Data 1. iWAT inguinal white adipose tissue, eWAT epididymal white adipose tissue, iBAT interscapular brown adipose tissue, GE genkwanin.

compounds did not exert any relevant bioactivity as summarized in Supplementary Table 1.

### Genkwanin promotes insulin sensitivity by increasing GLUT1 protein and AKT phosphorylation

The crude PNe promoted blood glucose clearance into adipose depots possibly by upregulating the protein levels of the insulin-independent glucose transporter GLUT1 (Figs. 1s, t and 2w–x). Therefore, we determined if GE induced the same molecular pathways. We could confirm a stimulatory effect of GE on glucose uptake into iBAT and iWAT in the fasted state without co-administration of insulin after 6 weeks of treatment (Fig. 6a–c). Glucose disposal into eWAT was unchanged (Fig. 6d). In an independent experiment, we extended the DI to 9 weeks and collected tissues in the random fed state to evaluate GLUT levels and AKT phosphorylation states (Fig. 6e). In line with the

enforced adipose glucose disposal in the fasted state, mice supplemented with GE showed a trend towards increased GLUT1 protein levels in the iBAT (*p* = 0.089) but not iWAT (Fig. 6f, g). Western blots for all targets are shown in Fig. 6j, k and Supplementary Fig. 10a, b. This depot-specific GLUT1-regulation implies that GE acts predominantly on metabolically active brown adipocytes, which exist in form of beige adipocytes in the WAT depots of mice housed at room temperature[41]. GLUT4 levels were unaltered in both adipose types (Fig. 6h, i). We speculated that a mild but persistent increase in baseline glucose clearance rates might cumulatively lead to a glucose-lowering response, which reduces systemic glucotoxicity and assists in preserving insulin responsiveness. This is suggested by increased AKT phosphorylation at S473 (Fig. 6n, o) in the random fed state in the iWAT and iBAT of GE-fed mice (Fig. 6r, s). AKT phosphorylation at T308 (Fig. 6l, m) and total AKT levels were unchanged (Fig. 6p, q).

GLUT protein, total AKT and AKT T308 phosphorylation were unchanged in soleus and gastrocnemius muscles (Supplementary Fig. 10c–p). AKT S473 phosphorylation was reduced in gastrocnemius but not soleus muscle (Supplementary Fig. 10k, l). These Western blot results are further summarized in Supplementary Table 1. As one experimentally supported mechanism, we propose that GE operates as glucose sensitizer of metabolically active adipocytes, which secondarily strengthens global insulin sensitivity. To corroborate this mechanistic model, we treated primary brown adipocytes with GE for 3 days. We did not observe any stimulatory effects on glucose uptake rates, GLUT1 and GLUT4 protein levels or mRNA expression (Supplementary Fig. 11a–e). GE stimulation was further without effects on iBAs or 3T3-L1 adipocytes (Supplementary Fig. 11f–i). Due to the oral application of GE to mice, we last contemplated that active biotransformation products may be generated from GE in vivo. Combining literature research with the in silico prediction software Biotransformer 3.0, we identified seven putative GE-derived metabolites for functional assessment in immortalized adipocyte lines (Supplementary Fig. 12a). We first evaluated the effects of apigenin as a primary metabolization product, which reduced glucose uptake rates in iBAs and 3T3-L1 adipocytes at commonly applied concentrations (Supplementary Fig. 12b, c)[42,43]. Similarly, none of the other substances promoted glucose uptake rates into iBAs (Supplementary Fig. 12d–i), implying that the translation of our in vitro to in vivo findings is limited.

**Genkwanin ameliorates systemic glucose homeostasis comparably to metformin**

Lastly, we evaluated how GE compares against metformin in the hyperinsulinemic-euglycemic clamp to support its effectiveness (Fig. 7a). It needs to be accentuated that the selected GE dosage (3.5 mg/kg BW) originates from nutrition-based intake levels as opposed to metformin supplementation (0.25%), which relates to a pharmacologically relevant human dose. Compared to HFD-controls (27.89 ± 10.21 mg/min × kg BW), MET (39.8 ± 6.7 mg/min × kg BW) and GE (39.3 ± 12.13 mg/min × kg BW) supplementation increased glucose infusion rates (Fig. 7b, c) required to stabilize blood glucose at 6 mM (Fig. 7d). Consistently, GE and MET elevated insulin-stimulated glucose disappearance rate (Fig. 7e) and whole-body glucose turnover (Fig. 7f) to a comparable extent. Basal glucose turnover (Fig. 7g), equaling to basal hepatic glucose production (Fig. 7h), was unchanged between treatment groups. Not only was insulin-induced glucose clearance promoted by GE, but also hepatic insulin sensitivity improved as reflected in stronger suppression of endogenous glucose production (Fig. 7i). $^{14}$C-deoxyglucose tracing during the last 30 min of the clamps revealed higher glucose disposal rates into the eWAT (Fig. 7j) and iWAT (Fig. 7k), for both treatments and a trend for soleus muscle (Fig. 7m) for GE. No differences were measured in the iBAT or gastrocnemius (Fig. 7l, n). Together, these data summarized in Supplementary Table 1 emphasize the therapeutic potential of orally applied GE to improve systemic insulin resistance and potentially restore normal glycaemia.

## Discussion

In Palau, a *Phaleria nisidai* (PN) leaf decoction has been consumed for decades[34], among other indications, as an herbal antidiabetic agent without formally documented clinical and mechanistic evidence for its efficacy. The plant was selected based on a retrospective treatment outcome survey[18], which aimed at evaluating the traditionally used remedies with the best reported outcomes for metabolic and non-communicable diseases. A randomized clinical trial confirmed the efficacy of the PN decoction in humans[19] and our investigation aimed at identifying the active principles and determining their mechanisms of action.

To elucidate possible mechanistic activity, we report here the glucoregulatory effect of PNe in DIO mice and delineated genkwanin glycosides as some of the mediators triggering the beneficial action of

PNe in an interdisciplinary study. Although insulin-stimulated glucose uptake into adipose tissue is 4–5 times lower compared to muscle[44], the adipose organ is a fundamental regulator of whole-body glucose homeostasis as can be seen in adipocyte-specific *Glut4* knockout animals[45], which develop insulin resistance in the liver and skeletal muscle, causing glucose intolerance. Prolonged HFD supplementation with PNe elevated insulin sensitivity and based on our in vitro data from brown and white adipocytes, we propose that PNe exerts part of its advantageous effect on glucose metabolism by stimulating PKC-ERK1/2 signaling. A chronic PNe challenge led to elevated GLUT1 protein content in brown and white adipocytes but also in the corresponding murine fat depots, which emerged from the parallel upregulation of *Glut1* mRNA as observed in vitro. This transcriptional regulation of *Glut1* was suppressed when cells were pretreated with an ERK1/2 inhibitor, a known activator of *Glut1* transcription. Incubation of adipocytes with the PKC-agonist PMA initiates *Glut1* expression through activated ERK1/2[14]. Similarly, FGF21 stimulates *Glut1* expression in murine eWAT via the sequential activation of ERK1/2 and the downstream transcription factors SRF/ELK1[46]. Consequently, blocking of ERK1/2 prevented the PNe-mediated increase in adipocyte glucose uptake and supports that an increase in the total GLUT1 pool is required for PNe's effects. Subcellular fractionation analysis verified that PNe promotes both GLUT1 and GLUT4 plasma membrane levels in immortalized brown adipocytes. We have not deciphered the PKC subtype(s) induced by PNe, but certain types of PKCs are known to convey positive regulatory signals on glucose uptake routes in various cell models[47]. Overexpression of *Prkcz* in 3T3-L1 adipocytes enhances GLUT4 plasma membrane translocation without stimulating AKT phosphorylation[48]. Similarly, *Prkcz* overexpression in *Glut4-myc* tag expressing CHO cells remodels the actin cytoskeleton, thereby facilitating GLUT4 PM insertion[49]. PKCζ signaling contributes to contraction-induced glucose transport mediated by GLUT4 in L6 muscle cells triggered by carbachol treatment[50]. Therefore, PKC recruitment by PNe might induce long-term adaptations by upregulating GLUT1 synthesis via ERK1/2, while regulating glucose transport acutely by promoting exocytosis of GLUT4 storage vesicles.

ERK-signaling is critical during early adipogenesis by promoting the activity of key transcription factors, including PPARG and CEBPA[51]. ERK1 knockout mice have reduced adiposity and isolated pre-adipocytes from these mice exhibit impaired adipogenesis[25]. Conversely, ERK phosphorylation at later stages of adipogenic differentiation hampers adipocyte maturation[24]. This aligns with our observation that chronic, high-dose PNe treatment throughout differentiation decreases adipocyte formation and identifies PNe as robust ERK-activator. In accordance with upregulated *Glut1* mRNA expression, we suspect that the mature adipocytes may undergo a stress response following prolonged ERK stimulation by PNe (3 days treatment), cumulating in increased basal glucose uptake and glycolytic flux. Persistent ERK1/2 activation induces insulin resistance in 3T3-L1 adipocytes[52] and disrupts adipocyte function, which corroborate our findings of suppressed AKT S473 phosphorylation in vitro and the lacking additive effect of PNe-induced basal and insulin-stimulated glucose uptake rates. This contrasts with our DI studies, where PNe supplementation enhances insulin sensitivity and was associated with higher GLUT1 protein content in iWAT and iBAT. Such discrepancies underscore the need for caution when translating mechanistic insights from in vitro assays to in vivo context.

Leveraging insulin-independent glucose clearance is an attractive strategy as it bypasses dysfunctional insulin receptor signaling. Known mechanisms such as AMPK activation[53] and β2-adrenergic receptor stimulation in skeletal muscle[54] promote glucose clearance downstream of the insulin receptor via GLUT4, thereby converging on the same GLUT as insulin-stimulated glucose uptake. *Glut4* expression is reduced in skeletal muscle[55,56] (and adipose tissue of diabetic subjects[57]), which might affect their responsiveness to such treatments

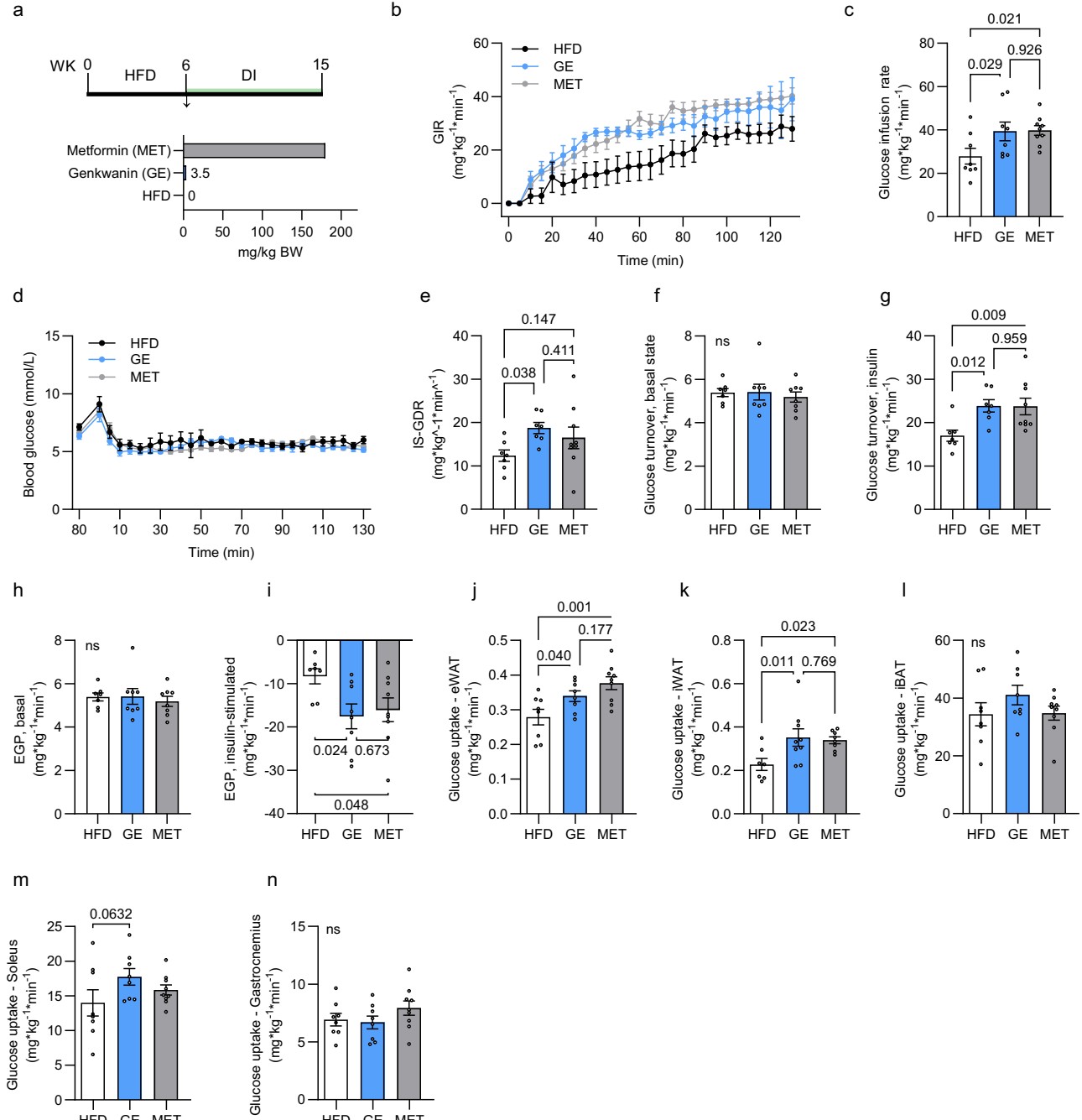

**Fig. 7 | GE competes against metformin in hyperinsulinemic-euglycemic clamps. a** Experimental design for hyperinsulinemic-euglycemic clamps. **b** Representative glucose infusion rate (GIR) over the course of the clamps until 130 min and **c** GIR during the steady state condition. HFD $n = 8$, GE, $n = 8$, Met $n = 9$. **d** Blood glucose concentration during the clamp procedure. HFD $n = 7$, GE, $n = 8$, Met $n = 9$. **e** Insulin-stimulated glucose disappearance rate (IS-GIR) at steady state condition. HFD $n = 7$, GE, $n = 7$, Met $n = 8$. **f, g** Whole body glucose turnover in **f** basal state (HFD $n = 7$, GE $n = 8$, Met $n = 8$) and in **g** insulin-stimulated conditions (HFD $n = 7$, GE $n = 8$, Met $n = 9$). **h, i** Endogenous glucose production (EGP) in **h** basal state (HFD $n = 7$, GE $n = 8$, Met $n = 8$) and in **i** insulin-stimulated conditions (HFD $n = 7$, GE

$n = 8$, Met $n = 9$). **j–n** $^{14}$C-2-deoxyglucose uptake into **j** eWAT (HFD $n = 8$, GE $n = 8$, Met n = 9), **k** iWAT (HFD $n = 7$, GE, $n = 9$ Met $n = 8$), **l** iBAT (HFD $n = 8$, GE $n = 8$, MET $n = 8$), **m** soleus (HFD $n = 8$, GE, $n = 8$, Met $n = 9$) and **n** gastrocnemius (HFD $n = 8$, GE, $n = 8$, Met $n = 9$) in clamped condition. Results are reported as mean ± SEM. One-way ANOVA with uncorrected Fisher's LSD post-hoc test was applied in (**c**, **e–n**). Comparisons were performed between all groups. Statistical test results are indicated as exact $p$-values with *$p < 0.05$ considered significant. Source data are provided in Source Data 1. iWAT inguinal white adipose tissue, eWAT epididymal white adipose tissue, iBAT interscapular brown adipose tissue.

and contribute the pathology of insulin resistance. In contrast, PNe and GE at least partially enhance basal glucose disposal by exploiting GLUT1 in adipose tissue, which could offer alternative therapeutics to control blood glucose concentrations.

Our fractionation strategy targeting the enrichment in minor constituents of PNe and depletion of mangiferin revealed that all

tested fractions exert biological activities, manifesting in distinct metabolic profiles and suggesting different underlying molecular mechanisms. Due to the chemical complexity of PNe and potential additive, synergistic, or antagonistic interactions between substances, we cannot rule out that the observed activities of the fractions are maintained in the whole extract and contribute to PNe's effects. This is

evident in the inconsistent physiological effects elicited by the fractions, which are sometimes exclusively reported for one fraction but not the PNe. Apontes et al.[26], described the protective effects of a MG-enriched HFD in C57BL6/J mice, which caused reduced weight gain, improved insulin sensitivity, and increased energy expenditure due to enhanced carbohydrate utilization in muscle. We did not observe any change in bodyweight development and the physiological response to the MG-containing F3 was less pronounced than for F4, although MG is the most abundant constituent in the PN decoction. A plausible explanation for this difference lies in the applied dosages. In our study, MG intake through the PNe equals to 34 mg/kg BW, whereas the dietary supplementation of Apontes et al., results in an approximate dose of 400 mg/kg BW. In vitro, MG was shown to exert DPP-4 inhibitory activity and a single dose of MG was effectual to elevate plasma GLP-1 levels in streptozotocin-induced diabetic rats[58] F3 was the only fraction that augmented circulating GLP-1 concentrations after an oral glucose load, which might arise from its weakened or delayed degradation due to DDP4-inhibition by MG.

Metformin is the most frequently prescribed first-line therapy for type 2 diabetes. In our study, the applied metformin concentration (0.25% w/w) translates to a daily human dose of 1300 mg, which is within the recommended therapeutic range of 500–2500 mg per day[59]. In contrast, GE was dosed referring to a daily nutritional exposure by the PN decoction and was not titrated for its full pharmacological response. The results from the hyperinsulinemic-euglycemic clamp emphasize the potency of GE even at this low dose, which had comparable effects on glucose infusion rates as metformin. Future dose-response and toxicology studies will be required to determine the optimal dosing for GE treatment. Although metformin is used as an isolated substance since 1957[60], an unambiguous mode of action is still debated. Its glucose-lowering properties are attributed to the modulation of multiple molecular pathways rather than a single drug-target interaction. This is also reflected in the pleiotropic effects described for metformin, such as suppression of hepatic gluconeogenesis[61] or stimulation of glucose uptake into skeletal muscle[62] due to alterations in cellular redox homeostasis[63,64], reduced hepatic energy state[65], activation of AMPK[62] or mitochondrial complex I inhibition[66]. Similarly, we suggest that genkwanin glycosides improve glucose homeostasis, in part by upregulating *Glut*1 in metabolically active adipocytes. We speculate that these adipocytes function as glucose sink, which blunts glucotoxicity in other tissues, thereby improving or preserving systemic insulin sensitivity and safely storing excess glucose in its dedicated organ. GE triggered glucose disposal into the iWAT and iBAT but not eWAT and this tissue distribution pattern coincides with the occurrence of brown or beige/brite adipocytes in mice housed at room temperature[67]. Data from female BAT-specific *Rab10* knockout mice support the notion that catabolic adipocytes contribute to systemic glucose metabolism independent of their thermogenic role[68]. Moreover, GE treatment was associated with a decreased eWAT to iWAT ratio, indicating that more glucose is directed to the subcutaneous depot causing a shift in fat distribution. This preferential accumulation of subcutaneous rather than visceral fat could additionally impact metabolic health[40,69]. At least in extreme conditions like long-term cold exposure, stable glucose isotope tracing studies in mice revealed that anabolic and catabolic processes co-occur in brown adipocytes[70]. It remains unclear whether GE simultaneously stimulates de novo *lipogenesis* and glucose utilization in metabolically active adipocytes or whether improved insulin sensitivity in other adipocyte types promotes subcutaneous lipid storage.

Likewise, we cannot exclude that GE upregulates *Glut*1 in other tissues and *Glut*1 overexpression in skeletal muscle was shown to induce insulin resistance[71]. However, GLUT1 protein levels were unchanged in skeletal muscles and skeletal muscle insulin sensitivity was not modulated by GE as evidenced by our clamp data. This data

further supports that reduced ATK S473 phosphorylation in the gastrocnemius of GE mice does not impair insulin signaling efficiency and may be influenced by food intake when measured in the random fed state. Apart from the proposed GLUT1-mediated action, our data supports alternative or additional mechanisms including direct insulin-sensitizing effects, increased insulin-dependent or independent glucose uptake via GLUT4-PM redistribution, urinary glucose excretion or liver-mediated effects due reduced gluconeogenesis. Indeed, higher levels of AKT phosphorylation at S473 in the iWAT and iBAT of random fed GE mice point up towards a relevant contribution of GLUT4-mediated glucose clearance. If this is consequential to ameliorated insulin sensitivity or a primary mechanistic pathway engaged by GE that adds to higher adipose tissue glucose uptake in the fasted state remains yet elusive. Moreover, a molecular docking study identified GE as potential glucokinase activator, which could additionally modulate hepatic glucose metabolism as suggested from our clamp experiment[72]. We were unable to elucidate a detailed mode of action for GE activity, as our functional and mechanistic assessment of GE on adipocyte glucose metabolism in cell culture models revealed no detectable effects. The same holds true for the assayed biotransformation products of GE. Apigenin reduced glucose uptake, which aligns with studies reporting the inhibitory action of apigenin on adipogenesis[42] and GLUT1-mediated glucose uptake in cancer cells[73]. Although we did not observe any positive effect of tectochrysin (TEC, 1 µM, and 10 µM) on glucose uptake in iBAs (Supplementary Fig. 12), Zhang et al., demonstrated that TEC at 0 to 40 µM promotes glucose disposal in 3T3-L1 and C2C12 cells by stimulating AKT S473 phosphorylation. However, direct comparability with our experiment is limited as the duration and timing (progenitor *versus* differentiated state) of TEC treatment is unclear. Interestingly, they also reported that 5 weeks of TEC administration by oral gavage (20 mg/kg BW) improves systemic insulin sensitivity during hyperinsulinemic-euglycemic clamps and enhances glucose disposal into metabolic tissues (iWAT, gastrocnemius muscle[74]), concomitant with higher AKT activity. Naringenin, another GE-derived compound, exerts insulin-sensitizing effects and restores glucose tolerance after 16 weeks of DI (3% w/w in HFD) in obese mice[75]. These metabolic improvements were linked to reduced eWAT accumulation and adipocyte size, but no definite mode of action was identified[75]. These examples underpin relevant aspects of our findings. First, the exact treatment regime is key to capture potential bioactivity in vitro, ensuring sufficient stimulation without oversaturating the model system. Second, distinct metabolization products of GE (3.5 mg/kg BW) applied at higher doses ameliorate metabolic dysfunction associated with obesity by modulating different physiological parameters and molecular targets. This ultimately encourages future studies on GE using escalating doses and supports the concept that distinct pathways may contribute to its efficacy.

In conclusion, we could show that reverse pharmacology is a valuable tool to identify undescribed compounds with pharmacological activities when their chronic human use and observed outcomes have been recorded in traditional medicines. To our knowledge, this is the first comprehensive study, which successfully completed the reverse pharmacology approach targeting insulin resistance. Selected based on a statistical association between traditional treatment and outcome as well as the measured clinical efficacy of PNe, we identified GE glycosides as one active principle and we elucidated adipocyte GLUT1 regulation and elevated AKT signaling as contributive mode of actions. Additionally, the effects of GE are comparable to metformin and highlight future possibilities to counteract hyperglycemia by augmenting glucose clearance possibly in a GLUT1-dependent manner.

## Limitations
Our study is limited to the DIO mouse model due to regulatory constraints and only one dose adopted from dietary PNe supplementation

was tested to date. Despite known effects of PNe in subjects with diabetes and the study on human adipocytes presented here, the efficacy and proposed mode of action of GE glycosides remain to be investigated in humans, which requires an in-depth toxicological assessment of GE as well as dose-finding studies.

Our exploratory study intended to characterize the metabolic improvements induced by a PNe fortified diet in DIO mice and to decipher the relevant bioactive molecules. In our experimental design, we therefore included different durations of initial HFD feeding or dietary interventions. While this allowed us to describe the effects of the investigatory supplements at varying degrees of insulin sensitivity or depending on exposure time, it limits the comparability of the results between experiments. There are notable discrepancies and limited translatability between our in vitro and in vivo results regarding the bioactivity of GE, which hindered in-depth mechanistic studies for GE.

A genetic mouse model with adipose-specific deletion of *Glut1* could further strengthen our proposed molecular mechanism of action via GLUT1-mediated glucose uptake. However, it can be expected that in such a model compensatory mechanism might impact systemic metabolism. Furthermore, our study does not resolve if the insulin-sensitizing effect of GE is a consequence of the reduced systemic glucose load or a distinct primary event. We could conclude that GE glycosides confer some of the therapeutic effects of PNe, but we cannot exclude that other compounds contribute to the biological activity of PNe phenotype or influence GE glycoside action in an additive or synergistic effect. In line with this, we reiterate that our hypothesis suggesting that GE is a metabolite originating from genkwanin 5-*O*-β-primeveroside and genkwanin 5-*O*-β-glucoside after oral intake and is found within trace amounts only in the PNe per se. Although the cleavage of *O*-glycosylated flavonoids by intestinal metabolism was previously described[37], this phenomenon on PN *O*-glycosides needs to be verified by additional studies.

## Methods

### Ethics and inclusion statement

This collaborative and global study adheres to the principles of ethical research, ensuring equitable collaboration, fair acknowledgment of all contributors in line with the Global Code of Conduct for Research in Resource-Poor Settings and Nature Portfolio's authorship guidelines. Material transfer agreements (MTAs) were signed between the ETH Zurich, University of Geneva, and the Pacific Academic Institute for Research in Palau, a Palau-based non-profit organization (PAIR, represented by BG). The MTAs regulated the use of Palauan genetic material usage, intellectual property derived from the research findings and the applicable framework to negotiate potential benefits, if any. Local scientists and medical doctors from the Republic of Palau (CK, VY) were involved throughout the research process. The medicinal plant (*Phaleria nisidai*) was selected based on local research findings and the freeze-dried *Phaleria nisidai* extract used in this study was prepared according to the traditional knowledge[18,19,34]. This study informs the local usage of the traditional *Phaleria nisidai* tea within the formal health services in the Republic of Palau, where its application as adjuvant to stabilize hyperglycemic was identified in a locally conducted clinical trial.

### Materials availability

This study did not generate new unique reagents. The *Phaleria nisidai* leaves are available upon request to the lead contacts.

### Experimental models

**Mouse models.** All animal experiments were approved by the Veterinary Office of the Canton of Zürich, Switzerland (ZH220/2020 and ZH221/2020). Male C57B6N mice were obtained from in house breeding or purchased from Charles River laboratories. Experimental mice were housed in groups of 2–4 animals in individually ventilated cages with a reversed 12-h dark/light cycle (lights on 7 pm) at a housing temperature of 23 °C with 40% humidity. Mice had *ad libitum* access to standard laboratory chow diet (18% proteins, 4.5% fibers, 4.5% fat, 6.3% ashes, Provimi Kliba SA) and water. Diet was switched to high fat diet (23.9% proteins, 4.9% fibers, 35% fat, 5.0% ashes, Provimi Kliba SA) at age of 6–7 weeks. Dietary intervention started after 6–12 weeks of initial HFD-feeding depending on the experiment as indicated in the figure legends. The health state of the mice was regularly checked according to FELASA guidelines and food intake measured at the beginning of the DI to exclude potential food aversion.

**Dietary supplementation.** Animal diets supplemented with PN, F2, F3, F4, isolated compounds or metformin were produced in house from powdered high fat diet with the same composition as above (Provimi Kliba SA). Metformin was added at a dose of 0.25% and has previously shown to be effective (Matsui et al., 2010). For the concentrations of compounds and fractions see section" Fraction preparation and dosage for in vivo testing". The required amounts of supplements were calculated based on a daily food intake of 3 g per mouse. The PN extract and fractions were first dissolved in water and then mixed with the HFD powder (20 mL water per 100 g HFD) using an electronic kitchen machine. The right amounts of single individual compounds were dissolved in DMSO (final DMSO concentration in food 0.001%) small droplets of human grade food colorant (Betty Bossi, Coop) were added and dough mixed until homogenous. Pellets were formed by hand, dried overnight and stored at 4 °C. Food was prepared every 10 days to ensure stability of the compounds.

**Cell culture—immortalized brown adipocytes.** This cell line was kindly provided by Prof. Klein[76] and produced from stromal-vascular fraction isolated brown preadipocytes of the iBAT of male and female C57BJ mice by immortalization with the SV40 antigen. Cells were used at passages 4–10. Preadipocytes were cultured in complete media consisting of high glucose DMEM supplemented with 10% heat-inactivated FBS (Gibco) and 1% Pen/Strep (Gibco) in a normoxic, humidified cell culture incubator (37 °C, 5% $CO_2$). Brown preadipocytes were grown on collagen-I (1:500) coated p10 plates and grown until confluency. Post-confluent cells were induced for differentiation into adipocytes with an induction cocktail containing insulin (20 nM), IBMX (500 μM), dexamethasone (1 μM) and T3 (1 nM) in complete media. 2-days post-induction, media was replaced by maintenance media with insulin (20 nM) insulin and T3 (1 nM) and this media was exchanged every other day. At day 5 of differentiation, cells were trypsinized and replated on 24- or 12-well dishes for experiments at a density of 400,000 cells/mL. Long-term treatments with PNe were performed on three subsequent days (d6, d7, d8). PNe was directly dissolved into the media at the indicated concentrations and the solution freshly prepared before use every day. Cells were harvested or experiments performed on day 9 of differentiation. Short-term PNe stimulation was performed at day 9 of differentiation in serum-free media.

**Cell culture—3T3-L1 adipocytes.** 3T3-L1 fibroblasts are a subclone of the 3T3-Swiss albino cell line, which was generated from murine embryonic tissue by spontaneous clonal immortalization[77]. 3T3-L1 fibroblasts were obtained from the ATCC whereas the original passage of this cell line is unknown we used the cells up to 11 passages. 3T3-L1 fibroblasts were cultured in complete media consisting of high glucose DMEM supplemented with 10% heat-inactivated FBS (Gibco) and 1% Pen/Strep (Gibco) in a normoxic, humidified cell culture incubator (37 °C, 5% $CO_2$). 3-days post-confluent 3T3-L1 cells grown on collagen-I coated (1:500) p10 dishes were induced for 48 h with an induction cocktail containing insulin (170 nM), IBMX (500 μM), rosiglitazone (and dexamethasone (1 μM). Media was replaced with insulin media (170 nM) for another 2 days and cells were subsequently maintained in

standard DMEM without additives and media changed every second day. Treatments and replating were performed as outlined above under "immortalized brown adipocytes".

**Cell culture—hMADS.** Human multipotent adipose-derived stem (hMADS) cells originating from the prepubic fat pad of a 4-month-old male were kindly provided by Dr. Amri and cultured according to refs. [78,79]. Briefly, cells (passage 15) were grown in low glucose DMEM supplemented with 15 mM HEPES, 10% FBS, 2 mM L-glutamine, 1% Penicillin/Streptomycin, and 2.5 ng/ml recombinant human FGF-2 in normoxic humidified cell culture incubator (5% $CO_2$ and 37 °C). Differentiation of 48 h post-confluent cells was induced (day 0) by adipogenic medium (DMEM/Ham's F12 media containing 10 µg/ml Transferrin, 10 nM insulin and 0.2 nM triiodothyronine) supplemented with 1 µM dexamethasone and 500 µM isobutyl-methylxanthine (IBMX) From day 2 to 9, cells were cultured in adipogenic medium containing 100 nM rosiglitazone and medium was refreshed every 2–3 days. Cells were kept in culture until day 18 in absence of rosiglitazone to obtain mature white adipocytes. To obtain brown adipocytes, cells were exposed to an additional rosiglitazone pulse between days 14 and 17. To investigate the effect of the plant extract on white and brown adipocyte functionality and GLUT1 levels, cells were exposed to different concentrations of the extract (1.6–200 µg/ml) between day 14 and 17. Fresh medium was provided every 24 h. Adipocytes were cultured until day 17, when cellular respiration was determined, or cells were harvested for protein analysis.

**Primary adipocyte isolation and culture.** Inguinal white and interscapular brown preadipocytes were isolated from 4 to 7-week-old wild-type, male C57BL6/N mice. Inguinal white (iWAT) and interscapular brown (iBAT) adipose tissue was collected, quickly washed in ice-cold PBS (Thermo Fisher, 10010056) and minced into small pieces. The minced tissue was suspended in sterile-filtered, pre-warmed collagenase buffer (25 mM $NaHCO_3$ (Sigma, S5761), 12 mM $KH_2PO_4$ (Sigma, P9791), 1.2 mM $MgSO_4$ (Sigma, 63138), 4.8 mM KCl (Sigma, P9333), 120 mM NaCl (Merck, 106404), 1.2 mM $CaCl_2$ (Sigma, C1016), 5 mM Glc (Sigma, G7021), 2.5% BSA (Sigma, A7906), 1% P/S (Gibco, 15140122), pH 7.4) containing 2 mg/ml type 2 collagenase (Sigma, C6885) and incubated for 1 h at 37 °C (5% $CO_2$) with gentle resuspension after 45 min. Following digestion, the stromal vascular fraction (SVF) was equally diluted in full DMEM (Gibco, 41966029) containing 10% FBS (Gibco, 10500-064) and 1% P/S (Gibco, 15140122), sterile filtered (40 µM sieve for brown preadipocytes, 100 µM sieve for white preadipocytes) and centrifuged for 4 min at 300 g. The supernatant was discarded and the SVF resuspended in full DMEM. After another centrifugation at 300 g for 4 min, the supernatant was again discarded and the pellet resuspended in an appropriate volume of full DMEM, plated and placed into an incubator (37 °C, 5% $CO_2$). Upon reaching confluency, cell differentiation was induced for 2 days. Brown preadipocytes received full DMEM complemented with 20 nM insulin (Sigma, I9278), 0.5 mM IBMX (Sigma, I7018), 125 µM indomethacin (Sigma, I7378), 1 µM dexamethasone (Sigma, D4902), 1 nM T3 (Sigma, T6397), and 1 µM rosiglitazone (Adipogen, AG-CR1-3570-G001). White preadipocytes were induced in full DMEM containing 5 µg/ml insulin (Sigma, I9278), 0.5 mM IBMX (Sigma, I7018), and 1 µM dexamethasone (Sigma, D4902). The following days, the primary brown adipocytes received full DMEM containing 20 nM insulin (Sigma, I9278) and 1 nM T3 (Sigma, T6397), and the primary white adipocytes' full DMEM was complemented with 5 µg/ml insulin (Sigma, I9278). Experiments were conducted on day 8 or 9 after induction.

**Primary human subcutaneous adipocytes.** Primary human preadipocytes were kindly shared by Dr. Jozef Ukropec. The cells were isolated from an abdominal subcutaneous adipose tissue biopsy from a male subject as previously described[80,81]. The study protocol was approved by the Ethical committee of the University Hospital in Bratislava and it conforms to the ethical guidelines of the 2000 Helsinki declaration. Written informed consent was obtained from each patient. In brief, adipose tissue samples were washed with PBS supplemented with gentamycin and digested with collagenase I for 40 min at 37 °C. The digested fat was centrifuged at 350 g for 5 min and pellet containing stroma-vascular fraction (SVF) was incubated in erythrocyte lysis buffer for 10 min at room temperature. Cells were collected by centrifugation and resuspended in proliferation medium DMEM/F12 supplemented with L-glutamine, 33 µM biotine, 17 µM d-panthotenate, 10 µg/ml transferrin, 50 µg/ml gentamycin, and 1% penicillin/streptomycin, 10% FBS, 1 ng/ml FGF-2). The proliferation medium was changed every 2–3 days until the cells reached 70% confluence, then they were subcultivated. For experiments, cells were seeded on 12-well plates and adipogenic differentiation was induced on 3 days post-confluent cells with a cocktail of 200 nM Rosiglitazone, 100 nM Dexamethasone, 100 nM insulin, 2 nM T3, 540 µM IBMX, and 10 ug/mL transferrin dissolved in DMEM/F-12 containing 1% Pen/Strep, 1 mM Hepes[82]. The same media was added 3 days later without IBMX, after day 6 of differentiation the Rosiglitazone was omitted from the media. Mature adipocytes were treated with PNe from day 9 to day 12 of differentiation.

**C2C12 skeletal muscle cells.** C2C12 skeletal muscle cells were grown in high-glucose DMEM supplemented with 10% heat-inactivated FBS (Gibco) and 1% Pen/Strep (Gibco) in a normoxic, humidified cell culture incubator (37 °C, 5% $CO_2$). Cells were seeded on 24-well plates or 5 cm dishes depending on the experiment and differentiation into myotubes was induced by reducing the serum content of the media to 2% horse serum (Gibco). Media was changed every second day and treatment performed at days 4, 5, and 6 of differentiation.

## Methods details

**Tissue harvest.** Animals were euthanized individually with a CO2 overdose and blood collected by cardiac puncture with EDTA (0.5 M) flushed syringes. Whole blood was spun for 20 min at 2000 × g and blood plasma stored at −80 °C until downstream analysis. Tissues were carefully dissected, snap frozen in liquid nitrogen and stored at −80 °C for protein extraction. For tissue specific glucose uptake measurements and glucose clamps studies, the animals were euthanized by cervical dislocation.

**Body composition.** Lean and fat mass of concious mice were determined with a magnetic resonance imaging system (EchoMRI 130, Echo Medical Systems). Data were calculated by the EchoMRI 14 Software.

**Cumulative food intake.** Phenomaster metabolic cage system (TSE system) was used to measure cumulative food intake (Supplementary Fig. 7b) and data acquired with Phenomaster software v5.6.5. Mice were hold individually and adopted to the single caging for 2 days before measurements started.

**Intraperitoneal insulin tolerance test.** Mice were fasted for 6 h at the beginning of the active phase. An intraperitoneal dose of insulin (0.75–1,2 U/kg BW, Actrapid human insulin, Novo Nordisk) was injected after recording of fasting blood glucose concentration (ACCU-CHEK Aviva, Roche) through a small incision in the tail. Blood glucose concentrations were monitored for 2 h, and blood samples measured at 0, 15, 30, 60, 90, and 120 min. Area of the curve (AOC) was calculated as described in by Virtue and Vidal-Puig[83].

**Oral and intraperitoneal glucose tolerance test.** Mice were fasted for 6 h at the beginning of the active phase. An intraperitoneal or oral dose of glucose (2 g/kg BW) was administrated after recording (Accu-Check, Roche) of fasting blood glucose concentration (Accu-Check, Roche)

through a small incision in the tail. Blood glucose concentrations were monitored for 2 h and blood samples measured at 0, 15, 30, 60, 90, and 120 min. Area of the curve (AOC) was calculated as described by Virtue and Vidal-Puig[83].

**Glucose-stimulated insulin secretion.** Mice were fasted for 6 h at the beginning of the active phase. A blood sample was collected with a glass capillary from the tail to determine fasting blood glucose levels. An oral dose of glucose (2 g/kg BW) was administrated via gavage and 30 min later a second blood sample collected to determine glucose-stimulated insulin levels. Blood samples were centrifuged for 8 min at $8000 \times g$, and plasma collected. Insulin concentrations were determined with the MescoScale Discovery kit.

**GLP-1 measurement.** Mice were fasted for 6 h and were orally given 200 μL of 40% glucose solution. 2 min later, a blood sample was collected from the tail vein with a micro cuvette, and DPPV-4 inhibitors added. Whole blood was spun at 4 °C at $8000 \times g$ for 8 min to collect the plasma. Circulating, active GLP-1 levels were measured with the active GLP-1 MesoScale Discovery Elisa Kit.

**Urinary glucose extraction.** Urine was collected after 6 h of food deprivation at the beginning of the light cycle by mildly restraining the mouse and massaging the bladder. Urine was stored at −80 °C until glucose concentrations were determined using a colorimetric glucose assay kit (MAK263-1KT, Sigma).

**Metabolic blood parameters.** The following commercially available kits were used determine triglycerides (11877771 216, Roche/Hitachi), ultrasensitive mouse Insulin Elisa Kit (90080, Crystal Chem), non-esterified fatty acids (436-91995 and 434-91795, FUJIFILM Wako Chemicals), cholesterol (11491458 216, Roche/Hitachi) or ALT activity assay (MAK052-1KT, Sigma) in blood plasma samples according to the manufacture's guidelines.

**Tissue-specific glucose uptake.** Animals underwent an initial HFD-feeding period of 6 weeks and glucose uptake was assessed after 6–7 weeks of dietary intervention with PNe or GE. were fasted for 6 h at the beginning of the dark cycle. 100 μL of were injected via the tail vein and animals were sacrificed after 30 min by cervical dislocation, tissues harvested and snap frozen in liquid nitrogen. To determine tissue-specific glucose uptake, the tissue was lysed in 2 mL of $H_2O$ and homogenized using a tissue homogenizer from POLYTRON. Phosphorylated $^{14}C$-2-deoxyglucose was isolated from the tissue lysate using chromatographic columns (7316212, BioRad). 1 mL of lysate was incubated in 5 mL of liquid scintillation fluid (Ultima Gold) overnight and counted with a Perkin Elmer TriCarb 2000 CABeta-Counter for 5 min per sample. DPM were normalized to initial tissue weight.

**Hyperinsulinemic-euglycemic clamps.** Animals underwent an initial HFD-feeding period of 6 weeks followed by 7–8 weeks of dietary intervention with F4 or GE. Hyperinsulinemic-euglycemic clamps were performed in line with published protocols[84,85]. A polyurethane catheter was inserted into the left jugular vein and exteriorized at the next region under isoflurane anesthesia. Prior to surgery, Vitamin A salve was applied onto the eyes and the analgesic fentanyl (50 μg/kg bodyweight) was administrated. The catheters were filled with heparinized-glycerol solution. A single subcutaneous dose of nor-ocarp was given 16 h post-surgery. Body weight and health state were monitored daily. Animals with less than 10% loss of preoperative weight were included in the subsequent clamps and animals could recover for at least four days. Animals were starved for 6 h prior to clamping. Catheters were flushed with 0.9% NaCl and mice placed into restrainers, which allowed them to move freely. Basal glucose production was measured for 80 min with a continuous $^3H$-glucose

infusion. The clamps were performed with the HARVARD apparatus and insulin (Actrapid Human Insulin, Novo Nordisk) infusion rate was set to 12 mU/kg$^{-1}$ × min$^{-1}$. Glucose was infused at a variable rate and target blood glucose concentration set to 6 mM. Blood glucose levels were monitored every 5 or 2.5 min with a glucometer (Akku-Check Aviva, Roche) and infusion rates adjusted accordingly. Steady state glucose infusion rate was recorded when the glucose infusion rate remained constant with only minor deviations for 15–20 min at a blood glucose concentration of 6 mM. Once steady state conditions were reached, a single bolus of $^{14}C$-2-deoxyglucose was administrated through the catheter to determine tissue-specific glucose uptake rates over 35 min. Blood samples were collected at times = 0 (bolus), 2-, 15-, 25-, and 35-min post injection. Mice were sacrificed by cervical dislocation and the following tissues collected for analysis: iWAT, eWAT, soleus, gastrocnemius, BAT, and liver. The area under the curve of the disappearing plasma $^{14}C$-2-deoxyglucose was combined with the tissue counts of phosphorylated $^{14}C$-2-deoxyglucose to calculate tissue specific glucose uptake rates. Tissue samples were weighed and homogenized as above under "Tissue-specific glucose uptake". Phosphorylated 2-deoxyglucose was isolated using chromatographic columns (7316212, BioRad)[86] and 1 mL of lysate counted in 5 mL of liquid scintillation cocktail (Ultima Gold) for 5 min with the Perkin Elmer TriCarb 2000 CA Beta-Counter after overnight incubation.

**mRNA isolation and qPCR.** mRNA was isolated with TRIzol reagent (Invitrogen) according to the manufacture's instruction. A 30 min DNAse treatment was included to remove genomic DNA (NEB Bio-Labs). 1000 ng of total RNA was reverse transcribed into cDNA using the High Capacity Reverse Transcription kit (Applied Biosystems). cDNA was diluted to 5 ng/uL with $ddH_2O$. Quantitative RT-PCR was performed with Fast SYBR green (Applied Biosystmes) on a 384-well format Viia7 machine (Applied Biosystems). qPCR Primers were designed with the online NCBI primer design tool and synthesized by Microsynth. Relative mRNA expression was calculated by the delta-delta Ct method in Microsoft Excel. TBP served as housekeeping genes. All qPCR primer sequences are listed in Supplementary Table 2.

**Protein extraction and western blot analysis.** Whole cell protein was extracted from adipose depots and cell cultures using RIPA Buffer (50 mM Tris-HCl pH 7.4, 150mMNaCl, 2 mM EDTA, 1.0% Triton ×100, 0.5% sodium deoxycholate and 10% glycerol for adipose tissue) supplemented with protease (Complete, Roche) and phosphatase inhibitors (Haltphosphatase inhibitor cocktail, ThermoFisher). The lysate was spun at $12,000 \times g$ for 15 min at 4 °C and the supernatant carefully collected. Protein concentrations were determined with the DC Protein Assay (Bio-Rad). For Western Blotting, equal amounts of proteins were loaded onto 12% or 8% SDS-Polyacrylamide gels and transferred on nitrocellulose (Bio-Rad) membranes. Membranes were blocked for 1 h at RT with either 5% BSA or 5% fat free milk depending on the antibody. The following primary antibody bodies were used and incubated overnight at 4 °C GLUT4 (1:1000, Merck 07-1404), GLUT1 (1:1000, Merck 07-1401), GLUT1$^{pS226}$ (1:200, Merck ABN991), AKT (1:0000, Cell Signaling 9272), AKT$^{pS473}$ (1:1000, Cell Signaling 4060), AKT$^{pT308}$ (Cell Signaling 13038), ERK1/2 (1:1000, Cell Signaling 4695). ERK1/2$^{pThr202/Tyr204}$ (1:1000, Cell Signaling 4370), HSP90 (1:1000, Cell Signaling 4877), GADPH (1:1000, Cell Signaling 5174), phosphoPKC Substrates (1:1000, Cell Signaling 2261), PPARγ(1:1000, Cell Signaling 2443). The HRP-conjugated secondary antibody (1:10,000 Cell Signaling) signal was detected with the Image Quant system (GE Healthcare Life Sciences). GLUT1 and GLUT4 bands appear between approximately 50–55 kDa and are marked as 55 kDa in the figures.

**Plasma membrane fractionation.** Cell lysate and plasma membrane fraction was prepared as described by Yamamoto et al.[87]. In brief, cells

were grown, differentiated, and treated in collagen-coated 60-mm culture dishes. After two washes with cold PBS (Thermo Fisher, 10010056), cold 180 µl buffer A0.1% (Buffer A (50 mM Tris-HCl (AppliChem, A1086), 0.5 mM DTT (Sigma, D0632), 1 mM Na$_3$VO$_4$ (Sigma, S6508), complemented with fresh 100 × protease inhibitor, 100 × phosphatase inhibitor and 0.1% Igepal (Sigma, I3021)) was added and cells were scraped from the dish. Cell suspension was transferred to 1.5 ml Eppendorf tube and homogenate was first passed through a 200 P pipette tip, then through a 24 G needle to shear the DNA and liberate the proteins. 30 µl of the cell lysate was transferred to a 1.5 ml Eppendorf tube, diluted with 30 µl cold 2× RIPA (20 mM Tris-HCl (AppliChem, A1086), 300 mM NaCl (Merck, 106404), 1% sodium deoxycholate (Sigma, D6750), 0.1% SDS (Sigma, 75746), 1 mM DTT (Sigma, D0632), 2 mM Na$_3$VO$_4$ (Sigma, S6508), complemented with fresh 50 × protease inhibitor, 50 × phosphatase inhibitor and 2% Igepal (Sigma, I3021)), and stored on ice for 60 min. The rest was centrifuged for 1 min, 200 × $g$ at 4 °C. The supernatant was transferred to a new 1.5 ml Eppendorf tube, the pellet got resuspended in 90 µl buffer A0.1% and again centrifuged for 1 min, 200 × $g$ at 4 °C. The supernatant was added to the previous, and the combined supernatants were then centrifuged for 10 min, 750 × $g$ at 4 °C. The supernatant, which is the cytoplasmic fraction, is transferred to a new 1.5 ml Eppendorf tube and stored on ice for 60 min. The pellet, which contains the membrane proteins with detergent and membrane lipid, is suspended in 90 µl buffer A0.1% and centrifuged for 10 min, 750 × $g$ at 4 °C. The supernatant containing soluble proteins was discarded, and the pellet is resuspended in 45 µl buffer A1% (buffer A complemented with fresh 100 × protease inhibitor, 100 × phosphatase inhibitor, and 1% Igepal (Sigma, I3021)) and incubated on ice for 60 min. All fractions got mixed every 10 min by hand tapping. Following incubation, the fractions were centrifuged for 20 min, 12,000 × $g$ at 4 °C. The supernatants were transferred to new 1.5 ml Eppendorf tubes and stored at −80 °C until further use. For Western blots, 7.5 µg–15 µg total protein was loaded. Due to the highly limited amount of protein available in the PM-fraction and available cell material, GLUT1 western blot was prioritized for the primary human adipocytes.

**Glucose uptake assay.** Glucose uptake assays were done in 24-well plates as described elsewhere[88] after 1 h of insulin stimulation. Cells were serum-starved for 3 h in the presence or absence of the specified stimuli. ERK1/2 inhibitor (PD184352) or PKC inhibitor (Gö−6983) were added 30 min before end of starvation. Insulin (Actrapid Human Insulin, Novo Nordisk) was added to a final concentration of 100 nM and cells stimulated for 50 min. Plates were carefully washed 2× with Krebs-Ringer-Hepes (KRH) Buffer (50 mM HEPES, 137 mMNaCl, 4.7 mMKCl, 1.85 mMCaCl2,1.3 mM MgSO4, pH 7.4) with 0.1% fatty acid free BSA to remove any extracellular glucose. Cells were then incubated at 37 °C with 0.1 µCi $^{14}$C-2-Deoxyglucose in 200 µL of KRH Buffer. The uptake reaction was stopped with 4 thorough, ice-cold PBS washes. Cells were lysed in 0.1 M NaOH, and lysate transferred to 2 mL of liquid scintillation cocktail. 10 µL of cell lysate was kept aside for protein determination. Radioactive decays were counted with a Perkin Elmer TriCarb Beta-Counter for 2 min.

**Glycolytic stress test.** Preadipocytes were grown and differentiated on p10 plates until day 5 of differentiation. Differentiating adipocytes were collected by trypsinization and replated at a density of 7000 cells per well. The cells were treated daily for 72 h. On the day of the assay, cells were washed 3× with PBS and incubated for one hour in the Seahorse XF Base Medium (Agilent, 102353-100) supplemented with 2 mM glutamine (GlutaMAX, Gibco 35050-061). Oxygen consumption rates and extracellular acidification rates of the media were measured with extracellular flux analyzer (Seahorse XFe96, Agilent Technologies) and acquired with the Wave software Wave 2.6.0. After baseline measurements were acquired, the following compounds were successively injected into the media (a) glucose (10 mM final) (b) oligomycin (1 µM final) (c) 2-deoxyglucose (100 mM final). Three measurements were performed after each injection within 3 min intervals. Read out were normalized to protein content per well. Glycolysis, glycolytic capacity and spare capacity were calculated from the normalized values according to the manufacture's guidelines.

**Glycolytic flux analysis.** Glycolytic flux was measured in mature adipocytes after 3 days of PN treatment as described in ref. [89]. Cells were starved with or without PN in KRH (see above) for 2 h and then incubated with labelling media (high glucose DMEM, 0.4 µCi/mL of $^3$H-5-glucose) and cells incubate for 2 h at 37 °C. Empty wells were used to determine the background signal. Meanwhile, hanging wells were inserted into rubber stoppers and a piece of filter paper (1 × 6 cm) was placed in each hanging well. At the end of the incubation period, the filter paper was hydrated with 200 µL of ddH$_2$0. 200 µL of labelling media were then transferred to glass vials and 50 µL of 3 M perchloric acid were added to prevent any further metabolic activity from potentially transferred cells. The glass vials were tightly closed with the rubber stoppers containing the hanging wells and incubated for 48 h at 37 °C. Each filter paper was then carefully transferred into scintillation vials containing 5 mL of liquid scintillation cocktail. The samples were left at room temperature overnight and subjected to liquid scintillation counting (2 min per vial, Perkin Elmer TriCarb). Glycolytic flux was calculated as pmol glucose per hour and normalized to protein content.

**Lipolysis assay.** 3T3-L1 and iBAs were grown to confluency and induced in collagen-coated 12-well plates. On day 8 after induction, cells were serum-starved in low-glucose DMEM (Thermo Fisher, 11880036) for 4 h, whereby for the last 2 h, 1 µM isoproterenol (Sigma, I5627) or PBS (Thermo Fisher, 10010056) was added. The supernatant was collected, and lipolysis activity was measured by assessing glycerol, triglyceride (TAG), and non-esterified fatty acid (NEFA) levels. To measure glycerol levels, 100 µl glycerol reagent (Sigma, F6428) was added to 40 µl sample or standard (Sigma, G7793) and absorbance was read at 540 nm after 15 min incubation at room temperature. For determination of TAG levels, we followed the manufacturer's protocol of the assay kit (Spinreact, 41031). NEFA levels were measured as follows: 100 µl of reagent 1 (Fujifilm, 434-91795) was added to 50 µl sample or standard (Fujifilm, 270-77000), with subsequent incubation at room temperature for 30 min. Next, 50 µl of reagent 2 (Fujifilm, 436-91995) was added to sample or standard. After incubation at room temperature for 7 min, absorbance was measured at 546 nm.

**Adipogenesis assay.** 3T3-L1 or immortalized brown adipocytes were cultured and differentiated in collagen-coated black 96-well plates. Treatment with varying concentrations of PNe started on day −3 for 3T3-L1 cells and on day −1 for immortalized brown adipocytes. On day 6 of differentiation, the cells were washed twice with PBS (Thermo Fisher, 10010056) for 5 min, then fixed in 4% formaldehyde (Sigma, 441244) for 20 min, followed by an additional 5-min wash. The cells were then incubated in a staining cocktail containing 2 µM Hoechst (Thermo Fisher, H3570) and 100 ng/µl LD540 (provided by Dr. Christoph Thiele, Bonn, Germany[90]) in PBS for 30 min at room temperature in the dark. After two more 5-min washes with PBS in the dark, the cells were prepared for imaging with the Operetta 96-well High-Content Imaging System (Perkin Elmer).and images analysed with Harmony v3.5 (Perkin Elmer).

**Immunofluorescence.** Differentiating adipocytes were replated at day 5 of differentiation into glass-bottom 96-well plates at a density of 35,000 cells. Cells were washed 2× with PBS, fixed (4% formaldehyde, 20 min at 4 °C) and washed again 3× with PBS. Blocking (3% BSA) and permeabilization (0.1% Triton X-100) were performed at room

temperature for 1 h or 10 min, respectively. Primary anti-ERK1/2 antibody (1:500, Cell Signaling 4695) was incubated overnight at 4 °C. After extensive washing, alexa488-conjugated secondary antibody (1:750, Life Technologies A21206) was applied for 1 h at RT. Nuclei were stained with Hoechst (1:10,000, Cell Signaling 4082) in PBS for 10 min. Images were acquired with the automated Operetta microscope using a 20× objective for quantification or with a 40× objective in an independent experiment for representative pictures (Perkin Elmer). The acquired images were analyzed to determine the percentage of cells positive for nuclear ERK1/2 localization in response to PNe stimulation. The image processing was performed with a custom scripts developed in Python using scikit-image library and standard image processing steps. Initially, nuclei were segmented based on Hoechst signal intensities, followed by filtering for size and circularity. Next, ERK1/2 (Alexa488) staining defined the cell areas. We then quantified the average Alexa488 signal within the segmented nuclei and a 3-pixel wide ring around the nucleus perimeter, representing the cytosolic area, constrained to the cell areas identified in the second step. To improve segmentation accuracy, the nuclear area was reduced by 1 pixel around the perimeter, and the surrounding ring began 1 pixel away from the nuclear border. A cell was deemed positive for nuclear ERK1/2 localization if the ratio of average nuclear to estimated cytosolic Alexa488 signal intensities exceeded 4.

**Quantification and statistical analysis.** Data are presented as mean ± SEM or ±SD as indicated in the figure legends. Statistical analysis was performed using GraphPad Prism (versions 6–10). Sample sizes were selected based on previous experience and findings for a certain method. The authors were not blinded during the procedures or data analysis. The exact statistical test and n-number for experimental group are reported in each figure legend. In vitro experiments with murine cell lines were performed with 3 technical replicates for protein, 4–6 replicates for radioactive assays and 3–6 replicates for RNA isolation. Each experiment was independently reproduced 2–3 times and the displayed datapoints are the pooled measurements from the independent repetitions. Few experiments were performed once, while the exact number of independent experiments for is specified in the corresponding figure legend. Potential outliers were identified and removed with the Grubb's test. Statistical significance was tested as stated for each panel in the figure legend with $*p < 0.05$ considered as significant difference. Comparisons were performed against the HFD-control group to evaluate the effects of the PN extract, fractionations or individual compounds. Comparisons were performed between all groups when metformin was introduced as positive control and statistical significance is calculated with uncorrected Fisher's LSD as represented in the respective graphs. This approach is justified because of the low number of groups being compared and the positive control group was included primarily to assess the effect size rather than to test for a significant difference between treatment (F4 or GE) and metformin. Adjusted $p$-values using Tukey's multiple comparison adjustment are reported in Source Data 1. Statistical differences are indicated as * for $p < 0.05$, ** for $p < 0.01$, *** for $p < 0.001$ and **** for $p < 0.0001$.

**Preparation of *Phaleria nisidai* leaf extract.** The leaves of *Phaleria nisidai* Kaneh. (Thymelaeaceae) were collected at the time of the clinical study[19] and reference specimens were deposited at the Belau National Museum Herbarium. The leaves of PN were extracted by decoction according to the traditional recipe used by the population of Palau[19]. The dried leaves were extracted by decoction for 1 h, then freeze-dried to facilitate mouse feeding and chemical investigation. The decoction was performed without a lid to evaporate potentially irritating compounds since the leaves have been reported to contain toxic volatile compounds removed upon boiling[91]. After filtration (60 °C ± 5 °C) and first evaporation with rotavapor to reduce the water

quantity (Büchi SARL, Rungis, France), the decoction was freeze-dried (−80 °C, 0.01 bar, Christ Alpha 2–4 LD plus, Osterode am Harz, Germany) and stored at 4 °C. For the complete study, $200 \times g$ of dried leaves were extracted in two batches of $100 \times g$, in portions of 30 to $40 \times g$, which required a total of 20 L of water. In both batches, the yield of extraction was 28.6% ± 0.7% (average ± standard deviation, $n = 7$) (w/w). The volume of water lost by evaporation during the decoction was measured at 1.4 L ( ± 0.0 L, $n = 3$), independent of the initial volumes (respectively at 4.0 L and twice 3.0 L).

**Chemical profiling.** Chemical profiling was acquired on two independent UHPLC systems, one equipped with photo diode array (PDA) and Charged Aerosol Detector (CAD) (UHPLC-PDA-CAD), and the second with PDA and high-resolution mass spectrometer (UHPLC-PDA-HRMS/MS), to obtain qualitative (UV and HRMS/MS spectra) and semi-quantitative (CAD) information. The chromatographic conditions were the same for both systems: samples were injected (2 μL) into an Acquity UPLC BEH C18 column (2.1 × 150 mm i.d., 1.7 μm, Waters, Milford, MA, USA) and eluted (0.4 mL/min, column temperature set at 40 °C) with water (A) and acetonitrile (B) both containing 0.1% formic acid with the following gradient: from 5 to 50% of B from 0 to 25 min (curve 7), 50 to 95% of B from 25 to 30 min, an isocratic step at 95% for 5 min and a re-equilibration step of 5 min. Semi-quantitative profiling (UHPLC-PDA-CAD) were acquired with a Vanquish system (Thermo Fisher Scientific, Waltham, MA, USA), which was equipped with an UHPLC hyphenated with a PDA (HL-type) and a CAD (VF-D20A type) (set at 35 °C, data collection rate at 10 Hz), controlled by the software Chromeleon 7.2.9. HRMS/MS chemical profiling were performed on an Acquity UPLC system interfaced to an Orbitrap Q-Exactive Focus mass spectrometer (Thermo Scientific), using a heated electrospray ionization (HESI-II) source. Xcalibur 2.1 software (Thermo Scientific) was employed for instrument control. An Acquity UPLC PDA detector acquired the UV trace from 200 to 500 nm. The full MS analyses were performed in positive mode with a mass range of 150–1500 at a resolution of 35000 full width at half maximum (FWHM) (at $m/z$ 200). In positive mode, diisooctyl phthalate $C_{24}H_{38}O_4$ [M + H]$^+$ ion ($m/z$ 391.28429) was used as internal lock mass. The optimized HESI-II parameters were the following: source voltage: 3.5 kV (pos); sheath gas flow rate (N2): 50 units, auxiliary gas flow rate: 12.50 units, spare gas flow rate: 2.5; capillary temperature: 262.5 °C; S-Lens RF Level: 45. The mass analyzer was calibrated according to the manufacturer's directions using a mixture of caffeine, methionine-arginine-phenylalanine-alanine-acetate (MRFA), sodium dodecyl sulfate, sodium taurocholate and Ultramark 1621 in an acetonitrile/methanol/water solution containing 1% acetic acid by direct injection. The data dependent HRMS/MS events were performed on the three most intense ions detected in full scan MS (Top3 experiment). The HRMS/MS isolation window width was 1 $m/z$ and the normalized collision energy (NCE) was 35 units. In data dependent HRMS/MS experiment, full scans were acquired at a resolution of 35000 FWHM (at $m/z$ 200) and HRMS/MS scans at a resolution of 17 500 FWHM with a maximum injection time of 50 ms. After being acquired in HRMS/MS scan, parent ions were placed in a dynamic exclusion list for 2.0 s.

The samples whose chemical profiling were acquired were solubilized in a solution of water and methanol (7/3 v/v). The freeze-dried decoction of PN and the fractions had a concentration of 2 mg/mL, while precipitated MG was prepared at 250 mg/mL. After sonication (10 min), samples were centrifuged (14,750, 10 min) (Prism R, Labnet international Inc., Edison, NJ, USA).

**Mangiferin quantification in PN extract.** Mangiferin (MG) in PNe was quantified by external quantification at the UV wavelength of 256 nm on the UHPLC-PDA system described in the section above. The chromatographic conditions were as followed: PN extract and calibrations were injected (1 μL) into an Acquity UPLC BEH C18 column

(2.1 × 100 mm, i.d. 1.7 µm, Waters, Milford, MA, USA) and eluted (0.4 mL/min, column temperature set at 40 °C) with water (A) and acetonitrile (B) both containing 0.1% formic acid with the following gradient: from 5 to 50% of B from 0 to 8 min (curve 7), 50 to 95% of B from 8 to 9 min (curve 6), an isocratic step at 95% for 2 min and a re-equilibration step of 1.9 min. The PNe was solubilized at 1 mg/mL in a solution of water and methanol (8/2 v/v) and sonicated 10 min. Four calibration solutions were prepared with MG standard from 50 to 400 µg/mL in the same solution than PN extract. The calibration curve was: y = 178.02× −1001.4 and R² of 0.999. The MG concentration measured in PNe was of 14.5 ± 0.4% (*n* = 4).

**Large scale fractionation of PNe to obtain fractions with complementary compositions for in vivo studies.** To remove the polar compounds that are not retained under chromatographic reverse phase conditions (Supplementary Fig. 5a, b), a vacuum liquid chromatography (VLC) system was employed with a Zeoprep 60 C18 15–25 µm reverse phase (Zeochem® Silicagel, Rüti, Switzerland). After equilibration with 1 L of water, 2 to 2.5 g of extract previously suspended in water and sonicated for 15 min were deposited on the stationary phase and eluted first with 1 L of water and second with 1 L of methanol. The fractionation yield was evaluated for the aqueous fraction (F1) at 41% ± 1% (*n* = 3) and for the methanolic fraction at 50% ± 2% (*n* = 3), for a total yield of 92 ± 1% (w/w).

The methanolic fraction obtained by VLC was then fractionated at a large scale on medium pressure liquid chromatography (MPLC). Gradient optimization was performed at high pressure liquid chromatography (HPLC) scale before being transferred to the MPLC system. At HPLC scale, the enriched extract was separated thanks to an Agilent 1260 HPLC system (Agilent Technologies, Santa Clara, CA, USA) equipped with PDA detection on a Zeoprep column (A90 C18, 250 × 4.6 mm i.d., 15–25 mm, Zeochem, Rüti, Switzerland) at a flow of 1 mL/min of water (A) and methanol (B) both with 0.1% of formic acid (Fisher Scientific, Bishop, UK). After an equilibration time of 20 min, the gradient was optimized from 5 to 25% of B in 10 min, followed by an isocratic step at 25% of B from 10 to 50 min, and finally from 25 to 95% of B from 50 to 70 min. Gradient transfer calculation[92,93] indicated the following MPLC conditions: 140 min from 5 to 25% of B, isocratic step at 25% of 515 min, followed by an increase to 95% of B in 258 min for a total of 15 h of separation at a flow of 15 mL/min. The MPLC system (Büchi) was equipped with a modules pump C-660, a UV detector C-640, and a fraction collector C-684 and was controlled by the software Sepacore Control. The MPLC column (460 × 49 mm i.d., Büchi) was packed with the same solid phase as the HPLC scale. The methanolic fraction of PN (10 g) was introduced in MPLC by a dry load cell and fractionated into 58 fractions of 250 mL, labelled with M for MPLC. All fractions were dried by rotavapor and eventually freeze-dried. After fraction conditioning, the recovery yield of the total MPLC was 83% (w/w). It should be noted that 1.1 g of MG (2) precipitated in four fractions (M17 to M20).

A second batch of MPLC was performed on 7 g of the methanolic fraction. Given the observations made on the first MPLC, the isocratic step at 25% of B was maintained for two additional hours. This increase avoided the semi-preparative step required on the first batch of the crucial fractions (M46 to M48) between the end of MG elution and the beginning of the elution of the most apolar compounds as described below. In addition to MG, iriflophenone 3-*C*-β-glucoside (**1**) and iriflophenone-2-*O*-α-rhamnoside (**3**) were identified in the second batch of MPLC in fractions M12 and M18, respectively.

For the first MPLC fractionation, all fractions were checked in short chromatographic conditions as described in ref. 94 on two complementary systems. The first one was an UPLC system (Waters, Milford, MA, USA), equipped with a PDA detector (Waters) and an ELSD set at 45 °C, gain 8 (Sedex 85, Sedere, Alfortville, France). The second system was a UHPLC-HRMS-TOF, set as described in ref. 94. As the goal was the preparation of fractions depleted in MG in sufficient quantities

for in vivo testing, the fraction control focused on the presence or absence of this constituent. In this first batch, fractions M1 to M14 presented no ELSD signal of MG, whereas fractions M15 to M45 presented MG as the major signal. Notably, precipitation of MG occurred in the fractions M17 to M20, and the precipitate was separated from its solution, yielding 1.097 g of precipitated MG. In addition to MG, the fractions M15 to M18 contained the second most abundant constituent of NP, identified as iriflophenone 2-*O*-β-glucoside (**1**). Among the last fractions, M46 to M48 contained other compounds in addition to MG, whereas M49 to M56 were depleted in MG. A step of purification on these three fractions was subsequently implemented, as described below.

For the second MPLC and the semi-preparative fractionations of fraction F4, fractions were monitored by a UHPLC system equipped with PDA, single quadrupole (QMS), and ELSD (UHPLC-PDA-QMS-ELSD). This three-detector system, controlled by MassLynx® V4.2 (Waters), was equipped with an Acquity UPLC system (Waters), which included a binary pumping system, an auto-sampler (set at 10 °C), a column manager with a pre-column heater (set at 40 °C), a PDA detector and an isocratic solvent manager which directed 10% of the flow to the single quadrupole (Acquity QDA, Waters) while adding a flow of 200 mL/min of water-acetonitrile (1:1) containing 0.1% formic acid. The remaining 90% of the flow was directed to an evaporative light scattering detector (ELSD) (Büchi ELS Detector C-650), set at 45 °C, gain 8. The QDA, equipped with an ESI source, was set as follows in negative mode: probe temperature 600 °C, ESI capillary voltage 1.2 kV, cone voltage 15 V, source temperature 120 °C, acquisition range 30–1250 Da. Fractions were injected (2 µL) into an Acquity UPLC BEH C18 column (1.7 µm, 1 × 50 mm) (Waters) and eluted (0.3 mL/min, column temperature set at 40 °C) with water (A) and acetonitrile (B), both containing 0.1% formic acid with the following gradient: from 2 to 50% of B from 0 to 3.9 min, 50 to 98% of B from 3.9 to 4.4 min, an isocratic step at 98% for 1 min and a re-equilibration step of 2 min.

**MG depletion in hinge fractions of the first MPLC.** To remove residual amounts of MG in the fractions at the end of MG elution (M46 to M48), an additional purification step was implemented. Gradient optimization was performed on the HPLC system previously described on a Uptisphere strategy column (C18-HQ, 250 × 4.6 mm i.d., 15 µm, Interchim, Montluçon, France) before its transfer to Flash chromatography performed on an Interchim system equipped with UV detection. The gradient was optimized as followed: isocratic step at 35% of B for 20 min, followed by a gradient from 35 to 100% of B in 30 min. Gradient transfer calculation[93] indicated the following conditions for a Puriflash column (PF-C18HP, 224 × 36 mm i.d., Interchim, Montluçon, France): isocratic step at 35% of B for 55 min, followed by a gradient to 100% of B in 82 min at a flow of 12 mL/min. A dry load injection was used to purify 400 mg of a mixture of the three fractions proportional to their MPLC yield, mixed with 2 × g of Zeoprep 60 stationary phase (C18 40–63 µm, Zeochem, Rüti, Switzerland). Three fractions were generated, two of which were depleted in MG and represented 46% of the loaded quantity. This additional purification was performed on the first batch of MPLC fractions only and was not required for the second MPLC batch due to the increased time of the isocratic steps.

**Purification of fraction F4 constituents.** Given the in vivo results, the F4 fraction was fractionated to formally identify its constituents. Gradient optimization was performed on the same HPLC system described above on a X-bridge C18 column (250 × 4.6 mm i.d., 5 µm, Waters, Milford, MA, USA) equipped with a pre-column cartridge holder (20 × 4.6 mm i.d., 5 µm, Waters, Milford, MA, USA). After an equilibration time of 20 min, the optimized gradient was from 22 to 40% of B in 100 min, and from 40 to 100% in 30 min.

Semi-preparative HPLC-UV separation was conducted on a Shimadzu system controlled by the LabSolutions software (Shimadzu,

Kyoto, Japan) equipped with a LC-20A module pumps, an SPD-20A UV/VIS, a 77251 Rheodyne® valve and an FRC-10A fraction collector (Shimadzu). The column was a X-bridge C18 (250 × 19 mm i.d., 5 μm, Waters, Milford, MA, USA) equipped with a Waters C18 pre-column cartridge holder (10 × 19 mm i.d.). Gradient transfer calculation (Challal et al., 2015) indicated the following conditions: from 22 to 40% of B in 103 min and to 100% in 32 min, at a flow of 17 mL/min. Two injections were performed by dry loading, by mixing F4 with stationary phase (Zeoprep® C18 40−63 μm, Rüti, Switzerland), respectively 50 mg of F4 with 70 mg of stationary phase and 25 mg of F4 with 230 mg, according to the protocol developed by Queiroz et al.[95]. This step permitted to isolate and identify: 2′-O-acetylmangiferin (**4**) (1.6 mg), 2-C-β-glucofuranosylmangiferin (**5**) (0.2 mg), isovitexin (**6**) (0.5 mg), 2-C-α-glucofuranosylmangiferin (**7**) (0.1 mg), 6′-O-acetylmangiferin (**8**) (2.7 mg), genkwanin-5-O-β-primeveroside (**9**) (2.1 mg) and genkwanin-O-β-glucoside (**10**) (0.1 mg) (see Fig. 5a for the structure of **6**, **9** and **10**, and Supplementary Fig. 6b for the structure of **4**, **5**, **7**, and **8**).

**Fraction preparation and dosage for in vivo testing.** The MPLC MG-free fractions were grouped according to their weight proportion: fractions M1 to M14 were grouped to prepare the fraction labelled F2, fractions M15 to M18 constituted the fraction F3 and fractions M46 to M56 constituted the fraction F4 after further depletion in MG of M46 to M48 as described above. Then, mice dosages were estimated by including the fractionation yield of grouped fractions, corrected by the yield of MPLC (83%) and VLC (50%), which were considered to define a proportional dosage, considering each step of the fractionation process. Thus, fraction F2 represented 13.7% of PN extract, fraction F3, 9.8% and F4, 10.3%, and mice doses were adapted proportionally. Eventually, to verify the fractionation process, the four fractions (F1 to F4) were proportionally pooled and analyzed by UHPLC-CAD, which demonstrated the similarity between the decoction and the reconstituted extract (Supplementary Fig. 6f).

To test in vivo the compounds identified in the F4 fraction (GE, AP) and their potential metabolites biotransformed in the digestive system after ingestion, standard substances of IX, GG, GE, and AP were purchased. Only GP could not be obtained, and the assay was performed with the fraction M50 of the first MPLC, which contained mainly this constituent. To estimate the proportion of these compounds, chemical profiling obtained by UHPLC-PDA-QDA-MS-ELSD was employed. ELSD peak areas were used to estimate the proportions of IX (4%) and AP (4%), GP (20%), GG (4%), and GE (4%). Mice doses were adjusted proportionally.

**Characterization of isolated constituents.** All isolated constituents (**1**–**10**) were analyzed to determine their chemical structure. NMR spectra were recorded on a Bruker Avance III HD 600 MHz NMR spectrometer equipped with a CQI 5 mm Cryoprobe and a SampleJet automated sample changer (Bruker BioSpin). Chemical shift were reported in parts per million (Δppm) using the deuterated dimethyl sulfoxide (DMSO-$d_6$) signal ($\delta_H$ 2.50; $\delta_C$ 39.5) as internal standards for $^1$H and $^{13}$C NMR, respectively, and coupling constants ($J$) were reported in hertz. Assignments were obtained based on two-dimensional (2D) NMR experiments (COSY, NOESY, HSQC, and HMBC). Accurate masses were measured with the UHPLC-HRMS/MS system described above in the following conditions: samples were injected (1 μL) into an Acquity UPLC BEH C18 column (2.1 × 100 mm i.d. 1.7 μm, Waters, Milford, MA, USA) and eluted (0.6 mL/min, column temperature set at 40 °C) with water (A) and acetonitrile (B) both containing 0.1% formic acid with the following gradient: from 5 to 100% of B from 0 to 7 min (curve 6), an isocratic step at 100% for 1 min and a re-equilibration step of 1.9 min. The full MS analyses were performed in negative mode with a mass range of 150−1800 at a resolution of 35000 full width at half maximum (FWHM) (at $m/z$ 200). The optimized HESI-II parameters were the following: source voltage: 2.5 kV (neg); sheath gas flow rate (N2): 55

units, auxiliary gas flow rate: 15 units, spare gas flow rate: 3; capillary temperature: 450 °C; S-Lens RF Level: 45.

Optical rotations were measured for the constituents 4, 5, and 7, and were obtained in a 10 cm cell on a Jasco polarimeter (Easton, USA). UV spectra of 4, 5, and 7 were acquired on a Hach UV–vis DR/4000 instrument (Loveland, CO, USA).

**Description of the isolated compounds.** The structures are provided in Source Data 2 (SD2), in Fig. 5 for compounds 6, 9, and 10, and in Supplementary Fig. 6b for compounds 4, 5, 7, and 8.

The identification of the following known compounds was supported by NMR spectral data comparison with those of literature: iriflophenone 3-C-β-glucoside (1)[96], mangiferin (2)[97], iriflophenone 2-O-α-rhamnoside (3)[96], isovitexin (6)[35], 6′-O-acetylmangiferin (8)[33], genkwanin 5-O-β-primeveroside (9)[34], and genkwanin-5-O-β-glucoside (10)[36].

An additional acetylmangiferin, 2′-O-acetylmangiferin (4) was identified, which was never reported to our knowledge. The position of the acetate was defined thanks to correlations from the carbonyl at $\delta_C$ 168.5 to the methyl at $\delta_H$ 1.73 and to the methine H-2′ of the glucose at $\delta_H$ 5.45 observed on the 2D HMBC spectrum. Furthermore, two additional isomers of MG were described for the first time to our knowledge. They were identified as 2-C-β-glucofuranosylmangiferin (5) and 2-C-α-glucofuranosylmangiferin (7). The presence of glucofuranosyl moieties in 5 and 7 instead of a glucopyranosyl as in MG (2) were proved by the low field $^{13}$C NMR chemical shift of C-4′ ($\delta_C$ 79.9 and 80.3 for β and α, respectively) and the high field $^{13}$C NMR chemical shift of C-5′ ($\delta_C$ 68.4 for β and α) indicating a free hydroxyl group in C-5′ and an ether in C-4′. The ROESY correlation between H-1′ and H-4′ in compound 5 indicated the α configuration of H-1′ whereas no correlation was observed between these two protons in 7.

***Iriflophenone 3-C-β-glucoside*** (**1**). Amorphous solid; $^1$H NMR (DMSO-d6, 600 MHz) δ 3.20 (3H, m, H-3′, H-4′, H-5′), 3.49 (1H, dd, J = 11.3, 4.3 Hz, H-6′b), 3.58 (1H, t, J = 9.5 Hz, H-2′), 3.62 (1H, d, J = 11.3 Hz, H-6′a), 4.59 (1H, d, J = 9.5 Hz, H-1′), 5.95 (1H, s, H-5), 6.78 (2H, d, J = 8.7 Hz, H-10, H-12), 7.56 (2H, d, J = 8.7 Hz, H-9, H-13); $^{13}$C NMR (DMSO-d6, 151 MHz) δ 60.5 (C-6′), 69.7 (C-4′), 71.9 (C-2′), 74.7 (C-1′), 78.3 (C-3′), 81.1 (C-5′), 94.8 (C-5), 103.7 (C-3), 107.0 (C-1), 114.6 (C-10, C-12), 130.8 (C-8), 131.5 (C-9, C-13), 156.7 (C-6), 157.4 (C-2), 159.0 (C-4), 161.4 (C-11), 194.6 (C-7). (SD 2 Supplementary Fig. 2.1−7 HRESIMS $m/z$ 407.0979 [M-H]$^-$ (calculated for $C_{19}H_{19}O_{10}$, 407.0978, Δppm 0.3). (SD 2 Supplementary Fig. 2.8-9) Observed UV max in UHPLC-PDA: 294 nm. SMILES: C1 = CC(O) = C(C(C2 = CC = C(O)C = C2) = O)C(O) = C1[C@H]3[C@H](O)[C@@H](O)[C@H](O)[C@@H](CO)C3. InChIKey: BQRGILCJOGVSOP-ACTWTYFVSA-N[96]. https://gnps.ucsd.edu/ProteoSAFe/gnpslibraryspectrum.jsp?SpectrumID=CCMSLIB00015641320#%7B%7D.

***Mangiferin*** (**2**) (MG). Clear yellow crystalline solid; $^1$H NMR (DMSO-$d_6$, 600 MHz) δ 3.12 (1H, m, H-4′), 3.16 (1H, m, H-5′), 3.19 (1H, m, H-3′), 3.39 (1H, dd, J = 12.4, 5.1 Hz, H-6′b), 3.68 (2H, d, J = 12.4, H-6′a), 4.04 (1H, t, J = 9.8 Hz, H-2′), 4.47 (1H, t, J = 6.2, 5.3 Hz, 6′OH), 4.59 (1H, d, J = 9.8 Hz, H-1′), 6.36 (1H, s, H-4), 6.86 (1H, s, H-5), 7.38 (1H, s, H-8), 13.75 (1H, s, 1OH); $^{13}$C NMR (DMSO-$d_6$, 151 MHz) δ 61.5 (C-6′), 70.2 (C-2′), 70.6 (C-4′), 73.1 (C-1′), 79.0 (C-3′), 81.6 (C-5′), 93.3 (C-4), 101.3 (C-9a), 102.6 (C-5), 107.6 (C-2), 108.1 (C-8), 111.7 (C-8a), 143.7 (C-7), 150.8 (C-10a), 154.0 (C-6), 156.2 (C-4a), 161.8 (C-1), 163.8 (C-3), 168.9, 179.1 (C-9). (SD2 Supplementary Fig. 2.10−16). HRESIMS $m/z$ 421.0776 [M-H]$^-$ (calculated for $C_{19}H_{17}O_{11}$, 421.0771, Δppm 1.2). $m/z$ 423.0925 [M + H]$^+$ (calculated for $C_{19}H_{19}O_{11}$ 423.0922 Δppm 0.7) (Supplementary Fig. 2.17−18). Observed UV max in UHPLC-PDA: 258, 318, 365 nm. SMILES: O = C1C2 = C(C = C(O)C([C@H]3[C@H](O)[C@@H](O)[C@H](O)[C@@H](CO)O3) = C2O)OC4 = CC(O) = C(O)C = C41. InChIKey: AEDDIBAIWPIIBD-ZJKJAXBQSA-N[97]. https://gnps.ucsd.edu/ProteoSAFe/gnpslibraryspectrum.jsp?SpectrumID=CCMSLIB00015641321#%7B%7D.

***Iriflophenone 2-O-α-rhamnoside*** (**3**). Amorphous solid: $^1$H NMR (DMSO-$d_6$, 600 MHz) δ 1.04 (3H, d, J = 6.2 Hz, H-6′), 3.07 (1H, d, J = 9.5 Hz,

H-3′), 3.13 (1H, t, $J$ = 9.5 Hz, H-4′), 3.27 (1H, m, H-5′), 3.32 (1H, m, H-2′), 5.10 (1H, d, $J$ = 1.7 Hz, H-1′), 6.03 (1H, d, $J$ = 2.0 Hz, H-5), 6.13 (1H, d, $J$ = 2.0 Hz, H-3), 6.79 (2H, d, $J$ = 8.5 Hz, H-10, H-12), 7.54 (2H, d, $J$ = 8.5 Hz, H-9, H-13); [13]C NMR (DMSO-$d_6$, 151 MHz) δ 17.7 (C-6′), 69.3 (C-5′), 69.8 (C-2′, C-3′), 71.3 (C-4′), 98.7 (C-1′), 109.3 (C-1), 114.8 (C-10, C-12), 131.2 (C-9, C-13), 155.4 (C-2), 156.2 (C-6), 159.2 (C-4), 161.5 (C-11), 192.5 (C-7). HRESIMS $m/z$ 391.1033 [M-H]$^-$ (calculated for $C_{19}H_{19}O_9$, 391.1029, Δppm 0.9). (SD2 Supplementay Fig. 2.19–24). Observed UV max in UHPLC-PDA: 291 nm.), $m/z$ 465.1029 [M + H]$^+$ (calculated for $C_{21}H_{21}O_{12}$ 465.1028 Δppm 0.3) (SD2 Supplementary Fig. 2.25-26) Smiles: OC1 = CC(O) = C(C(C2 = CC = C(O)C = C2) = O)C(O[C@@H]3 O[C@@H](C)[C@H](O)[C@@H](O)[C@H]3 O) = C1. InChIKey: BDUFDLBIUJUAJE-KONKAKAUSA-N[96]. https://gnps.ucsd.edu/ProteoSAFe/gnpslibraryspectrum.jsp?SpectrumID=CCMSLIB00015641322#%7B%7D.

**2′-O-Acetylmangiferin** (**4**). Amorphous solid; [α]$^{20}_D$ + 81 ( ± 4 (SD, $n$ = 5)), c 0.075, MeOH), UV λ$_{max}$ (MeOH) (log ε) 241 (4.27), 256 (4.22), 314 (3.91) 367 (3.92) nm (SD 2 Supplementary Fig. 2.35); [1]H NMR (DMSO-$d_6$, 600 MHz, T = 343 K) δ 1.73 (3H, s, H-2′b), 3.28 (1H, m, H-5′), 3.31 (1H, t, $J$ = 9.0 Hz, H-4′), 3.46 (1H, d, $J$ = 8.7 Hz, H-3′), 3.51 (1H, m, H-6′b), 3.73 (1H, m, H-6′a), 4.83 (1H, d, $J$ = 10.0 Hz, H-1′), 5.45 (1H, t, $J$ = 9.6 Hz, H-2′), 6.33 (1H, s, H-4), 6.83 (1H, s, H-5), 7.39 (1H, s, H-8), 13.73 (1H, s, 1OH); [13]C NMR (DMSO-$d_6$, 151 MHz, T = 343 K) δ 20.2 (C-2′b), 61.0 (C-6′), 70.3 (C-4′), 70.5 (C-1′), 72.2 (C-2′), 75.9 (C-3′), 81.2 (C-5′), 93.1 (C-4), 100.8 (C-9a), 102.3 (C-5), 105.4 (C-2), 107.8 (C-8), 111.4 (C-8a), 143.6 (C-7), 150.7 (C-10a), 154.1 (C-6), 156.2 (C-4a), 161.3 (C-1), 163.5 (C-3), 168.5 (C-2′a), 178.7 (C-9). (SD2 Supplementary Fig. 2.27–32). HRESIMS $m/z$ 463.0902 [M-H]$^-$ (calculated for $C_{21}H_{19}O_{12}$, 463.0877, Δppm 5.4), $m/z$ 465.1029 [M + H]$^+$ (calculated for $C_{21}H_{21}O_{12}$ 465.1028 Δppm 0.3) (Supplementary Fig. 2.33–34). SMILES: O = C1C2 = C(C = C(O)C([C@@H]3 O[C@H](CO)[C@@H](O)[C@H](O)[C@H]3OC(C) = O) = C2O)OC4 = CC(O) = C(O)C = C41. InChIKey: ZWQDCGBIYCPHCX-DPEQZQOSSA-N. https://gnps.ucsd.edu/ProteoSAFe/gnpslibraryspectrum.jsp?SpectrumID=CCMSLIB00015641323#%7B%7D.

**2-C-β-Glucofuranosylmangiferin** (**5**). Amorphous solid; [α]$^{20}_D$ + 146 ( ± 16 (SD, $n$ = 5)), c 0.010, MeOH), UV λ$_{max}$ (MeOH) (log ε) 241 (4.24), 257 (4.26), 316 (3.98) 365 (3.85) nm (SD2 Supplementary Fig. 2.44); [1]H NMR (DMSO-$d_6$, 600 MHz) δ 3.39 (0H, dd, $J$ = 11.3, 5.9 Hz, H-6′b), 3.56 (1H, dd, $J$ = 11.3, 3.0 Hz, H-6′a), 3.70 (1H, dd, $J$ = 8.6, 2.6 Hz, H-4′), 3.77 (1H, ddd, $J$ = 8.76, 5.9, 3.0 Hz, H-5′), 4.03 (1H, d, $J$ = 2.6 Hz, H-3′), 4.06 (1H, t, $J$ = 3.5 Hz, H-2′), 5.07 (1H, d, $J$ = 3.5 Hz, H-1′), 6.29 (1H, d, $J$ = 7.1 Hz, H-4), 6.84 (1H, s, H-5), 7.37 (1H, s, H-8), 13.95 (1H, s, 1OH); [13]C NMR (DMSO-$d_6$, 151 MHz) δ 63.6 (C-6′), 68.4 (C-5′), 76.8 (C-3′), 79.3 (C-1′), 79.9 (C-4′), 81.1 (C-2′), 94.2 (C-4), 100.5 (C-9a), 102.4 (C-5), 106.8 (C-2), 107.7 (C-8), 111.3 (C-8a), 143.5 (C-7), 150.6 (C-6), 153.9 (C-10a), 160.1 (C-1), 163.2 (C-3), 178.6 (C-9) (SD2 Supplementary Fig. 2.36–41). HRESIMS $m/z$ 421.0791 [M-H]$^-$ (calculated for $C_{19}H_{17}O_{11}$, 421.0771, Δppm 4.8)., $m/z$ 423.0921 [M + H]$^+$ (calculated for $C_{19}H_{19}O_{11}$ 423.0922 Δppm −0.3) (SD2 Supplementary Fig. 2.42–43). SMILES: O = C1C2 = C(C = C(O)C([C@H]3[C@H](O)[C@@H](O)[C@H](O3)[C@H](CO)O) = C2O)OC4 = CC(O) = C(O)C = C41. InChIKey: ULVKMRNLBMAPTI-BBNOFYSXSA-N. https://gnps.ucsd.edu/ProteoSAFe/gnpslibraryspectrum.jsp?SpectrumID=CCMSLIB00015641324#%7B%7D.

**Isovitexin** (**6**) (IX). Amorphous solid; [1]H NMR (DMSO-$d_6$, 600 MHz) δ 3.12 (1H, t, $J$ = 9.0 Hz, H-4″), 3.16 (1H, m, H-5″), 3.20 (1H, t, $J$ = 8.7 Hz, H-3″), 3.41 (1H, m, H-6′b), 3.68 (1H, m, H-6′a), 4.03 (1H, t, $J$ = 8.9 Hz, H-2″), 4.58 (1H, d, $J$ = 9.7 Hz, H-1″), 6.51 (1H, s, H-8), 6.79 (1H, s, H-3), 6.93 (2H, d, $J$ = 8.8 Hz, H-3′, H-5′), 7.93 (2H, d, $J$ = 8.8 Hz, H-2′, H-6′), 13.55 (1H, s, 5OH); [13]C NMR (DMSO-$d_6$, 151 MHz) δ 61.5 (C-6″), 70.2 (C-2″), 70.6 (C-4″), 73.0 (C-1″), 78.9 (C-3″), 81.6 (C-5″), 103.4 (C-10), 108.9 (C-6), 116.0 (C-3′, C-5′), 128.5 (C-2′, C-6′), 156.2 (C-9), 160.7 (C-5), 161.2 (C-4′), 163.4 (C-7), 163.5 (C-2), 182.0 (C-4). (SD2 2 Supplementary Fig. 2.45–51). HRESIMS $m/z$ 431.0999 [M-H]$^-$ (calculated for $C_{21}H_{19}O_{10}$, 431.0978, Δppm 4.8), $m/z$ 433.1129 [M + H]$^+$ (calculated for $C_{21}H_{21}O_{10}$ 433.1129 Δppm −0.1) (SD2 Supplementary Fig. 2.52–53).Observed UV max in UHPLC-PDA: 259,

323 nm. SMILES: O = C1C = C(C2 = CC = C(O)C = C2)OC3 = CC(O) = C([C@@H]4 O[C@@H](CO)[C@H](O)[C@@H](O)[C@@H]4 O)C(O) = C31. InChIKey: MYXNWGACZJSMBT-UBFQGWSASA-N[35]. https://gnps.ucsd.edu/ProteoSAFe/gnpslibraryspectrum.jsp?SpectrumID=CCMSLIB00015641326#%7B%7D.

**2-C-α-Glucofuranosylmangiferin** (**7**). Amorphous solid; [α]$^{20}_D$ + 232 ( ± 12 (SD, $n$ = 5), c 0.005, MeOH), UV λ$_{max}$ (MeOH) (log ε) 241 (4.50), 257 (4.50), 316 (4.26) 365 (4.10) nm; (SD2 Supplementary Fig. 2.63) 1H NMR (DMSO-d6, 600 MHz) δ 3.48 (1H, dd, J = 10.9, 5.2 Hz, H-6′b), 3.59 (1H, m, H-6′a), 3.78 (1H, m, H-5′), 4.16 (2H, m, H-3′, H-4′), 4.20 (1H, t, J = 1.8, 1.4 Hz, H-2′), 4.52 (1H, t, J = 5.7 Hz, 6′OH), 4.65 (1H, d, J = 6.0 Hz, 5′OH), 5.33 (1H, d, J = 3.7 Hz, 3′OH), 5.51 (1H, d, J = 2.9 Hz, H-1′), 6.30 (1H, s, H-4), 6.86 (1H, s, H-5), 7.38 (1H, s, H-8), 13.72 (1H, s, 1OH); 13 C NMR (DMSO-d6, 151 MHz) δ 63.8 (C-6′), 68.4 (C-5′), 74.9 (C-3′), 77.1 (C-2′), 79.3 (C-1′), 80.3 (C-4′), 94.2 (C-4), 101.0 (C-9a), 102.6 (C-5), 108.0 (C-8), 111.7 (C-8a), 143.8 (C-7), 150.9 (C-10a), 154.1 (C-6), 156.1 (C-4a), 159.2 (C-1), 164.5 (C-3), 179.0 (C-9). (SD2 Supplementary Fig. 2.54–60). HRESIMS $m/z$ 421.0792 [M-H]$^-$ (calculated for $C_{19}H_{17}O_{11}$, 421.0771, Δppm 4.9). $m/z$ 423.0918 [M + H]$^+$ (calculated for $C_{19}H_{19}O_{11}$ 423.0922 Δppm −1.0) (Supplementary Fig. 2.61–62). SMILES: O = C1C2 = C(C = C(O)C([C@@H]3[C@H](O)[C@@H](O)[C@H](O3)[C@H](CO)O) = C2O)OC4 = CC(O) = C(O)C = C41. InChIKey: ULVKMRNLBMAPTI-QRDWVMTRSA-N. https://gnps.ucsd.edu/ProteoSAFe/gnpslibraryspectrum.jsp?SpectrumID=CCMSLIB00015641325#%7B%7D.

**6′-O-Acetlymangiferin** (**8**). Amorphous solid; [1]H NMR (DMSO-$d_6$, 600 MHz) δ 1.98 (3H, s, H-6′b), 3.16 (1H, t, $J$ = 8.7 Hz, H-4′), 3.21 (1H, t, $J$ = 8.7 Hz, H-3′), 3.37 (1H, m, H-5′), 3.90 (1H, dd, $J$ = 12.0, 7.0 Hz, H-6′b), 4.09 (1H, t, $J$ = 9.9, 8.7 Hz, H-2′), 4.35 (1H, dd, $J$ = 12.0, 1.8 Hz, H-6′a), 4.59 (1H, d, $J$ = 9.8 Hz, H-1′), 6.37 (1H, s, H-4), 6.86 (1H, s, H-5), 7.37 (1H, s, H-8), 13.78 (1H, s, 1OH); [13]C NMR (DMSO-$d_6$, 151 MHz) δ 20.8 (C-6′b), 64.6 (C-6′), 69.9 (C-2′), 70.4 (C-4′), 73.1 (C-1′), 78.1 (C-5′), 78.7 (C-3′), 93.2 (C-4), 101.3 (C-9a), 102.6 (C-5), 107.3 (C-2), 108.0 (C-8), 111.6 (C-8a), 143.8 (C-7), 150.8 (C-10a), 154.1 (C-6), 156.2 (C-4a), 161.9 (C-1), 163.8 (C-3), 170.4 (C-6′a), 179.1 (C-9). (SD2 Supplementary Fig. 2.64–70). HRESIMS $m/z$ 463.0900 [M-H]$^-$ (calculated for $C_{21}H_{19}O_{12}$, 463.0877, Δppm 5.0) (SD2 Supplementary Fig. 2.71–72). Observed UV max in UHPLC-PDA: 257, 317, 366 nm. SMILES: O = C1C2 = C(C = C(O)C([C@@H]3 O[C@H](COC(C) = O)[C@@H](O)[C@H](O)[C@H]3 O) = C2O)OC4 = CC(O) = C(O)C = C41. InChIKey: MJZHKTMNVNMAPP-RDZBXBSQSA-N[33]. https://gnps.ucsd.edu/ProteoSAFe/gnpslibraryspectrum.jsp?SpectrumID=CCMSLIB00015641327#%7B%7D.

**Genkwanin 5-O-β-primeveroside** (**9**) (GP) (synonym *yuankanin*): Amorphous solid; 1H NMR (DMSO-d6, 600 MHz) δ 2.98 (1H, td, J = 8.3, 4.8 Hz, H-2‴), 3.03 (1H, t, J = 10.8 Hz, H-5‴b), 3.10 (1H, td, J = 8.8, 4.6 Hz, H-3‴), 3.21 (1H, td, J = 9.1, 5.3 Hz, H-4″), 3.28 (1H, m, H-4‴), 3.29 (1H, m, H-3″), 3.36 (1H, m, H-2″), 3.57 (1H, ddd, J = 8.8, 6.5, 1.9 Hz, H-5″), 3.65 (1H, dd, J = 11.4, 6.6 Hz, H-6″b), 3.69 (1H, dd, J = 11.3, 5.4 Hz, H-5‴a), 3.90 (3H, s, 7OMe), 3.98 (1H, dd, J = 11.5, 1.9 Hz, H-6″a), 4.19 (1H, d, J = 7.6 Hz, H-1‴), 4.78 (1H, d, J = 7.6 Hz, H-1″), 4.87 (1H, d, J = 5.0 Hz, 2‴OH), 4.94 (2H, d, J = 4.8 Hz, 3‴OH, 4‴OH), 5.17 (1H, d, J = 5.6 Hz, 4″OH), 5.18 (1H, d, J = 6.0 Hz, 3″OH), 5.59 (1H, d, J = 2.0 Hz, 2″OH), 6.72 (1H, s, H-3), 6.88 (1H, d, J = 2.4 Hz, H-6), 6.92 (2H, d, J = 8.7 Hz, H-3′, H-5′), 7.04 (1H, d, J = 2.4 Hz, H-8), 7.93 (2H, d, J = 8.8 Hz, H-2′, H-6′), 10.28 (1H, s, 4′OH); 13 C NMR (DMSO-d6, 151 MHz) δ 56.1 (C-7OMe), 65.7 (C-5‴), 68.7 (C-6″), 69.5 (C-4‴), 69.8 (C-4″), 73.4 (C-2‴), 73.5 (C-2″), 75.6 (C-3‴), 75.9 (C-5″), 76.6 (C-3″), 96.7 (C-8), 103.0 (C-6), 103.8 (C-1″), 104.1 (C-1‴), 105.8 (C-3), 109.2 (C-10), 115.9 (C-3′, C-5′), 121.1 (C-1′), 128.2 (C-2′, C-6′), 158.1 (C-5), 158.4 (C-9), 160.9 (C-4′), 161.3 (C-2), 163.6 (C-7), 176.9 (C-4). (SD2 Supplementary Fig. 2.73–79). HRESIMS $m/z$ 577.1585 [M-H]$^-$ (calculated for $C_{27}H_{29}O_{14}$, 577.1557, Δppm 4.8), $m/z$ 579.1696 [M + H]$^+$ (calculated for $C_{27}H_{31}O_{14}$ 579.1708 Δppm −2.2) (SD2 Supplementary Fig. 2.80–81). Observed UV max in UHPLC-PDA: 260, 330 nm SMILES: O[C@H]([C@H]([C@@H]([C@H](O1)CO[C@H]2[C@H](O)[C@@H](O)[C@H](O)CO2)O)O)[C@@H]1OC3 = CC(OC) = CC(OC(C4 = CC = C(C = C4)O) = C5) = C3C5 = O.

InChIKey: ZKIXACXWZQFVAB-MUCJXJSVSA-N[34]. https://gnps.ucsd.edu/ProteoSAFe/gnpslibraryspectrum.jsp?SpectrumID=CCMSLIB0001564 1328#%7B%7D.

***Genkwanin 5-O-β-glucoside*** (**10**) (GG), (synonym *glucogenkwanin*): Amorphous solid; $^1$H NMR (DMSO-$d_6$, 600 MHz) δ 3.15 (1H, m, H-4″), 3.28 (1H, m, H-3″), 3.35 (1H, m, H-2″), 3.36 (1H, m, H-5″), 3.50 (1H, dt, $J$ = 12.1, 6.1 Hz, H-6″b), 3.90 (3H, s, 7OMe), 4.71 (1H, t, $J$ = 6.0 Hz, 6″OH), 4.75 (1H, d, $J$ = 7.7 Hz, H-1″), 5.07 (1H, d, $J$ = 5.4 Hz, 4″OH), 5.12 (1H, d, $J$ = 5.2 Hz, 3″OH), 5.61 (1H, d, $J$ = 1.8 Hz, 2″OH), 6.73 (1H, s, H-3), 6.91 (1H, d, $J$ = 2.4 Hz, H-6), 6.92 (2H, d, $J$ = 8.8 Hz, H-3′, H-5′), 7.07 (1H, d, $J$ = 2.4 Hz, H-8), 7.94 (2H, d, $J$ = 8.8 Hz, H-2′, H-6′), 10.29 (1H, s, 4′OH); $^1$H NMR (DMSO-$d_6$, 151 MHz) δ 55.8 (C-7OMe), 60.7 (C-6″), 69.7 (C-4″), 73.4 (C-2″), 75.5 (C-3″), 77.4 (C-5″), 96.4 (C-8), 103.3 (C-6), 104.0 (C-1″), 105.5 (C-3), 108.9 (C-10), 115.7 (C-3′), 120.8 (C-1′), 128.0 (C-2′), 157.9 (C-5), 158.1 (C-9), 160.5 (C-4′), 161.1 (C-2), 163.2 (C-7). (SD2 Supplementary Fig. 2.82–87). HRESIMS *m/z* 445.1155 [M-H]$^-$ (calculated for $C_{22}H_{21}O_{10}$, 445.1135, Δppm 4.5). *m/z* 447.1272 [M + H]$^+$ (calculated for $C_{22}H_{23}O_{10}$ 447.1286 Δppm −3.1) (SD2 Supplementary Fig. 2.88–89). Observed UV max in UHPLC-PDA: 262, 327 nm. SMILES: COC1 = CC(O[C@H]2[C@@H]([C@H]([C@@H]([C@H](O2)CO)O)O)O) = C3C(C = C(OC3 = C1)C(C = C4) = CC = C4O) = O. InChI-Key: QLZMOQILAYMPIF-MIUGBVLSSA-N[36]. https://gnps.ucsd.edu/ProteoSAFe/gnpslibraryspectrum.jsp?SpectrumID=CCMSLIB00015641329#%7B%7D

**Prediction of the biotransformation of Genkwanin (GE).** The aglycone GE is expected to be the common metabolite of the glycosides GG and GP after ingestion, resulting from small intestinal biotransformation[37]. The further biotransformation of GE was also explored, using the BioTransformer 3.0 open-access software. The software predicted 6 metabolites resulting from phase 1, phase 2 and human gut microbial biotransformation[35]. Among the predicted metabolites, rhamnocitrin, apigenin, hydroxygenkwanin, and tectochrysin were tested in vitro on immortalized brown adipocytes to assess their potential to stimulate glucose uptake. Furthermore, as the gut biotransformation of apigenin was previously described[39], three additional metabolites were also tested.

### Reporting summary

Further information on research design is available in the Nature Portfolio Reporting Summary linked to this article.

## Data availability

All source data generated for the panels in main and Supplementary Figs. are available in this paper (Source Data 1). All images quantified in Fig. 3f are deposited at: https://zenodo.org/records/15231500. All source data related to bioactivity and phytochemistry are available in this manuscript or on repositories. Bio guided fractionation data generated in this study have been deposited in the YARETA repository under https://doi.org/10.26037/yareta:2j2b4d5ivjeudbcsbxmvxcpgiq. This data includes the chemical profiling of PNe and its four fractions obtained by UHPLC-PDA-HRMS/MS (raw data), the chemical profiling of fractions F2 and F4 obtained by UHPLC-PDA-QDA-MS-ELSD, and NMR spectra of the four fractions and all isolated constituents. UHPLC-HRMS/MS mzML data have been deposited in MassIve under https://doi.org/10.25345/C51G0J70D. The fragmentation spectra of all isolated compounds have been deposited in the GNPS library. The direct links to the spectra are provided in the "Methods" sections "Description of the isolated compounds". NMR, HRMS, and UV data generated in this study are also provided in this paper in Source Data 2. Source data are provided with this paper.

## Code availability

The Python code (Phyton 3.12.7) for the nuclear localization of ERK1/2 is deposited at GitHub (https://doi.org/10.5281/zenodo.15767222) or https://github.com/arcsi1989/nuclear_translocation.

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

## Acknowledgements

We thank Manuel Klug and Myrtha Arnold for experimental assistance with the hyperinsulinemic-euglycemic clamps. We thank Dr. Adhideb Ghosh for statistical advice. CH was supported by an SNF and Innosuisse BRIDGE Proof of Concept Fellowship (40B1-0_191463). Funding for collection and conservation of *Phaleria nisidai* leaves was provided by PAIR and Antenna Foundation. The Institute of Pharmaceutical Sciences of Western Switzerland of the University of Geneva (JLW) is thankful to the Swiss National Science Foundation for the support in the acquisition of the NMR 600 MHz (SNF R'Equip grant 316030_164095).

## Author contributions

C.H., J.H., C.W., E.F.Q., and J.L.W. conceptualized the study and wrote the manuscript. C.K., B.G., L.M., and A.R., wrote and revised the manuscript. CH designed, performed and analyzed in vitro and in vivo studies. C.M. assisted with glucose clamps. A.K. designed, performed and analyzed in vitro experiments. A.K. revised the manuscript. H.D., I.R., and V.E. contributed to in vitro experiments. L.B. and M.B. generated the hMADS data. A.H. analyzed the immunofluorescence images. J.H. performed extraction, fractionation, and purification of PN extract under supervision of E.F.Q. and J.L.W. L.M. performed NMR analyses. A.R. performed UHPLC-HRMS/MS analyses. C.K., B.G., and V.Y. represent the

Republic of Palau, provided the genetic resources and shared traditional knowledge.

## Funding

## Competing interests
The authors declare no competing interests.

## Additional information

[1]Institute of Food, Nutrition and Health, ETH Zurich, Schwerzenbach, Switzerland. [2]School of Pharmaceutical Sciences, University of Geneva, CMU, Geneva, Switzerland. [3]Institute of Pharmaceutical Sciences of Western Switzerland, University of Geneva, CMU, Geneva, Switzerland. [4]Biomedical Research Center, Slovak Academy of Sciences, Bratislava, Slovakia. [5]Department of Animal Physiology and Ethology, Faculty of Natural Sciences, Comenius University, Bratislava, Slovakia. [6]Institute for Stem Cell Biology and Regenerative Medicine, Stanford University School of Medicine, Stanford, CA, USA. [7]Institute of Biomechanics, ETH Zurich, Balgrist Campus, Zürich, Switzerland. [8]Pacific Academic Institute for Research, Koror, Palau. [9]Community Health Association – Geneva, Champvent, Switzerland. [10]Nanyang Technical University (NTU), Singapore, Singapore. [11]These authors contributed equally: Carla Horvath, Joëlle Houriet, Alexandra Kellenberger. ✉e-mail: jean-luc.wolfender@unige.ch; christian.wolfrum@ntu.edu.sg

