## [Peer Review File · Nature Communications]

Genkwanin glycosides are major active compounds in *Phaleria nisidai* extract mediating improved glucose homeostasis by stimulating glucose uptake into adipose tissues

Corresponding Author: Professor Christian Wolfrum

Version 0:

Reviewer comments:

Reviewer #1

(Remarks to the Author)

This manuscript suggests that Genkwanin glycosides from the *Phaleria nisidai* extract may improve glucose homeostasis by stimulating insulin-independent glucose uptake. It is huge amount of work and the authors used multiple approaches to support their findings in the paper, which should be appreciated. Although the paper is interesting, there are several issues that need attention, and there are some deficiencies in the evidence. According to the author, the innovation of this study lies in the successful completion of the reverse pharmacological method for insulin resistance for the first time. But the certain efficacy and proposed mode of action of PNe and GE glycosides remain to be investigated in humans. Besides that, the author mentioned the shortcomings of the study in line 393-408, but for the study of molecular mechanism, gene knockout experiment is necessary. Many deficiencies can and need to be improved.

In terms of experimental design and result presentation, the following problems may exist:

- 1) In Fig. 2i, phospho-sites which treated by PNe seem significantly lower than CTRL at insulin-stimulated conditions. How to draw this conclusion 'Both phospho-sites were not affected by PNe exposure at baseline or insulin-stimulated conditions'?
- 2) When calculating the expression of phosphorylated proteins (Fig. 2j-m), why use the value of AKTpS473or pT308/HSP90 instead of AKTpS473or pT308/AKT? More importantly, how the authors determine the way and days of drug administrations? Why treat with PNE for 3 days?
- 3) In Fig. 3a-c and 3d-e, what is the reason for the different duration of treatment with PNe? Is there a time when all the above changes can occur?
- 4) 'All investigated fractions improved insulin sensitivity (Fig. 4c-d) after 4 weeks of treatment, while only F4 effectively ameliorated insulin sensitivity after prolonged DI (Fig. 4e-f).' When is the specific time of prolonging DI?
- 5) There may be some errors in the significance symbol of Fig. 4j.
- 6) Why is the DI duration in Fig. 4k different from that in Fig. 4a?

Reviewer #2

(Remarks to the Author)

The manuscript "Genkwanin glycosides from the *Phaleria nisidai* extract improve glucose homeostasis by stimulating insulin-independent glucose uptake" studies the function of natural compounds isolated of *Phaleria nisidai* (PN). Obesity and diabetes have reached pandemic dimensions and novel therapies are urgently needed. The manuscript consists of two parts (a) the analysis of the effects and molecular mechanisms of PN extracts in adipocytes and mice; (b) the identification of genkwanin as one active compound present in PN.

Overall this is a very interesting paper with high translational relevance since it defines natural compounds that have antidiabetic actions. Nevertheless, the following points should be addressed before publication:

General comments:

- The abbreviations for *Phaleria nisidai* and its extract are confusing: PN versus PNe. PN is not clearly defined; PNe is defined but not always used as abbreviation.

- Fig. 2: 3T3L1 cells were used throughout the ms, but this cell model has several disadvantages. It would be important that primary adipocytes derived from SVF of BAT or WAT should be used for the major experiments. Especially since the authors observe no effect of insulin on glut4, which is puzzling given the previous studies showed a clear increase of Glut1 in primary adipocytes treated with insulin (Hajduch 1992). Importantly, human primary white adipocytes should be used to further corroborate the relevance of the findings in murine cells.
 - Glut1 and GLUT4 plasma membrane levels should be measured for PNe-treated cells. It was previously shown in adipocytes that insulin leads to translocation of GLUT1 from intracellular vesicles to the plasma membrane (Holman 1990). This could be another important mechanism of PNe action.
 - PNe increases glycolysis, does PNe also increase lipolysis?
 - The authors state (line 134): "We could translate these findings to our animal model system, where we detected higher GLUT1 but not GLUT4 protein content in the iWAT and iBAT of PNe-fed animals compared to HFD controls (Fig. 2t-v)." Pls rephrase and give % values and significance in the text, there is a clear increase in glut4 levels in PNe treated BAT (Fig 2v).
 - Fig. 2w-y: again there is a clear increase in glut4 in "brown" adipocytes; only n=3 experiments were done, the number of experiments should be increased; it would be very interesting to see the effects in human brown adipocytes.
 - Fig. 3b-c: erk signaling promotes adipogenesis (prusty 2002), the authors should investigate the effect of PNe on markers for adipogenesis.
 - Fig. 3e-f: in 3f PNe-treated cells are missing. Moreover, it is very difficult to see clear differences in the pictures shown (higher magnifications, insets might be helpful).
 - Fig. 6: a thorough analysis of the molecular mechanisms and pathways of GE in adipocytes (especially brown) is missing; especially Glut1 and 4 expression and membrane localization would be interesting. The quality of the western blots shown in fig. 6 (especially p and q) does not permit a clear quantification of Glut1 phosphorylation.
- Minor points:
- Fig. 2: It should be clearly stated in the text and/or legend of figure 2 which cells were used. Is fig. 2a missing? Only legends are shown.
 - line 132: "Figure 3x"? Should be 2x.

Reviewer #3

(Remarks to the Author)

In this study the authors aim to define the active compound of the Phaleria nisidai plant that is responsible for the glucose-lowering effects observed in Palaun traditional medicine concepts.

While this is per se a very interesting and promising study, and the authors invested a lot of work in conducting the presented experiments, I have serious concerns about the set-up of the animal studies and interpretation of the data based on the way they were presented.

A substantial revision of the manuscript is required in order to be considered for publication.

General comments:

All of the GTT and ITT data are presented as normalized to initial blood value.
For correct interpretation of the data, the presentation as raw data in mmol/l is crucial.

Please refer for correct presentation of these data and calculation of AUC to "GTTs and ITTs in mice: simple tests, complex answers" (Nature Metabolism volume 3, pages 883–886 (2021)).

This is specifically important, because there are significant differences between the RBG in the treatment groups. A lower initial blood glucose value (e.g., 8 mmol/l vs. 10 mmol/l) will show a higher percentage drop (25% vs. 20%) even if both groups decreased by the same value (e.g., 2 mmol/l).

No final conclusion about the effects of the compounds on glucose or insulin tolerance can be drawn at this point.

Also, show body weight development in g.

Further, the effects of PNe are described as insulin-independent, yet it is claimed that insulin sensitivity is increased. 'Increasing insulin sensitivity' means enhancing the action of insulin on insulin-sensitive tissues, which per se cannot be an 'insulin-independent' process.

Overall, the study is confusing, because multiple possible mechanisms are introduced for the plant fractions, that are not necessarily reflected in the PNe data. There is a lack of consistency.

- PNe decreases insulin levels and urinary glucose excretion, and increases glucose uptake into adipose tissues
- All fractions decrease random or fasting blood glucose levels, but PNe does not (Fig. 4b, 4j)
- F3 increases GLP1 levels. Is there an effect on insulin secretion?
- F4 increases glucose uptake into adipose tissues AND soleus muscle, there is no effect on basal glucose turnover
- GE decreases RBG (Fig. 5g) but PNe does not, GE increases urinary glucose excretion – PNe decreases it
- GE increases AKT phosphorylation in fat, PNe has no effect on AKT phosphorylation
- GE has also an effect on hepatic insulin sensitivity and muscle glucose uptake, but PNe does not

Detailed comments:

- No real conclusions can be made of the data set presented in Figure 1, because all of the tests were performed at completely different stages of the dietary interventions
- There is a huge difference between 6 and 12 weeks of HFD when it comes to the development of insulin resistance. The accumulation of lipids in tissues like liver and muscle differs significantly based on the length of the HFD.
- In order to compare the tests with each other, there should be a max. of 1-2 weeks in between the tests.
- ITT was performed after 12 weeks of HFD and 5 weeks of DI, but the GTT was performed after 6 weeks of HFD and 2 weeks of DI. Insulin levels should be measured in parallel to the GTT. These data are not relatable and thus cannot be interpreted.
- Figure 1m, insulin levels measured after 30 min. The methods describe a measurement after 15 min
- In order to draw conclusions from the tissue uptake data in Fig. 1o-t, blood insulin levels are needed

- Figure 2j: Why normalization to GAPDH? GAPDH not shown in blots. Since several glycolytic enzymes are upregulated, is GAPDH protein content different?

- GLUT4 mRNA content is higher in Fig. 2s and trends higher (p-val = ?) in Fig. 2r. Is there is a difference in GLUT4 translocation? GLUT4 translocation assays should be performed or GLUT4 membrane content measured.

- What are the expression levels of other enzymes involved in the glycolysis pathway?

- While AKT phosphorylation can facilitate GLUT4 translocation (line 120), there are AKT-independent mechanisms that promote GLUT4 translocation (T. McGraw, Mol Biol Cell. 2006 Oct;17(10):4484-93. So that's not necessarily an exclusion criterion

- Line 189, Figure 4b: 'All fractions lowered RBG' is too generalized. It really depends on the time point of the intervention and the fraction

- How do the authors explain, that over a time period of intervention the control mice do not show an increase in RBG (Fig. 4b) although there is a constant weight gain? 8 mmol/l seems low for fed blood glucose levels on HFD

- Why are the fasting blood glucose levels in Figure 4j similar to or even higher than RBG in Fig. 4b?

- Figure 4k displays PNe as an experimental group. Were insulin clamps performed in PNe treated group in comparison? 4k says 6 weeks of HFD, line 200, however, mentions 12 weeks of HFD. Which one is it?

- Is the effect of F4 on glucose turnover in Fig. 4m statistically significant? The basal glucose turnover rate is not increased and PNe does not improve fasting blood glucose levels (Fig. 4j). How do you explain that in context of the proposed 'insulin-independent GLUT1 mediated glucose uptake' mechanism? Similar in Figure 7f

- Figure 4r. Because of the glucose-6-phosphatase activity in liver, 14C-2-DG-P content in liver is not representative of liver glucose uptake. See also Figures 1q, 7k
- F4 and GE both increase glucose uptake into soleus muscle. A muscle insulin-dependent mechanism should be considered. Were muscle tissues isolated for signaling analysis? Or muscle cell lines treated with the compounds?

- Figure 5, iWAT shows increased glucose uptake, but no increase in GLUT1. A GLUT4 translocation-mediated mechanism should be considered here. Were insulin levels measured in these mice? The increase in AKT phosphorylation might be due to differences in insulin levels and thus differences in membrane GLUT4.

Methods:

- Why were mice gavaged with glucose by BW for glucose-stimulated insulin secretion tests, but were give a fixed glucose dose for GLP-1 measurements?
- In order to interpret the tissue-specific uptake data for PN or GE, blood insulin levels are needed.

Minor comments:

- Check Figure references, e.g., line 132, line 262 Fig. 1u doesn't exist
- Define PD-98059 at first mentioning, line 161
- Since the mice received the compounds in the food, the dosage per BW can only be an estimation. It would more correct, to give the concentration of PNe (or compounds) in food and then what daily dose it would correspond to based on daily food intake.

Version 1:

Reviewer comments:

Reviewer #1

(Remarks to the Author)

This study demonstrates that Genkwanin glycosides, isolated from *Phaleria nisidai* extract, exhibit potential therapeutic effects on glucose homeostasis through the enhancement of insulin-independent glucose uptake mechanisms. The manuscript presents comprehensive experimental data and employs rigorous methodological approaches to support its findings. Following thorough revisions that have addressed previous limitations, the study now provides a well-substantiated scientific contribution that warrants publication.

Reviewer #2

(Remarks to the Author)

The authors have addressed all my questions. They performed an impressive amount of experiments and the new data really improved the manuscript.

Reviewer #3

(Remarks to the Author)

In the revised version of their manuscript the authors performed several additional experiments and acknowledged the study's limitations in the experimental design.

Although the quality of the manuscript has improved, certain deficiencies still need to be addressed.

- a) The numerous phenotypes induced by the various fractions of PNe make the manuscript challenging to follow. It would benefit from a graphical abstract, table or a flow chart summarizing the effects of PNe and the corresponding fractions (e.g. GTT up, glucose in WAT uptake unchanged, AKT S473 phosphorylation up, etc.)
- b) Western blots displaying the membrane fractions of GLUT1/4 lack a membrane marker such as Na⁺/K⁺-ATPase
- c) Some figures don't show the number of individual experiments/mice as indicated in the legend (e.g. food intake in Suppl Fig. 7b F4 fraction)
- d) Multiple data sets show a p-value that is close to significance, but is not <0.05 (e.g. Suppl. Fig 8f, Fig. 5d). But data are described as significant differences in the main text
- e) PKC-ERK1/2-GLUT1 pathway was explored with PNe, but not with GE. Here the authors focused on AKT pathway only. Why?
- f) AKT-phosphorylation analysis was adapted based on reviewer's comment in Fig. 2, but not in Fig. 6. Further, why different AKT phosphorylation sites examined in fat vs. muscle (Suppl. Fig. 10)?
- g) Fig. 3 uses PMA, no description in main text
- h) minor: Suppl. Fig. 3 beige or brown fat?; Suppl. Fig. 2e/m glycolysis typo; line 186 typo in phospho-PKC

Reviewer #4

(Remarks to the Author)

This is a statistical review. I do not comment on the soundness of the science.

Although experimental conditions and statistical methods are reasonably well-documented in the figure legends, things could be improved. Below, I provide some comments about each figure's figure legend.

Figure 1's figure legend.

- In panels b, c, and d, the n/group is not provided. Please provide the exact n of each group, especially if the two groups have different sample sizes.
- In panels g-j, the sample size is given as a range, n=5-8 per group. That is too vague. Please say which group has n=5, which group has n=6, which group has n=7, and which group has n=8. If the n varies from panel to panel, please document that also.
- In panels l-m, it says n=7-8 per group. That is too vague. Please say which group has n=7 and which group has n=8.
- Please do likewise for panels n and o, i.e., please provide the exact n of each group.
- It is unclear from what the authors wrote which specific panels were analyzed using 2-way ANOVA and which specific panels were analyzed using repeated-measures 2-way ANOVA. Please clarify which procedures were used on which panels.
- In panel n, do the data points at Time = 30 minutes come from the same mice as the data points at Time = 0 minutes? If Yes, then (1) spaghetti plots would probably be better than dynamite plots for showing the data, and (2) this probably calls for repeated-measures 2-way ANOVA instead of ordinary 2-way ANOVA. The repeated-measures ANOVA should include a test of the Diet-x-Time Interaction along with tests comparing diet groups at each time point.

- In panels b, e, and l, it is not clear what is being analyzed using repeated-measures 2-way ANOVA. Are the authors testing the main effect of Diet (i.e., HFD vs PNe)? Are they testing the main effect of time? Are they testing the Diet-x-Time Interaction? Or are they merely testing the simple effects of Diet at each individual time point? Please add some clarity.
 - o However, the time curves of panel e are connected to the AOCs of panel f, and the time curves of panel l are connected to the AOCs of panel m. This means that the t-tests comparing AOCs between Diet groups is 95% equivalent to testing the main effect of Diet using repeated-measures 2-way ANOVA on the time curves, which in turn means there may be some redundancy in the analyses. This is why it is important to describe what tests and what comparisons are being conducted using repeated-measures 2-way ANOVA.

Figure 2's figure legend.

- In every location where the n/group is given as a range, please replace the range with the exact n of each group. Please do not say n=6/group unless both groups have exactly n=6. Please also clarify whether the statement "(n=6)" for panels c-g means n=6 for each group or total n=6 for both groups combined.
 - o Because Figure 2 shows in vitro experiments, it's not clear what the sample sizes represent. The "Quantification and statistical analysis" section of the Methods say that each experiment was independently reproduced 2-3 times, but Figure 2 looks like it shows results from only single reproductions of each experiment. Please elaborate.
- The legend entry for panels a-b, glucose uptake in basal and insulin-stimulated cells, says that panel b is 3T3-L1 cells and panel c is iBAs. This does not look right. Please confirm.
- Figure 2's legend says that Two-way ANOVA was applied, with Sidak's post hoc test on a number of panels, and with Dunnet's post-hoc test on panel's m and n. The Quantification and statistical analysis section says instead that One-way ANOVA with these post-hoc tests was used. Please rectify the discrepancy.
 - o Panels m and n show the structure of a one-factor experiment.
 - o Although panels a, b, d-g, and i-l show the structure of two-factor experiments, the fact that Sidak's method was used to compare all groups pairwise suggests strongly that the experiments were analyzed using a one-way layout.
- What do the beige and white colors mean in panels t and u?

Figure 3's figure legend.

- In panels e through g, please do not describe the n/group using ranges. Please provide the exact n of each and every group.
- Where it says that one-way ANOVA with Holm-Sidak's or Tukey's adjustments between all groups was applied in d, g, and f, please indicate which panels had Holm-Sidak's method applied to them versus which panels had Tukey's method applied to them.
 - o Please also explain why Holm-Sidak was chosen over regular Sidak, why Tukey was chosen over regular Sidak, and what motivated using Holm-Sidak sometimes and Tukey other times. Could p-hacking have been involved?

Figure 4's figure legend.

- For every panel, please do not describe the n/group using ranges. Please provide the exact n of each and every group.
- I get the impression that panel a defines the color-to-treatment mapping used in panels b through j, and that panel k defines the color-to-treatment mapping used in panels l through s. Please confirm if this is true.
 - o In panels c, e, and g, which line color and line pattern corresponds to the HFD control? Inserting a line legend inside each panel might help the reader.
- I do not understand the sentence, "Two-way ANOVA with Tukey's post-hoc test between all groups was applied in m-n (between groups)". In panel m, and likewise in panel n, I see 6 groups, but they are clustered into 3 basal groups and 3 insulin groups. And as far as I can tell, each panel must have had two separate Tukey's procedures applied to it, one to the basal cluster (3 groups) and one to the insulin cluster (3 groups). If that is what the authors did, then they will want to Sidak-adjust the significance level of each Tukey procedure if they want to truly control each panel's overall Type-I error rate to 5%.
- Please discuss further the mixed-effects analysis with Dunnet's post-hoc test compared to HFD that was applied in panel b. I see eight different HFD controls, one for each week of DI, and I see a total of $8 \times 5 = 40$ groups spread out over the 8 weeks. Did the authors perform one single Dunnet's test on all 40 groups combined? Or did the authors perform 8 separate Dunnet's tests grouped by week of DI? If the authors did the latter, then they will want to Sidak-adjust the significance level of each of their 8 Dunnet's tests if they want to truly control panel b's overall Type-I error rate to 5%.

Figure 5's figure legend.

- For panels c, d, g, h, and i, please do not describe the sample sizes per group using ranges. Please provide the exact n of each and every group.
- Regarding the repeated measures two-way ANOVA with Dunnett's post-hoc test that was applied in panels c and e, it appears that each panel had six separate Dunnett's tests grouped by time in minutes, each of which used the time group's HFD control as its time-specific reference group. If the authors want to truly control the Type-1 error per panel to 5% overall, then they will wish to Sidak-adjust the significance level of each Dunnett's test to control for the fact that they are doing six separate Dunnett's tests per panel.
- In panel 5i, the p-value bracket connects HFD to IX. I think it was meant to connect HFD to GG. Please check.

Figure 6's figure legend.

- In every place where the sample sizes are described as n=9-10 per group, please get rid of the ranges and please provide the exact n of each and every group.
- Regarding the western blots, I see panels n and o, but not panels p and q.

Figure 7's figure legend.

- In every place where the sample sizes are described either as n=8-9 per group or as n=7-9 per group, please get rid of the

ranges, and please name which groups have exactly n=7, which groups have exactly n=8, and which groups have exactly n=9.

• For panels f and g, it says (A) that 2-way ANOVA with Tukey's post-hoc test was applied, and (B) that all comparisons were performed between all groups. If both statements are true, then why in panel f is there no p-value bracket comparing basal HFD to insulin HFD, no p-value bracket comparing basal GE to insulin GE, and no p-value bracket comparing basal MET to insulin MET? Those pairwise differences clearly are larger than the differences between members of the insulin subgroup. Could it be that only some comparisons were performed between all groups? In other words, could it be that Tukey's procedure was applied twice in panel f, once to the basal subgroup and once to the insulin subgroup, rather than just once to all six groups? Please clarify.

Version 2:

Reviewer comments:

Reviewer #3

(Remarks to the Author)

The authors have very thoroughly addressed all of my comments by performing additional experiments and providing a detailed and critical discussion of the data. Congratulations on compiling such an impressive amount of data and bringing this comprehensive work to publication!

(Remarks on code availability)

Reviewer #4

(Remarks to the Author)

I think the authors successfully addressed my previous questions and comments.

I have a few remaining questions.

Line 576. "(k) Experimental design for results in k-o." I think the authors meant panels l-o.

Line 582. "(p) Experimental design for results in k-o." I think the authors means panels q-u.

Lines 583 and 585. The authors say iBAT in line 583 but switch to BAT in line 585. Are iBAT and BAT the same or different?

Lines 588-589. "Repeated measures two-way ANOVA with diet x time interaction and Sidak's post-hoc test for each time point was applied in b." Was the same method used for panel e?

Figure 1, panel o. The two bars are almost the same height, yet there's a p-value bracket connecting them that indicates $p < 0.05$. Also, the bar height of PNe goes between the 2nd-highest & 3rd-highest data point (instead of being near the 5th-highest) when there's no high-glucose outlier pulling the mean up. I suspect that the p-value bracket is correct, but that the PNE bar height may be wrong. Please look into this.

Line 619, which says "(w-y) (w) Western blots of GLUT1 in iWAT (HFD n=4, PNe n=5) and iBAT (HFD n=6, PNe n=6) or (y)". I think the label "(x)" was accidentally omitted from the middle of that line.

Figure 2, panel u. The Y-axis label says relative mRNA expression, and for every gene except Pparg, the control group's bar heights are 1.0, indicating that relative mRNA expression must mean relative to the control group's average expression. Which raises the question of, why is the control group's bar height different from 1.0 in the case of Pparg expression.

Figure 2, panel v. The Y-axis label says relative mRNA expression, but unlike in panel u, none of the control group's bar heights are equal to 1.0. All of them are lower than 1.0. Why is that? Were the expression data in panel v expressed relative to something different from the control group?

Lines 623 and 624. "Two-Way ANOVA with Sidak's post-hoc test was applied in u, v." Applying 2-way ANOVA to the data in those two panels requires meeting the equal-variance assumption across all groups, which in this case means across all genes as well as across PNe vs Ctrl groups. How reasonable is it to assume that different genes have the same variances in gene expression? Perhaps it might be more reasonable to assume that different genes have different variances in gene expression, and accordingly to (1) compare PNe to Ctrl via t-test within each gene, and (2) perform the Sidak adjustment (or Bonferroni adjustment) by hand on all the t-test results.

Figure 3, panels l through p. I had the impression that all the data in these figure panels came from the same mice, not different mice, but the sample sizes described in lines 660 through 665 do not stay the same from panel to panel. Please explain.

Figure 6, panel s. The Y-axis label says "AKT^{p473}/AKT" when I think the authors meant to say "AKT^{pS473}/AKT". Also, the Y-axis label for panel s does not say what the S473 phosphorylation ratios are relative to, whereas the corresponding label for panel t says relative to Ctrl. That seems odd. Finally (lines 704-705), the sample sizes of the HFD mice change from panel s to panel t even though they are different tissues from the same mice. Why is that?

Line 724, referring to Figure 6 panels j-n. The sample sizes were omitted from description of panel l, the iBAT panel.

(Remarks on code availability)

Point-by-point Response to Reviewer's Comments

“Genkwanin glycosides from the *Phaleria nisidai* extract improve glucose homeostasis by stimulating glucose uptake into adipose tissues”

All major changes to address the reviewers' questions or comments are highlighted in red in the manuscript file.

Points raised by reviewer 1

This manuscript suggests that Genkwanin glycosides from the *Phaleria nisidai* extract may improve glucose homeostasis by stimulating insulin-independent glucose uptake. It is huge amount of work and the authors used multiple approaches to support their findings in the paper, which should be appreciated. Although the paper is interesting, there are several issues that need attention, and there are some deficiencies in the evidence. According to the author, the innovation of this study lies in the successful completion of the reverse pharmacological method for insulin resistance for the first time. But the certain efficacy and proposed mode of action of PNe and GE glycosides remain to be investigated in humans. Besides that, the author mentioned the shortcomings of the study in line 393-408, but for the study of molecular mechanism, gene knockout experiment is necessary. Many deficiencies can and need to be improved.

In terms of experimental design and result presentation, the following problems may exist:

1. In Fig. 2i, phospho-sites which treated by PNe seem significantly lower than CTRL at insulin-stimulated conditions. How to draw this conclusion 'Both phospho-sites were not affected by PNe exposure at baseline or insulin-stimulated conditions'?

We thank the author for the remark, and we have repeated the western blots once and re-analyzed all blots to address this concern, also in relation to point 2. With this optimized analysis, we do not observe a significant downregulation in AKT phosphorylation at T308. However, we can confirm the downregulation in AKT phosphorylation at AKT S473 for both 3T3-L1 and immortalized brown adipocytes (Fig. 1h, k-l). We propose that this reduction in AKT S473 phosphorylation may contribute to the lack of an additive effect on insulin stimulated glucose uptake (Fig1. a-b). We have rephrased the section accordingly in lines 132-137.

2. When calculating the expression of phosphorylated proteins (Fig. 2j-m), why use the value of AKTpS473or pT308/HSP90 instead of AKTpS473or pT308/AKT? More importantly, how the authors determine the way and days of drug administrations ? Why treat with PNe for 3 days ?

We chose to normalize the phosphorylated protein levels to the loading control of the same western blot to limit variance between western blots. As suggested by the reviewer, we have now performed the analysis where the phosphorylated and total proteins are first normalized to their respective loading controls and then to each other.

Our interest was to assess the effects of chronic PNe treatment on the fully differentiate, mature adipocytes. According to standard differentiation protocols, 3T3-L1 and immortalized brown adipocytes reach this state around day 6 of differentiation. The duration (72 hours) and dosing (200 ug/ml) were selected based on a literature study searching for phytoextracts tested in comparable assays (e.g. PMID: 12926517; PMID: 28854906)

3. In Fig. 3a-c and 3d-e, what is the reason for the different duration of treatment with PNe? Is there a time when all the above changes can occur ?

We were interested in the early changes and cellular signals that might drive the increase in glucose uptake and GLUT1 protein levels, which we observed during the 3 days treatment regime. Therefore, we stimulated the cells for 30 min and 180 min in Fig. 3. We observed increased phosphoERK1/2 signal up to 3 hours of stimulation (Fig. 3b-c) and we confirmed the nuclear localization of total ERK1/2 at this time point. In addition, we showed that this is a PKC-dependent process. Regarding 3d, we have pre-treated the cells for 30 min with the ERK1/2 inhibitor and stimulated the cells for 4 hours with PNe. We have increased the stimulation time from 180 to 240

min to ensure that we have a detectable effect at the mRNA level. We have now repeated the experiment (lines 194-195) and could show that PNe increases Glut1 mRNA already after 180 min of stimulation, consistent with the other time points (new Supplementary Fig. 4d). As expected, we did not observe a change in Glut4 expression (new Supplementary Fig. 4e).

4. 'All investigated fractions improved insulin sensitivity (Fig. 4c-d) after 4 weeks of treatment, while only F4 effectively ameliorated insulin sensitivity after prolonged DI (Fig. 4e-f).' When is the specific time of prolonging DI ?

We apologize for being not specific enough. The time of prolonging the dietary intervention (DI) was 6 weeks. A first ITT was performed after 4 weeks of (DI), the second ITT after the prolonged DI was at 10 weeks. We have now specified the duration of the DI more clearly in the results section (lines 237-239).

5. There may be some errors in the significance symbol of Fig. 4j

We apologize for the misalignment of the significance symbol in Fig 4j and we have now correctly aligned the symbols in Fig. 4j (Line 243).

6. Why is the DI duration in Fig. 4k different from that in Fig. 4a ?

This mistake has been pointed out by other reviewers as well and we are thankful for the thorough evaluation of our manuscript. The experimental design in Fig. 4k is wrongly placed and therefore the labeling of the HFD-feeding period is inconsistent. Fig. 4k (now new supplementary Fig. 8a) refers to the data in the new Supplementary Fig. 8b-e. In this experiment, we have increased the duration of initial HFD feeding to 12 weeks to compare the effectiveness of F4 against metformin in a model of more severe insulin resistance compared to the initial 6 weeks of HFD-feeding as shown in Fig. 4a. The duration of the DI is different between the two experiments because in the metformin cohort we were only interested in the effects on insulin sensitivity and the animals were sacrificed afterwards. The ITTs were performed after a comparable duration of DI (9 weeks vs 10 weeks).

Overall, we have corrected the experimental design in 4k and moved the layout to new Supplementary Fig. 8a. In addition, we have created a new Fig. 4k, which summarizes the experimental layout for the clamp experiment (data presented in in Figs. 4l-s).

Points raised by reviewer 2

The manuscript "Genkwanin glycosides from the *Phaleria nisidai* extract improve glucose homeostasis by stimulating insulin-independent glucose uptake" studies the function of natural compounds isolated of *Phaleria nisidai* (PN). Obesity and diabetes have reached pandemic dimensions and novel therapies are urgently needed. The manuscript consists of two parts (a) the analysis of the effects and molecular mechanisms of PN extracts in adipocytes and mice; (b) the identification of genkwanin as one active compound present in PN.

Overall this is a very interesting paper with high translational relevance since it defines natural compounds that have antidiabetic actions. Nevertheless, the following points should be addressed before publication:

General comments:

1. The abbreviations for *Phaleria nisidai* and its extract are confusing: PN versus PNe. PN is not clearly defined; PNe is defined but not always used as abbreviation.

We apologize for the inconsistency and the confusion. We have now clearly defined that PNe is the *Phaleria nisidai* extract used in the experiments and represents the freeze-dried extract prepared from the *Phaleria nisidai* leaf decoction (Lines 93-85). PN refers to the plant *Phaleria nisidai* (Lines 81-83).

2. Fig. 2: 3T3L1 cells were used throughout the ms, but this cell model has several disadvantages. It would be important that primary adipocytes derived from SVF of BAT or WAT should be used for the major experiments. Especially since the authors observe no effect of insulin on glut4, which is puzzling given the previous studies showed a clear increase of Glut1 in primary adipocytes treated

with insulin (Hajduch 1992). Importantly, human primary white adipocytes should be used to further corroborate the relevance of the findings in murine cells.

We thank the reviewer for these comments, and we have repeated the key assays in primary murine white and brown adipocytes (new Supplementary Fig. 2, lines 148-157), human beige or white hMADS (human mesenchymal adipose derived stem cells, new Supplementary Fig. 3a-f) to validate the effects of PNe on adipocyte glucose metabolism (lines 158-164). In summary, we can confirm the findings from the immortalized adipocyte models including a) increased glycolysis, b) increased basal glucose uptake rates c) elevated GLUT1 protein and mRNA expression following 72 hours of PNe (200 µg/mL) treatment. Furthermore, we measured glucose uptake in *in vitro* differentiated primary human adipocytes isolated from the subcutaneous depot of one donor that was paralleled by elevated GLUT1 protein levels (new Supplementary Fig. 3g-l, lines 164-168). In contrast to our findings in the mouse adipocyte models, we have observed a reduction in GLUT4 protein and mRNA expression in the human primary white adipocytes.

3. GLUT1 and GLUT4 plasma membrane levels should be measured for PNe-treated cells. It was previously shown in adipocytes that insulin leads to translocation of GLUT1 from intracellular vesicles to the plasma membrane (Holman 1990). This could be another important mechanism of PNe action.

We agree with the reviewer that measuring GLUT1 and GLUT4 PM-levels is required to provide further evidence that PNe increases glucose uptake via the basal glucose transporter GLUT1 (lines 173-175). We performed plasma membrane fractionation assays with immortalized brown adipocytes and could confirm an increase in PM GLUT1 but not GLUT4 protein after 72 hours of treatment (new Supplementary Fig. 3l-n). Importantly, we could validate these findings in the plasma membrane fractions from the human primary white adipocytes treated with PNe (new Supplementary Fig. 3o-p).

4. PNe increases glycolysis, does PNe also increase lipolysis?

We thank the reviewer for his suggestion and measured glycerol and non-esterified fatty acid (NEFA) levels in the media of immortalized brown adipocytes in the basal state and after isoproterenol stimulation (new Supplementary Fig. 2a-b). As expected, we observed an increase in NEFA and glycerol release following isoproterenol stimulation but no additive effects of PNe treatment in either the basal or stimulated condition (lines 140).

5. The authors state (line 134): "We could translate these findings to our animal model system, where we detected higher GLUT1 but not GLUT4 protein content in the iWAT and iBAT of PNe-fed animals compared to HFD controls (Fig. 2t-v)." Pls rephrase and give % values and significance in the text, there is a clear increase in GLUT4 levels in PNe treated BAT (Fig 2v).

We thank the reviewer for the careful inspection of our data. As suggested, we have rephrased the section and given % values for the increase in GLUT1, GLUT4 protein levels as well as the exact p-values in the text (lines 170-176). GLUT4 protein levels are not changed in the iBAT if a multiple t-test is performed, and p-adj. values are reported. However, if GLUT4 levels between HFD and PNe-fed animals are compared alone, PNe treatment leads to a significant increase in iBAT GLUT4 (p=0.046). We have rephrased the section accordingly: "Our findings evidence higher GLUT1 protein levels induced by PNe derive from increased GLUT1 transcription. We could translate these findings to our *in vivo* studies, where we detected higher GLUT1 protein content in the iWAT (+50%, p=0.032) and iBAT (+18% increase, p=0.035) of PNe-fed animals compared to HFD controls (Fig. 2s-t). GLUT4 protein levels were not modulated by PNe in the iWAT (-9%, p=0.33) but trended to increase in the iBAT (+20%, p=0.088)".

6. Fig. 2w-y: again there is a clear increase in GLUT4 in "brown" adipocytes; only n=3 experiments were done, the number of experiments should be increased; it would be very interesting to see the effects in human brown adipocytes.

We thank the reviewer for his/her interest in the data regarding the brown adipocytes. Due to the difficulty of obtaining human brown stromal vascular fraction for *in vitro* differentiation, we have repeated the experiments with human beige hMADS to increase the n-number (lines 158-165). These cells are derived from the subcutaneous depot of a male donor and can differentiate into beige adipocytes (Ucp1 expression, cAMP responsive) by rosiglitazone stimulation. When combining these two independent experiments, we observe a significant increase in both GLUT1 and GLUT4 protein levels (new Supplementary Fig. 3a-c).

7. Fig. 3b-c: erk signaling promotes adipogenesis (prusty 2002), the authors should investigate the effect of PNe on markers for adipogenesis

We thank the reviewer for the provided reference. To investigate a potential effect of PNe on adipogenesis, we have treated immortalized brown adipocytes (iBAs) and 3T3-L1 adipocytes with increasing concentrations of PNe throughout differentiation (lines 203-208). We did not observe a significant increase in adipogenesis in immortalized brown adipocytes as measured by the accumulation of lipid droplets in response to 0, 1.8, 8, 40 $\mu\text{g}/\text{mL}$ PNe, which was quantified using an automated microscope (new Supplementary Fig. 4 g-i). While we found a mild stimulatory effect of PNe on adipogenesis at lower doses in 3T3-L1 cells, differentiation capacity was declining at the highest concentration of 200 $\mu\text{g}/\text{mL}$ PNe. According to the literature, ERK1/2 signaling can promote or inhibit adipogenesis depending on the specific time window and duration of active ERK1/2 signaling. ERK-activity is required for adipogenesis during the early phase adipogenic differentiation after induction (PMID:12270934) but has negative effects at later stages (PMID:17210639). We speculate that the highest PNe dose acts as strong and persistent ERK activator, that blunts adipogenic differentiation capacity.

8. Fig. 3e-f: in 3f PNe-treated cells are missing. Moreover, it is very difficult to see clear differences in the pictures shown (higher magnifications, insets might be helpful).

We apologize for the incorrect labelling of Fig. 3f, the first sample labelled with "PNe+PD" is wrongly labelled and should be PNe only. As suggested by the reviewer, we have included images at higher magnification (40x) from an independent experiment in the new supplementary Fig. 4f (line 198).

9. Fig. 6: a thorough analysis of the molecular mechanisms and pathways of GE in adipocytes (especially brown) is missing; especially Glut1 and 4 expression and membrane localization would be interesting. The quality of the western blots shown in fig. 6 (especially p and q) does not permit a clear quantification of Glut1 phosphorylation.

We agree with the reviewer that western blots showing Glut1 phosphorylation don't allow for a reliable conclusion. We have removed this data from the manuscript. As the reviewer suggested, we have tested *in vitro* if GE stimulates glucose uptake *in vitro* (lines 335-339). We have tested the extract in different models (3T3-L1, iBAs as well as primary brown adipocytes). Unfortunately, we could not confirm a GE-mediated increase in glucose uptake or GLUT1 mRNA/protein levels which contrast our *in vivo* findings (new Supplementary Fig. 11a-i). We think that GE might be further modified *in vivo* and we have done an *in silico* prediction of potential metabolites which could be formed from GE (new Supplementary Fig. 12a, lines 340-432). Potential candidates were tectochrysin, apigenin, phloretin, rhamnocitrin, hydroxygenkwanin, phloroglucinol, and naringenin. We tested all 7 predicted metabolites *in vitro* on immortalized brown adipocytes but could not detect any effect on glucose uptake after 72 hours of stimulation (new Supplementary Fig. 12). While tectochrysin was reported to increase glucose uptake rates in 3T3-L1 adipocytes (PMID:37888838), we could neither confirm this effect in iBAs nor human primary white adipocytes (3 hours and 72 hours tested, not shown). This could potentially imply a combinatorial activity of different metabolites or distinct effect between *in vivo* and *in vitro* models.

Minor points:

10. Fig. 2: It should be clearly stated in the text and/or legend of figure 2 which cells were used. Is fig. 2a missing? Only legends are shown.

Fig. 2a was meant to display the color code for all displayed graphs in Fig. 2. To avoid confusion and increase clarity we have removed old Fig. 2a and have more clearly placed the figure legends. We also apologize for the unclear labelling and have now clearly labelled the cell types used in the figure legends (Lines 601).

11. line 132: "Figure 3x"? Should be 2x.

We apologize for the mislabeling and this error. This figure has now been moved to new Supplementary Fig. 3b.

Points raised by reviewer #3

In this study the authors aim to define the active compound of the *Phaleria nisidai* plant that is responsible for the glucose-lowering effects observed in Palaun traditional medicine concepts. While this is per se a very interesting and promising study, and the authors invested a lot of work in conducting the presented experiments, I have serious concerns about the set-up of the animal studies and interpretation of the data based on the way they were presented. A substantial revision of the manuscript is required in order to be considered for publication.

General comments:

1. All of the GTT and ITT data are presented as normalized to initial blood value. For correct interpretation of the data, the presentation as raw data in mmol/l is crucial.

We agree with the reviewer that the presentation of absolute concentrations is important for data interpretation. We are now displaying all data for ITTs and GTTs in mmol/L.

2. Please refer for correct presentation of these data and calculation of AUC to "GTTs and ITTs in mice: simple tests, complex answers" (Nature Metabolism volume 3, pages 883–886 (2021)). This is specifically important, because there are significant differences between the RBG in the treatment groups. A lower initial blood glucose value (e.g., 8 mmol/l vs. 10 mmol/l) will show a higher percentage drop (25% vs. 20%) even if both groups decreased by the same value (e.g., 2 mmol/l).

We thank the reviewer for drawing our attention to this guideline. We have taken them into consideration in the revised manuscript and analyzed our data accordingly. The new analysis does not change any major conclusions and supports the antidiabetic potential of PNe and its metabolite, genkwanin.

3. Also, show body weight development in g.

As requested by the reviewer, we show the body weight development now in g.

4. Further, the effects of PNe are described as insulin-independent, yet it is claimed that insulin sensitivity is increased. 'Increasing insulin sensitivity' means enhancing the action of insulin on insulin-sensitive tissues, which per se cannot be an 'insulin-independent' process.

We agree with the reviewer that the phrasing does not reflect the *in vivo* findings. We have also decided to change the title of the manuscript to address this concern. We have rephrased the corresponding sections more carefully throughout the manuscript, considering the observed improvement in insulin sensitivity. We claim that PNe increases the levels of GLUT1, which acts independently of insulin as glucose transporter while the parallel improvements in insulin sensitivity are clearly dependent on insulin action/signaling.

5. Overall, the study is confusing, because multiple possible mechanisms are introduced for the plant fractions, that are not necessarily reflected in the PNe data. There is a lack of consistency.

- PNe decreases insulin levels and urinary glucose excretion, and increases glucose uptake into adipose tissues
- All fractions decrease random or fasting blood glucose levels, but PNe does not (Fig. 4b, 4j)
- F3 increases GLP1 levels. Is there an effect on insulin secretion?
- F4 increases glucose uptake into adipose tissues AND soleus muscle, there is no effect on basal glucose turnover
- GE decreases RBG (Fig. 5g) but PNe does not, GE increases urinary glucose excretion – PNe decreases it
- GE increases AKT phosphorylation in fat, PNe has no effect on AKT phosphorylation
- GE has also an effect on hepatic insulin sensitivity and muscle glucose uptake, but PNe does not

We thank the reviewer for this critical comment, and we apologize for not being clear in our interpretation of the data. We agree that multiple possible mechanisms are presented for the extract and the fractionations. First, we would like to emphasize that the plant extract (PNe) is a complex mixture of different compounds and for some effects induced by PNe might be caused by additive or synergistic effects of individual components. In turn, not all of these bioactive compounds are present in the fractions, which might lead to different or dampened effects. In our opinion, this is the reason not all treatment conditions induce the same biological effect or might even result in opposing results. As the reviewer observed, we did not demonstrate an effect of PNe on AKT phosphorylation. While these experiments were performed in vitro, the AKT phosphorylation with GE was performed in adipose tissue samples. Considering PNe metabolism in vivo, it is possible that new metabolites are formed whose activities that are not revealed in cell culture experiments. F3 did not affect glucose-stimulated insulin secretion (15 minutes after glucose load). However, mangiferin is the main constituent present in F3 and was previously shown to promote insulin secretion (PMID: 24853132). Therefore, higher doses of F3 might also promote glucose-stimulated insulin secretion. We have adapted the manuscript throughout the results and discussion to better address these points.

6. Detailed comments:- No real conclusions can be made of the data set presented in Figure 1, because all of the tests were performed at completely different stages of the dietary interventions

- There is a huge difference between 6 and 12 weeks of HFD when it comes to the development of insulin resistance. The accumulation of lipids in tissues like liver and muscle differs significantly based on the length of the HFD.

We agree that the different duration of HFD-feeding leads to different metabolic and physiological states. We on purpose applied different durations of HFD-feeding to test for the efficacy of PNe or Genkwanin at various stages of metabolic impairment. While we agree that this limits the comparability of the results, our primary aim was to dissect if more severely metabolically challenged mice are still responsive to the dietary interventions.

- In order to compare the tests with each other, there should be a max. of 1-2 weeks in between the tests.

We agree that metabolic tests within the same experiment should be performed within a short time to allow comparability. We have revised the results section related to Fig 1 to make these differences more evident and clearly distinct between different experiments/cohorts of mice.

- ITT was performed after 12 weeks of HFD and 5 weeks of DI, but the GTT was performed after 6 weeks of HFD and 2 weeks of DI. Insulin levels should be measured in parallel to the GTT. These data are not reliable and thus cannot be interpreted.

We agree with the reviewer that the data presented in Fig. 1e and 1k are not reliable due to the differences in initial HFD-feeding, which as emphasized by the reviewer impacts metabolic health. The insulin levels shown in Fig. 1n are from the GTT in Fig. 1l-m at the 30 min time point. Our idea was to evaluate PNe efficacy in less metabolically impaired animals and add whether early onset of

dietary intervention might prevent or reduced metabolic complications. We have rephrased the results section (104-107) and tried to emphasize this difference for a transparent understanding

- Figure 1m, insulin levels measured after 30 min. The methods describe a measurement after 15 min.

We apologize for the inconsistency. The correct measurement time point was after 30 min, which was corrected in the methods sections (line 860).

- In order to draw conclusions from the tissue uptake data in Fig. 1o-t, blood insulin levels are needed.

Unfortunately, we did not collect blood samples during the ¹⁴C-2-deoxyglucose tracing because of the high radioactivity. We agree that complementary insulin measurements would allow for a detailed interpretation of the results. The glucose tracing was performed after 6 weeks of HFD and 7 weeks of DI. We have now measured fasting blood insulin levels in an independent cohort at a comparable time point (initial HFD 6 weeks, DI 8 weeks) and we could not detect any significant differences in fasting insulin concentrations (PNe vs HFD 783.8 vs 890.4 pg/mL p=0.784). Please refer to lines 116-119 and new Supplementary Fig. 1f.

- Figure 2j: Why normalization to GAPDH? GAPDH not shown in blots. Since several glycolytic enzymes are upregulated, is GAPDH protein content different?

As the respective graphs shows multiple western blots, we have displayed only one representative loading control. During the revision process, we repeated some of the western blots and now all blots use either HSP90 or γ Tubulin as loading control. GAPDH protein content is not different between control and PNe treated brown or 3T3-L1 adipocytes after 72 hours of PNe treatment as shown below.

- GLUT4 mRNA content is higher in Fig. 2s and trends higher (p-val = ?) in Fig. 2r. Is there is a difference in GLUT4 translocation? GLUT4 translocation assays should be performed or GLUT4 membrane content measured.

We agree that Glut4 mRNA tends to be increased (p=0.062). We have quantified GLUT4 PM levels in immortalized brown adipocytes after 3 days of PNe stimulation and could confirm a small but significant increase in GLUT4 plasma membrane levels (new Supplementary Fig. 3l-n). In addition, more pronounced effects GLUT1 PM levels were observed. Therefore, we conclude that PNe promotes baseline glucose uptake rates predominantly by the combined effects of increased GLUT1 and GLUT4 PM protein levels (lines 173-180).

- What are the expression levels of other enzymes involved in the glycolysis pathway?

We have measured additional glycolytic genes (lines 148-154) in the newly generated data from primary murine and brown adipocyte (new Supplementary Fig. 2 *Gpi, Aldoa, Pgam1, Eno1, Pkm2*), with most of them being significantly upregulated after PNe. We did not observe any significant, overall alterations in the expression levels of genes encoding subunits of pyruvate dehydrogenase complex (*Pdha, Pdhb, Pdhx*), although certain genes tended to be lower expressed with PNe stimulation. We think that increased glucose oxidation stems from increased expression of selected genes in the glycolysis pathways combined with increased glucose influx into the cells to accommodate the high glucose load.

12. While AKT phosphorylation can facilitate GLUT4 translocation (line 120), there are AKT-independent mechanisms that promote GLUT4 translocation (T. McGraw, *Mol Biol Cell*. 2006 Oct;17(10):4484-93. So that's not necessarily an exclusion criterion.

We agree with the reviewer that *Glut4* translocation is not exclusively mediated by AKT. We have rephrased this section in lines 129-134 more carefully:

“These measurements suggest that PNe does not primarily act on the insulin-dependent arm of glucose clearance but rather boosts basal glucose uptake. Consistent with augmented baseline glucose uptake, GLUT4 protein content was unchanged by PNe, whereas GLUT1 protein was upregulated (Fig. 2c-g). We next examined if PNe potentially modulates the phosphorylation state of Protein Kinase (AKT) at S473 and T308, which controls the translocation of insulin-responsive GLUT4 to the plasma membrane.”

13. Line 189, Figure 4b: ‘All fractions lowered RBG’ is too generalized. It really depends on the time point of the intervention and the fraction.

We thank the reviewer for this comment. We have now described the results concerning the RBG levels in more detail in lines 235-236:

“Over time, F4 consistently lowered random fed blood glucose (RBG) concentrations during the DI (Fig. 4b)”.

14. How do the authors explain, that over a time period of intervention the control mice do not show an increase in RBG (Fig. 4b) although there is a constant weight gain? 8 mmol/l seems low for fed blood glucose levels on HFD.

We thank the reviewer for the comment. We agree that 8 mmol/l seems low for fed blood glucose levels on HFD. However, we would like to point out that the mice used throughout the manuscript are the C57B/6N strain, which develop less hyperglycemia than the commonly used J strain for such studies. The animals were on a reversed light-cycle and blood sampling occurred in the morning around 10 am. The literature contains several comparable studies where RBG does not increase over the period of the intervention. We have addressed this concern in the revised manuscript (lines 236-237) as followed:

“Counterintuitively, RBG levels did not significantly increase during the study, which is consistent with literature reports”

Asia A et al., 2019 *Atheroscler Thromb*. PMID: 30971640

Tornovsky-Babeay S. et al., 2021, *Diabetologia* PMID: 33558985.

15. Why are the fasting blood glucose levels in Figure 4j similar to or even higher than RBG in Fig. 4b?

We appreciate the reviewer’s careful inspection of the data. The fasting blood glucose levels were measured terminally before sacrificing the animals and we speculate that the higher fasting blood glucose level are stress-induced due to transfer of the animals from the experimental to the dissection room. However, it should be noted that controls and treated mice were always handled in the exact same way.

16. Figure 4k displays PNe as an experimental group. Were insulin clamps performed in PNe treated group in comparison? 4k says 6 weeks of HFD, line 200, however, mentions 12 weeks of HFD. Which one is it?

This mistake has been pointed out by other reviewers as well and we are thankful for the thorough evaluation of our manuscript. The experimental design in Figure 4k is wrongly placed and therefore the labeling of the HFD-feeding period inconsistent. PNe treated mice were not clamped. Fig. 4k (now new supplementary Fig. 8a, lines 246-248) refers to the data in the new supplementary Figs. 8b-e. In this experiment, we have increased the duration of initial HFD feeding to 12 weeks to compare the effectiveness of F4 against metformin in a model of more severe insulin resistance compared to the initial 6 weeks of HFD-feeding as shown in Fig. 4a (lines 227-228). The duration of the DI is different between the two experiments because in the metformin cohort we were only interested in the effects on insulin sensitivity and the animals were sacrificed afterwards. The ITTs were performed after a comparable duration of DI (9 weeks vs 10 weeks).

Overall, we have corrected the experimental design in 4k and moved the layout to new Supplementary Fig. 8 (new 8a). In addition, we have created a new Fig. 4k, which summarizes the experimental layout for the clamp experiment (data presented in in Figs. 4l-s) in lines 253-255.

17. Is the effect of F4 on glucose turnover in Fig. 4m statistically significant? The basal glucose turnover rate is not increased and PNe does not improve fasting blood glucose levels (Fig. 4j). How do you explain that in context of the proposed 'insulin-independent GLUT1 mediated glucose uptake' mechanism? Similar in Figure 7f.

We would like to emphasize two important aspects:

First, as the reviewer has commented under point 16, the experimental layout in Fig. 4k is misleading. No glucose clamps were performed in PNe treated animals. However, there is a significant reduction in fasting blood glucose for F4 treated mice after 14 weeks of DI (Fig. 4j). A new set of F4 treated mice was then included in the clamp study (Figs. 4 k-s). As already recognized by the reviewer, experiments were performed at different time points, which unfortunately limits comparability. The glucose clamp studies were performed after 6 weeks of HFD feeding and 7 weeks of dietary intervention (lines 253-255). Therefore, there is a difference of 7 weeks of DI but also a difference of 7 weeks in HFD feeding between the data in Fig. 4j and Figs. k-l. As the glucose clamps are a challenging experiment requiring a catheter implantation surgery, we decided to perform this experiment once insulin sensitivity trends to improve as measured by ITT and before the animals suffer from a high body weight, which might affect surgical recovery. The glucose turnover presented in Fig. 4m is not changed. This might relate to the shorter DI period (compared to Fig. 4j), where the effects of F4 may not be fully established.

Second, the basal glucose turnover is determined after 6 hours fast during the active phase of the animal. Studies in humans show that adipose tissue glucose uptake is reduced during fasting (PMID 11237926). Importantly, fasting also reduces GLUT1 protein levels in mice, which could override our proposed GLUT1-mediated increase in glucose transport thereby biasing the read out for basal glucose turnover (PMID 35366539).

18. Figure 4r. Because of the glucose-6-phosphatase activity in liver, 14C-2-DG-P content in liver is not representative of liver glucose uptake. See also Figures 1q, 7k.

We thank the reviewer for pointing out the glucose-6-phosphatase activity in the liver. We agree that we cannot reliably measure glucose uptake in the liver with our data and we have removed the corresponding graphs.

19. F4 and GE both increase glucose uptake into soleus muscle. A muscle insulin-dependent mechanism should be considered. Were muscle tissues isolated for signaling analysis? Or muscle cell lines treated with the compounds?

We have measured AKT phosphorylation in soleus and gastrocnemius muscles of GE treated mice, which were collected in the random fed state (lines 332-333). We did not observe any significant changes in the phosphorylation of AKT T308 (new Supplementary Fig. 10c-f, k-l). We have also treated C2C12 skeletal muscle cells with the whole extract (PNe) and the fraction of interest (F4), where genkwanin was determined as secondary metabolite (lines 266-269). We did not observe an increase in glucose uptake in these cells (new Supplementary Fig. 1). Similarly, plasma membrane levels of GLUT1 or GLUT4 were not affected by PNe treatment in C2C12 cells (new Supplementary Fig. 7m-n).

20. Figure 5, iWAT shows increased glucose uptake, but no increase in GLUT1. A GLUT4 translocation-mediated mechanism should be considered here. Were insulin levels measured in these mice? The increase in AKT phosphorylation might be due to differences in insulin levels and thus differences in membrane GLUT4. In addition, an increased flux through GLUT1 could be considered as glucose transport rates through GLUTs are not only determined by protein abundance but also activity.

Fig. 5 does not show any glucose uptake data, we therefore assume that the reviewer was referring to Fig. 6. Notably, the results presented in experiment 6 are from two different cohorts of mice. We have adjusted the corresponding results section to make this difference obvious to the reader. Glucose uptake rates were determined after 6 weeks of initial HFD feeding and after 6 weeks of DI (new Fig. 6a). Although we did not measure insulin levels during the experiment, fasting insulin levels were not different after 6 weeks of initial HFD-feeding and 4 weeks of DI (Supplementary Fig. 9j). AKT phosphorylation at S473 was determined after 6 weeks of initial HFD and 9 weeks of DI in the random fed state (new Fig. 6e). At this time point, insulin concentrations tended to be higher in GE treated mice (2.37 vs 2.03 pg/mL) compared to HFD controls (see supplementary Figs. 9j, n). Suggesting that insulin levels might affect AKT phosphorylation status and thereby GLUT4-mediated glucose uptake could contribute to the glucose disposal into the iWAT. We have added a corresponding sentence in the discussion (lines 478-480). In addition, the increased insulin sensitivity in the iWAT of GE mice (Fig. 7i) supports that this trending increase in plasma insulin levels might be sufficient to induce AKT phosphorylation.

21. Why were mice gavaged with glucose by BW for glucose-stimulated insulin secretion tests, but were give a fixed glucose dose for GLP-1 measurements?

Plasma GLP-1 measurements were recorded after 13 weeks of DI. At this time point, there were differences in body weight between groups (means per group: HFD=45.08, PNe=46.54, F2=43.78, F3=44.73, F4=44.43g at week14). The administrated dose of glucose (40% glucose solution, 200 uL) therefore corresponds to an approximate dose of 1.8-2 g glucose /kg body weight. This glucose dose is comparable to what other studies report when measuring glucose stimulates GLP-1 secretion (e.g. PMIDs:15793244, 28526921; 12540373). We have added this information to the results section. To measure glucose-stimulated insulin secretion we applied our standard oGTT protocol described in the method section (lines 851-856).

22. In order to interpret the tissue-specific uptake data for PN or GE, blood insulin levels are needed.

As described above, we did not measure fasting insulin levels during the glucose tracing for PNe treated mice (Fig 1) but we did not measure a significant difference between PNe and HFD animals at a comparable time point (6 weeks HFD, 8 weeks PNe) in a different experimental group (lines 116-119). Regarding GE, no significant differences in fasting (new Supplementary Fig. 9j) or random fed insulin levels (new supplementary Fig. 9n) were observed when compared to HFD-controls at various stages of the dietary intervention (lines 305-308). Please also refer to remark 20 for additional details and explanations. Therefore, we argue that plasma insulin levels in the mentioned experiments did not affect glucose uptake rates in the tissues.

Minor comments:

23. Check Figure references, e.g., line 132, line 262 Fig. 1u doesn't exist.

We apologize for the incorrect references. We have now carefully revised all the figure references in the updated version of the manuscript.

24. Define PD-98059 at first mentioning, line 161.

We have now corrected this statement and defined the ERK inhibitor used in the study (PD184352) in line 193.

25. Since the mice received the compounds in the food, the dosage per BW can only be an estimation. It would more correct, to give the concentration of PNe (or compounds) in food and then what daily dose it would correspond to based on daily food intake.

We appreciate the suggestion, and we would like to describe the preparation of the supplemented diet in more detail. From our recordings during the HFD-feeding period we calculated an average food intake of 2.5-3g HFD per day per mouse. Next, we calculated how much extract or compound we must add to the diet to cover the daily dose through daily food intake (e.g. 130 mg/kg BW for PNe). Therefore, we have taken the average food intake into account, when calculating the dietary supplementation and with this approach, we think it is correct to state the dosage per BW.

Point-by-point Response to Reviewer's Comments – Revision Round 2

“Genkwanin glycosides from the *Phaleria nisidai* extract improve glucose homeostasis by stimulating glucose uptake into adipose tissues”

All major changes to address the reviewers' questions or comments are highlighted in red in the manuscript file.

Points raised by reviewer 3

In the revised version of their manuscript the authors performed several additional experiments and acknowledged the study's limitations in the experimental design.

Although the quality of the manuscript has improved, certain deficiencies still need to be addressed.

a) The numerous phenotypes induced by the various fractions of PNe make the manuscript challenging to follow. It would benefit from a graphical abstract, table or a flow chart summarizing the effects of PNe and the corresponding fractions (e.g. GTT up, glucose in WAT uptake unchanged, AKT S473 phosphorylation up, etc.)

As suggested by the reviewer, we have compiled a new overview table “Table 2 Overview In vivo effects” with three categories “Metabolic Phenotyping”, “AKT-signaling” and “Biochemical parameters” to summarize the effects of PNe and the corresponding fractions as well as individual compounds. We refer to this table throughout the results section as highlighted in red. We hope this comprehensive overview provides a better understanding of the numerous effects induced by the different treatments

b) Western blots displaying the membrane fractions of GLUT1/4 lack a membrane marker such as Na⁺/K⁺-ATPase.

We thank the reviewer for this important point. While we agree that the inclusion of a membrane marker such as Na⁺/K⁺-ATPase would strengthen the validation of membrane fraction purity, we were unable to probe all blots with a PM marker due to sample limitations. In particular, the protein yield from human in vitro differentiated primary adipocytes is very low. However, the applied membrane fractionation protocol was developed by Yamamoto et al and has been validated previously by others e.g. Gaddam et al (2022, PMID: 3506034), Yamashita et al., (2024, PMID: 3829212) or Seo et al., (2019, PMID: 30609251). We ensured minimal cytosolic contamination as confirmed by the absence (or low levels) of cytosolic marker (Hsp90, β-actin) in the membrane fraction. In more detail, we present the following controls. Supplementary Fig. 3o shows plasma membrane fractionation from immortalized brown adipocytes with a strong depletion of HSP90 in the plasma membrane fraction but enrichment in the whole cell lysate (WCL). We have now performed additional western blots for all replicates (new Supplementary Fig. 3o), where we use Insulin receptor B as membrane control and b-actin as cytosolic marker. The insulin receptor is an acceptable marker to confirm the enrichment of membrane proteins as reported by others (PMID: 38292121 or PMID: 15699039). As expected, B-actin is depleted in the PM-fraction while IR-B is enriched in the PM but also detectable to in the WCL. This is expected, given the internalization and recycling of the insulin receptor. The same controls were included in the PM fractions for C2C12 cells displayed in Supplementary Fig. 8n. For the primary human white adipocytes (Supplementary Fig. 3r), we again confirm the absence of cytosolic contamination by blotting for HSP90 and b-actin in the WCL and PM fractions.

c) Some figures don't show the number of individual experiments/mice as indicated in the legend (e.g. food intake in SupplFig. 7b F4 fraction).

We apologize for the mistake in the figure legend. In line with the comments from reviewer 4, we have now reported the exact n number for each treatment group instead of the currently displayed ranges for all groups. In addition, we have carefully double checked and corrected any potential mistakes in the reporter n-number in the figure legends.

d) Multiple data sets show a p-value that is close to significance, but is not <0.05 (e.g. Suppl. Fig 8f, Fig. 5d). But data are described as significant differences in the main text

We apologize for the misinterpretation. We have corrected and rephrased the corresponding sections more carefully and if necessary, indicate exact p-values.

e) PKC-ERK1/2-GLUT1 pathway was explored with PNe, but not with GE. Here the authors focused on AKT pathway only. Why?

In Fig. 2 we focused on the effects of chronic PNe (72 hours) treatment on adipocytes and discovered that it increases glucose uptake through the upregulation of Glut1 mRNA expression and protein levels. In Fig. 3, we aim to unravel the intracellular signaling cascades, which lead up to the elevated Glut1 protein levels. Here, we demonstrated that PKC-ERK1/2 signaling is required to upregulate Glut1 transcription and inhibition of this signaling axis diminishes the effects on Glut1 mRNA after 3 and 4 hours of PNe stimulation. This consequentially results in an absent effect on glucose uptake after 16 hours of PNe treatment. The 16 hours time point was selected here to ensure that sufficient Glut1 mRNA is translated and the total Glut1 pool is increased.

Notably, we did not observe any effects of GE on Glut1 mRNA expression after the 3 days of treatment (Supplementary Fig. 10c, f, g), suggesting that GE does not induce transcriptional events that would lead to an upregulation of Glut1. We therefore decided against investigating the PKC-ERK1/2 axis in response to GE. However, to address the reviewers' comments and in line with the experiments performed in Fig. 3 for PNe, we have now also analyzed phosphorylation of ERK1/2 in response to 30 min and 180 min of GE stimulation at 1 μM and 10 μM. Consistent with the absent effect of GE on glucose uptake (Supplementary Fig. 10a, i, h) and Glut1 mRNA expression, GE treatment did not induce ERK1/2 phosphorylation.

Figure 1 Effect of GE on ERK-signaling. Immortalized brown adipocytes treated with 1 or 10 μM of GE for (a) 30 min and (b) 180 min. n=3 per condition.

f) AKT-phosphorylation analysis was adapted based on reviewer's comment in Fig. 2, but not in Fig. 6. Further, why different AKT phosphorylation sites examined in fat vs. muscle (Suppl. Fig. 10)?

We thank the reviewer for this careful observation. We apologize for failing to update the analysis for the phosphorylated AKT sites in Fig. 6. We have now consistently analyzed both AKT phosphorylation sites (AKT T308, AKT S473) for adipose tissue as well as skeletal muscles and normalized the data to total AKT levels.

For the adipose tissue, we have re-run and re-analyzed all the western blots using the same samples as in the previous manuscript with an optimized transfer protocol. After normalization to total AKT, phosphoAKT T308 levels were not different in the iWAT or iBAT in response to GE treatment. AKT phosphorylation at S473 is increased in response to GE after normalization to total AKT levels, supporting increased insulin sensitivity in brown and white adipose tissues. We observe a small increase in total AKT levels in the iBAT, which is not significantly different between GE versus HFD-fed mice.

In the skeletal muscles, phosphoAKT T308 is unchanged in soleus and gastrocnemius muscles as shown in the previous version. While AKT S473 phosphorylation is unchanged in the soleus muscle, it is downregulated in the gastrocnemius muscle ($p=0.044$). However, our clamp data could not detect any differences in skeletal muscle insulin sensitivity in GE versus HFD treated mice (Fig. 7n), suggesting that this downregulation does not affect muscle skeletal uptake. Moreover, the samples were collected in the random fed state. Therefore, we cannot exclude that difference in AKT phosphorylation may depend on food intake dependent insulin release and receptor activation. We have added this improved and completed data to Fig. 6 and Supplementary Fig. 10 and we have adapted the results/discussion section to reflect this point.

g) Fig. 3 uses PMA, no description in main text

We apologize for not introducing PMA in the main text. PMA is a PKC agonist, which was used as positive control for the activation of the PKC signaling pathway.

h) minor: Suppl. Fig. 3 beige or brown fat?; Suppl. Fig. 2e/m glycolysis typo; line 186 typo in phospho-PKC

We have corrected the typos in Suppl. Fig. 2e/m.

Human beige adipocytes is correct and we have corrected the graph labelling accordingly. The human multipotent adipose-derived stem cells (hMADS) are differentiated into mature adipocytes with a standard protocol and only during the last days of cultivation, a beige phenotype can be induced with Rosiglitazone treatment. Please refer to Balazova et al., 2021 Nat Com, PMID:34880217 and Elabd et al., 2009 Stem Cells, PMID: 19697348 for details on hMADS differentiation

Points raised by reviewer 4

This is a statistical review. I do not comment on the soundness of the science.

Although experimental conditions and statistical methods are reasonably well-documented in the figure legends, things could be improved. Below, I provide some comments about each figure's figure legend.

We greatly appreciate the reviewer's feedback and the concerns raised about certain analyses that were performed. We agree that the figure legends were difficult to understand regarding

the statistical tests and it was not always clear which test was used for which panel. To address this issue, we now clearly state which analysis was used for which panel without ranges. In addition, for each graph we have now indicated the exact n-number per condition/treatment and removed all ranges.

Figure 1's figure legend.

1. In panels b, c, and d, the n/group is not provided. Please provide the exact n of each group, especially if the two groups have different sample sizes.

We have now provided the exact n-number for each group in figure legend 1.

2. In panels g-j, the sample size is given as a range, n=5-8 per group. That is too vague. Please say which group has n=5, which group has n=6, which group has n=7, and which group has n=8. If the n varies from panel to panel, please document that also.

We have now provided the exact n-number for each group for panels g-j.

3. In panels l-m, it says n=7-8 per group. That is too vague. Please say which group has n=7 and which group has n=8.

We have now provided the exact n-number for each group for panels l-m.

4. Please do likewise for panels n and o, i.e., please provide the exact n of each group.

We have now provided the exact n-number for each group for panels o-i

5. It is unclear from what the authors wrote which specific panels were analyzed using 2-way ANOVA and which specific panels were analyzed using repeated-measures 2-way ANOVA. Please clarify which procedures were used on which panels.

We apologize for the confusion caused by the figure legend. In the original manuscript, repeated measures 2-way ANOVA was used in panels b, e, l and n. A standard two-way ANOVA was applied in panel d, to evaluate the effects of the diet on body composition.

Referring to the reviewer's point 7 for Fig.1, we confirm that the time curves in panels e and l correspond to the area of the curves displayed in (f) and (m), respectively. To address the concern regarding redundancy in the analysis, we have now removed the statistical test from panels e and l. For panel b, we have used a repeated measures two-way ANOVA with a diet x time interaction and Sidak's post-hoc test for each time point. Please refer to point 6 for more details on panel n.

In the new manuscript version, the tests and figure legend have been adjusted according to the reviewer's suggestions as follows:

"Results are reported as mean \pm SEM. Two-tailed student's t-test for comparisons between two groups was applied in c, f, g, h, l, j, m, o, r, s, t, u. Two-way ANOVA with Sidak's post-hoc test was applied in d. Repeated measures two-way ANOVA with diet x time interaction and Sidak's post-hoc test for each time point was applied in b. Repeated measures mixed-effects analysis with diet x time interaction and Sidak's post-hoc test for each time point was applied in n."

6. In panel n, do the data points at Time = 30 minutes come from the same mice as the data points at Time = 0 minutes? If Yes, then (1) spaghetti plots would probably be better than dynamite plots for showing the data, and (2) this probably calls for repeated-measures 2-way ANOVA instead of ordinary 2-way ANOVA. The repeated-measures ANOVA should include a test of the Diet-x-Time Interaction along with tests comparing diet groups at each time point.

We thank the reviewer for the advice on the analysis. Yes, the data points at t=0 and t=30 minutes come from the same mice. In the revised manuscript, we have corrected our analysis accordingly and exchanged the dynamite plot with a spaghetti plot. One value in the HFD group is missing at t=30 due to blood sampling issues. Therefore, we have now performed a Mixed-effects analysis with repeated measures including the Diet x Time Interaction and comparisons between diet groups at each time point. Sidak's multiple comparisons test was then used to identify significant differences. The output of this analysis is now summarized in Table 1.

7. In panels b, e, and l, it is not clear what is being analyzed using repeated-measures 2-way ANOVA. Are the authors testing the main effect of Diet (i.e., HFD vs PNe)? Are they testing the main effect of time? Are they testing the Diet-x-Time Interaction? Or are they merely testing the simple effects of Diet at each individual time point? Please add some clarity. However, the time curves of panel e are connected to the AOCs of panel f, and the time curves of panel l are connected to the AOCs of panel m. This means that the t-tests comparing AOCs between Diet groups is 95% equivalent to testing the main effect of Diet using repeated-measures 2-way ANOVA on the time curves, which in turn means there may be some redundancy in the analyses. This is why it is important to describe what tests and what comparisons are being conducted using repeated-measures 2-way ANOVA.

We thank the reviewer for this insightful comment. Yes, the area of the curves (f and m) are calculated from the time curves in panels e and l. To prevent the mentioned redundancy of the analysis, we provide now only the statistical analysis for the AOC, to evaluate the overall effect of the treatment on the glycemic load and consistent with the recommendation from experts in the field (Virtue et al., 2021). As for panel b, we have now performed repeated measures two-Way ANOVA including a diet x time interaction with Sidak's post hoc test to compare the effect of the diet at each individual time point.

Virtue, S., Vidal-Puig, A. GTTs and ITTs in mice: simple tests, complex answers.

Nat Metab **3**, 883–886 (2021). <https://doi.org/10.1038/s42255-021-00414-7>

Figure 2's figure legend.

8. In every location where the n/group is given as a range, please replace the range with the exact n of each group. Please do not say n=6/group unless both groups have exactly n=6. Please also clarify whether the statement "(n=6)" for panels c-g means n=6 for each group or total n=6 for both groups combined.

We have now provided the exact n-number for each group for each panel. As for panels c-g, the following exact n-numbers apply. The reported n applies to each treatment (Ctrl vs PNe) and insulin-condition (0 vs 100 nM). To be specific, the following sample numbers per group were evaluated:

(3d) Ctrl n=6, PNe n=6 for 0 nM and Ctrl n=6, PNe n=6 for 100 nM insulin.
 (3e) Ctrl n=5 PNe n=6 for 0 nM and Ctrl n=6, PNe n=6 for 100 nM insulin.
 (3f) Ctrl n=6, PNe n=6 for 0nM and Ctrl n=6, PNe n=6 for 100 nM insulin.
 (3g) Ctrl n=6, PNe n=6 for 0 nM and Ctrl n=6, PNe n=6 for 100 nM insulin.

Of note, the blots displayed in (c and h) show only 1 independent experiment but the data points in the respective quantifications are from two independent experiments, that were pooled. The additional western blots are displayed in the source data provided with the paper.

9. Because Figure 2 shows in vitro experiments, it's not clear what the sample sizes represent. The "Quantification and statistical analysis" section of the Methods say that each

experiment was independently reproduced 2-3 times, but Figure 2 looks like it shows results from only single reproductions of each experiment. Please elaborate.

The datapoints shown in Figure 2 and likewise in Figure 3 display the individual replicates that were measured in 2-3 independent experiments as described in the methods section. Each independent experiment was performed in 12-well or 24-well plates with multiple replicates for each experimental condition. The datapoints displayed are therefore a pool of all datapoints from the 2-3 experiments. To improve clarity, we have added one additional sentence in the methods section.

10. The legend entry for panels a-b, glucose uptake in basal and insulin-stimulated cells, says that panel b is 3T3-L1 cells and panel c is iBAs. This does not look right. Please confirm.

Yes, there was a mistake in the figure. Data for 3T3-L1 cells are in panel a and for iBAs in panel b. We have corrected the figure legend.

11. Figure 2's legend says that Two-way ANOVA was applied, with Sidak's post hoc test on a number of panels, and with Dunnett's post-hoc test on panel's m and n. The Quantification and statistical analysis section says instead that One-way ANOVA with these post-hoc tests was used. Please rectify the discrepancy.

We apologize for this discrepancy and inconsistency; we agree that the figure legend was difficult to follow and not clear. Considering all the feedback and recommendations from the reviewer on the statistical analysis, we have updated the figure legend with the revised statistical analyses. Please refer also to points 12 and 13 related to Fig. 2 for more detailed explanation of the individual adaptations. In the revised manuscript, the figure legend was adjusted accordingly. We have removed detailed information on the statistical tests in the methods section and refer to the individual tests listed in the figure legends for clarity.

12. Panels m and n show the structure of a one-factor experiment.

We thank the reviewer for pointing this out and we agree that the PNe dose was the only factor changing in the experiment. We have now separated the graph into different metabolic read outs with the dose as only variable. In this experiment, our interest was not to compare differences between the doses but to investigate whether any of the tested PNe doses had an effect compared to untreated controls (= 0 ug/mL). Therefore, we applied a One-Way ANOVA with Dunnett's post-hoc test to compare each PNe dose against the control. Moreover, we have adjusted the panels in Supplementary Figure 2j-o and Supplementary Fig. 3d-g, which follow the same experimental design.

13. Although panels a, b, d-g, and i-l show the structure of two-factor experiments, the fact that Sidak's method was used to compare all groups pairwise suggests strongly that the experiments were analyzed using a one-way layout.

We thank the reviewer for this comment and would like to clarify our statistical analysis. We agree that the visual structure of the mentioned graphs (a, b, d-g, i-l) reflect a two-factor design. However, our primary interest was to evaluate the effect of the treatment (PNe) on glucose uptake rates within each insulin condition (basal or insulin-stimulated). We were not interested in the main or interaction effects of insulin on the examined parameter. Notably, the data does not reflect repeated measures in basal and insulin-stimulated conditions, but two independent conditions measured in different cells/wells. Accordingly, we analyzed each condition separately using a one-way layout. We have now applied multiple t-test's with Holm-Sidak multiple comparison adjustments as recommended by GraphPad Prism for panels a, b, d, e, f, g, i, j, k, l, x, y.

14. What do the beige and white colors mean in panels t and u?

We apologize for not providing a color coding legend. The white is inguinal or brown adipose tissue from control high fat diet (HFD)-fed animals, beige are PNe-fed mice. We have added this legend in the new Fig. 2x (former Fig. 2t)

Figure 3's figure legend.

15. In panels e through g, please do not describe the n/group using ranges. Please provide the exact n of each and every group.

We have now provided the exact n-number for each group for each panel.

16. Where it says that one-way ANOVA with Holm-Sidak's or Tukey's adjustments between all groups was applied in d, g, and f, please indicate which panels had Holm-Sidak's method applied to them versus which panels had Tukey's method applied to them.

We apologize that the figure legend was difficult to follow and incorrect. We have now corrected the figure legend and kindly refer to the next comment under point 17 for a detailed explanation. In brief, all data were analyzed with a one-way ANOVA using Tukey's multiple comparison adjustments between all groups.

17. Please also explain why Holm-Sidak was chosen over regular Sidak, why Tukey was chosen over regular Sidak, and what motivated using Holm-Sidak sometimes and Tukey other times. Could p-hacking have been involved?

We thank the reviewer for the careful reading and for pointing out the discrepancy in the figure legend. Upon review, we acknowledge that the explanation in Figure legend 3 was not adequately updated after the first round of revision and therefore does not fully reflect the analyses performed. We apologize for this oversight and the resultant confusion. All graphs displayed in Figure 3 (panels c, d, f and g) were analyzed with a One-way ANOVA with Tukey's post-hoc test as all groups were compared against each other. Notably, we also compared all time points in Figure 3g, which justifies the use of Tukey's test in this case as well. We wish to emphasize that no p-hacking or selective reporting was performed. The discrepancy was due to a poor updating of the figure text legend after the final statistical analysis was performed. For transparency we would like to summarize that only the biologically most relevant comparisons are displayed in the graphs to prevent visual overcrowding from too many brackets in the graphs, The exact p-values for all pairwise comparisons including those not marked with brackets are provided in table 1.

Figure 4's figure legend.

18. For every panel, please do not describe the n/group using ranges. Please provide the exact n of each and every group.

We have now provided the exact n for each and every group in Figure 4.

19. I get the impression that panel a defines the color-to-treatment mapping used in panels b through j, and that panel k defines the color-to-treatment mapping used in panels l through s. Please confirm if this is true.

Yes, we can confirm that the treatment mapping in Fig. 4a applies to the results displayed in panels b-j and treatment mapping in Fig. 4k applies to the results displayed in panels l-u (former l-s).

20. In panels c, e, and g, which line color and line pattern corresponds to the HFD control? Inserting a line legend inside each panel might help the reader.

The black line is the HFD control. As suggested by the reviewer, we have added a line legend inside each panel .

21. I do not understand the sentence, “Two-way ANOVA with Tukey’s post-hoc test between all groups was applied in m-n (between groups)”. In panel m, and likewise in panel n, I see 6 groups, but they are clustered into 3 basal groups and 3 insulin groups. And as far as I can tell, each panel must have had two separate Tukey’s procedures applied to it, one to the basal cluster (3 groups) and one to the insulin cluster (3 groups). If that is what the authors did, then they will want to Sidak-adjust the significance level of each Tukey procedure if they want to truly control each panel’s overall Type-I error rate to 5%.

We thank the reviewer for this valuable comment, and we apologize for the lack of clarity. In both panels (m and n), our primary interest was not comparing the two conditions (basal vs insulin stimulated state). Instead, our main goal was to compare the treatment effect of F4 against HFD in the respective conditions. MET was introduced as a positive reference group, which is known to improve metabolic health as compared to the HFD control condition. We acknowledge that the current data presentation gives the impression of a two-factor design and we have indeed performed two separate Tukey’s procedures: one for the basal cluster and one for the insulin cluster. As suggested by the reviewer, this would require the recommended Sidak-correction of the significance level to correctly adjust for all comparisons.

To clarify, we have now made the following revisions: We have separated panels m and n based on their condition (basal vs. insulin) to empathize that comparisons were performed only within each condition. Each panel now shows the results of a one-way ANOVA with Tukey’s post-hoc test between all treatment groups within each condition. Moreover, this separated representation of basal and insulin-stimulated conditions is consistent with the literature reporting data from hyperinsulinemic-euglycemic clamp experiments (e.g. PMID:29769440, 26858428)

22. Please discuss further the mixed-effects analysis with Dunnet’s post-hoc test compared to HFD that was applied in panel b. I see eight different HFD controls, one for each week of DI, and I see a total of $8 \times 5 = 40$ groups spread out over the 8 weeks. Did the authors perform one single Dunnet’s test on all 40 groups combined? Or did the authors perform 8 separate Dunnet’s tests grouped by week of DI? If the authors did the latter, then they will want to Sidak-adjust the significance level of each of their 8 Dunnet’s tests if they want to truly control panel b’s overall Type-I error rate to 5%.

We thank the reviewer for this helpful comment. We performed 8 separate Dunnet’s test to compare the effect of each diet type against the HFD-controls, grouped by week of DI. According to our understanding, we have performed a total of $8 \times 4 = 32$ comparisons (8 weeks, 4 treatments compared against HFD control group). We appreciate the reviewer’s recommendation to apply a Sidak-adjustment to truly control for the type-I error rate. In response, we have re-analyzed the data in panel Fig. 4b and performed a mixed-effects analysis with Sidak’s multiple comparison test, comparing each treatment group to the HFD control at each time point. A single family of comparisons was defined across all time points and treatment groups to control for the family-wise error rate using Sidak correction. The same analysis was performed in Supplementary Fig. 8b.

Figure 5's figure legend.

23. For panels c, d, g, h, and i, please do not describe the sample sizes per group using ranges. Please provide the exact n of each and every group.

We have provided now all the exact n number for each and every group.

24. Regarding the repeated measures two-way ANOVA with Dunnett's post-hoc test that was applied in panels c and e, it appears that each panel had six separate Dunnett's tests grouped by time in minutes, each of which used the time group's HFD control as its time-specific reference group. If the authors want to truly control the Type-1 error per panel to 5% overall, then they will wish to Sidak-adjust the significance level of each Dunnett's test to control for the fact that they are doing six separate Dunnett's tests per panel.

This interpretation of the analysis is correct. We would like to refer to the reviewer's point 7 related to Figure 1, where the structure of the data/panel is the same as in the panels discussed here (Fig. 5 c and e). The blood glucose measurements over time were used to calculate the area of the curve in panels Fig. 5d and f. We have now removed the statistical analysis for the panels c and d to avoid the mentioned redundancy in analysis raised under point 7. The corresponding area of the curves (Fig. 5d and f) were analyzed with a One-Way ANOVA using Dunnett's post-hoc test against HFD-controls.

25. In panel 5i, the p-value bracket connects HFD to IX. I think it was meant to connect HFD to GG. Please check.

We have checked the p-value bracket and confirm that it should connect HFD and GG. We have corrected this mistake.

Figure 6's figure legend.

26. In every place where the sample sizes are described as n=9-10 per group, please get rid of the ranges and please provide the exact n of each and every group.

We have provided now all the exact n number for each and every group.

27. Regarding the western blots, I see panels n and o, but not panels p and q.

We have corrected this mistake and adjusted the panel labelling as more blots (phosphoAKT T308) were added during the revision.

Figure 7's figure legend.

28. In every place where the sample sizes are described either as n=8-9 per group or as n=7-9 per group, please get rid of the ranges, and please name which groups have exactly n=7, which groups have exactly n=8, and which groups have exactly n=9.

We have provided now all the exact n number for each and every group.

29. For panels f and g, it says (A) that 2-way ANOVA with Tukey's post-hoc test was applied, and (B) that all comparisons were performed between all groups. If both statements are true, then why in panel f is there no p-value bracket comparing basal HFD to insulin HFD, no p-value bracket comparing basal GE to insulin GE, and no p-value bracket comparing basal MET to insulin MET? Those pairwise differences clearly are larger than the differences between members of the insulin subgroup. Could it be that only some comparisons were performed between all groups? In other words, could it be that Tukey's procedure was applied

twice in panel f, once to the basal subgroup and once to the insulin subgroup, rather than just once to all six groups? Please clarify.

Yes, this assumption is correct and we apologize for the inadequate reporting of the analysis. As described by the reviewer, we were interested in the treatment effects within one condition (basal vs insulin-stimulated) and we compared all groups within each condition, which resulted in two Tukey's post-hoc analyses. We now separated panels f and g into two panels based on their insulin-condition. Consistent with the analyses performed previously in the one-factorial design panels from the same experiment (now panels k-o, original panels j-k), we applied now a One-Way ANOVA with uncorrected Fisher's LSD post-hoc test. The Tukey's adjusted p-values are listed in the updated table 1. Please refer to the method section for details, where this approach was justified in the previous version of the manuscript.

Point-by-point Response to Reviewer's Comments – Final Revision Round 3

“Genkwanin glycosides are major active components in *Phaleria nisidai* extract mediating improved glucose homeostasis by stimulating glucose uptake into adipose tissues”

We thank reviewer 3 for his positive feedback and we greatly appreciate the effort, time and diligence reviewer 3 invested to improve our manuscript. We have addressed the remaining comments from reviewer #4. If necessary, we highlighted relevant changes in red in the manuscript text.

Points raised by reviewer #4

I think the authors successfully addressed my previous questions and comments. I have a few remaining questions.

1. Line 576. "(k) Experimental design for results in k-o." I think the authors meant panels l-o.

Yes, it should say panels l-o. We have corrected this mistake.

2. Line 582. "(p) Experimental design for results in k-o." I think the authors means panels q-u.

Yes, it should say panels q-u. We have corrected this mistake.

3. Lines 583 and 585. The authors say iBAT in line 583 but switch to BAT in line 585. Are iBAT and BAT the same or different?

We apologize for the inconsistency. iBAT and BAT are the same. We have corrected the figure legend accordingly.

4. Lines 588-589. "Repeated measures two-way ANOVA with diet x time interaction and Sidak's post-hoc test for each time point was applied in b." Was the same method used for panel e?

No, we did not apply any statistical test on panels e and l. We have calculated the respective areas of the curves (AOC) displayed in panels f and m. To avoid redundancy of statistical tests and as suggested by Virtue & Puig, we tested for significant differences in the AOC.

Virtue S, Vidal-Puig A. GTTs and ITTs in mice: simple tests, complex answers. *Nat Metab.* 2021 Jul;3(7):883-886. doi: 10.1038/s42255-021-00414-7. PMID: 34117483.

5. Figure 1, panel o. The two bars are almost the same height, yet there's a p-value bracket connecting them that indicates $p < 0.05$. Also, the bar height of PNe goes between the 2nd-highest & 3rd-highest data point (instead of being near the 5th-highest) when there's no high-glucose outlier pulling the mean up. I suspect that the p-value bracket is correct, but that the PNe bar height may be wrong. Please look into this.

We thank the author for the careful inspection of the panel. We have checked the source data and confirm that the p-value bracket is correct. The PNe bar graph was incorrectly displayed. We have corrected this error and replaced the graph.

Line 619, which says "(w-y) (w) Western blots of GLUT1 in iWAT (HFD n=4, PNe n=5) and iBAT (HFD n=6, PNe n=6) or (y)". I think the label "(x)" was accidentally omitted from the middle of that line.

We apologize for the incomplete figure legend related to w-y and have added the missing information.

6. Figure 2, panel u. The Y-axis label says relative mRNA expression, and for every gene except Pparg, the control group's bar heights are 1.0, indicating that relative mRNA expression must mean relative to the control group's average expression. Which raises the question of, why is the control group's bar height different from 1.0 in the case of Pparg expression.

We are thankful for pointing out this detail and we have re-analyzed the raw data. Upon review, we identified an error in the excel formula, where an incorrect cell was used for normalization. The expression of Pparg is now correctly normalized to the ctrl, consistent with all other genes shown. This correction adjusts the height of the control group's bar height to 1.0.

7. Figure 2, panel v. The Y-axis label says relative mRNA expression, but unlike in panel u, none of the control group's bar heights are equal to 1.0. All of them are lower than 1.0. Why is that? Were the expression data in panel v expressed relative to something different from the control group?

We thank the reviewer for this detailed observation that we have missed so far. Comparable to comment 6, we have noticed an error in the excel formula which in covered too many cells to calculate the average of the control group. We have corrected this mistake, which adjusts the height of the control group's bar height to 1. We would like to emphasize, that this error did not affect the reported biological findings.

8. Lines 623 and 624. "Two-Way ANOVA with Sidak's post-hoc test was applied in u, v." Applying 2-way ANOVA to the data in those two panels requires meeting the equal-variance assumption across all groups, which in this case means across all genes as well as across PNe vs Ctrl groups. How reasonable is it to assume that different genes have the same variances in gene expression? Perhaps it might be more reasonable to assume that different genes have different variances in gene expression, and accordingly to (1) compare PNe to Ctrl via t-test within each gene, and (2) perform the Sidak adjustment (or Bonferroni adjustment) by hand on all the t-test results.

We thank the reviewer for this suggestion, and we have corrected our analysis for panels u and v accordingly. As suggested by the reviewer, we have used t-test to compare PNe to Ctrl for each gene and subsequently performed Sidak adjustment on the t-test results. We have adjusted the figure legend with the updated statistical analysis.

9. Figure 3, panels l through p. I had the impression that all the data in these figure panels came from the same mice, not different mice, but the sample sizes described in lines 660 through 665 do not stay the same from panel to panel. Please explain.

We thank the reviewer for this observation. Based on the line numbers (660-665), we believe the reviewer may be referring to Figure 4, not Figure 3. The data shown in panels l to u were all acquired in the same cohort of mice using the experimental design outlined in Fig. 4k. At the end of the dietary intervention, we performed hyperinsulinemic-euglycemic clamp studies, which are technically very challenging experiments. Here, the mice are implanted in a catheter and connected to infusion pumps for the sequential delivery of two types of radioactive tracers, insulin and glucose for various durations and in specific orders. Throughout this process, samples were collected for the analyses shown in panels (q-u). Due to the complexity of this procedure, occasional technical issues such as difficulties in blood sampling, leaky or blocked catheters can result in insufficient or missing samples required for downstream processing and measurements. As a result, and depending on the time at which the problem occurred, not all read outs could be obtained from every mouse, which accounts for the varying samples sizes reported.

10. Figure 6, panel s. The Y-axis label says "AKT^{p473}/AKT" when I think the authors meant to say "AKT^{pS473}/AKT". Also, the Y-axis label for panel s does not say what the S473 phosphorylation ratios are relative to, whereas the corresponding label for panel t says relative to Ctrl. That seems odd. Finally (lines 704-705), the sample sizes of the HFD mice change from panel s to panel t even though they are different tissues from the same mice. Why is that?

We believe that the reviewer refers to panel n regarding the Y-axis labeling. We have corrected the missing information in panel n (AKT^{pS473}/AKT). We apologize for the missing information regarding the relative ratios and the inconsistency. The bands were quantified and normalized to HFD ctrl group for all displayed western blots. For clarity, we have now re-labelled all Y-axes and state that the data is expressed relative to the HFD group.

We think that the reviewer refers to panels n and o regarding the inconsistent sample size of different tissues from the HFD mice. In panel n, HFD n=9 for the iBAT whereas in panel o, n=10 for the iWAT. For the iBAT, we excluded one outlier based on the Grubbs test performed by Graph Pad Prism. The removed outlier is included in the source data file for the main figure.

11. Line 724, referring to Figure 6 panels j-n. The sample sizes were omitted from description of panel l, the iBAT panel.

We apologize for the missing information in Figure 7. We have added the exact n numbers for the iBAT panel in I.